# SGD-Based Knowledge Distillation with Bayesian Teachers: Theory and Guidelines

**Itai Morad[1], Nir Shlezinger[1] and Yonina C. Eldar[2]**
[1]School of ECE, Ben-Gurion University, Be'er Sheva, Israel
[2]Faculty of Math and CS, Weizmann Institute of Science, Rehovot, Israel
`itaimor@post.bgu.ac.il,nirshl@bgu.ac.il,`
`yonina.eldar@weizmann.ac.il`

## Abstract

Knowledge Distillation (KD) is a central paradigm for transferring knowledge from a large teacher network to a typically smaller student model, often by leveraging soft probabilistic outputs. While KD has shown strong empirical success in numerous applications, its theoretical underpinnings remain only partially understood. In this work, we adopt a Bayesian perspective on KD to rigorously analyze the convergence behavior of students trained with Stochastic Gradient Descent (SGD). We study two regimes: $(i)$ when the teacher provides the exact Bayes Class Probabilities (BCPs); and $(ii)$ supervision with noisy approximations of the BCPs. Our analysis shows that learning from BCPs yields variance reduction and removes neighborhood terms in the convergence bounds compared to one-hot supervision. We further characterize how the level of noise affects generalization and accuracy. Motivated by these insights, we advocate the use of Bayesian deep learning models, which typically provide improved estimates of the BCPs, as teachers in KD. Consistent with our analysis, we experimentally demonstrate that students distilled from Bayesian teachers not only achieve higher accuracies (up to +4.27%), but also exhibit more stable convergence (up to 30% less noise), compared to students distilled from deterministic teachers.

## 1 Introduction

Knowledge Distillation (KD) Hinton et al. (2015) is a fundamental technique in machine learning, widely used for model compression, transfer learning, and improving generalization Gou et al. (2021); Yim et al. (2017). A core idea in KD is to transfer knowledge from a "teacher" model to a (typically smaller) "student" model by training the student to match the teacher's output probabilities rather than categorical outputs (one-hot labels). This softened supervision has been shown to lead to improved performance across a range of tasks Mansourian et al. (2025). Consequently, a substantial body of research has focused on designing KD mechanisms, including strategies for dynamic temperature scaling, feature-based distillation, and task-aware teacher-student matching Zhu et al. (2024); Wang et al. (2024), typically assuming a large teacher which is optimized to maximize its own performance.

Despite its widespread adoption, the theoretical foundations of KD remain only partially understood Phuong & Lampert (2019). In particular, the impact of the teacher's output probabilities on the student's optimization trajectory and generalization has not been fully characterized. While insights have been developed for special cases such as self-distillation Safaryan et al. (2024), a principled understanding of how KD affects common learning algorithms, such as Stochastic Gradient Descent (SGD) and its variants, is still lacking. Recent works have begun to explore KD through a Bayesian lens Menon et al. (2021); Ye et al. (2024), interpreting the teacher's outputs as (possibly noisy) estimates of the true class posterior. This probabilistic perspective gives rise to the possibility of analyzing relatively unexplored aspects of KD; particularly in terms of the dynamics of SGD-based learning, as well as the statistical calibration of the teacher and its influence on the student.

**Contributions** Motivated by the Bayesian viewpoint of KD, our work provides a rigorous analysis of the interaction between probabilistic supervision and SGD-based learning. Our analysis considers two regimes: $(i)$ supervision with the Bayes Class Probability (BCP), i.e., the exact posterior probabilities,

which correspond to a perfect teacher from the Bayesian perspective on KD; and $(ii)$ supervision with noisy estimates of the BCP, i.e., a realistic imperfect teacher. Based on this modeling, we are able to show variance reduction compared to one-hot supervision in SGD-based learning. Through a numerical study, we show that this variance reduction translates into improved performance of the trained student.

Based on our analysis, which indicates that the effectiveness of KD depends on how well the teacher approximates the BCP (i.e., how well-calibrated the teacher is), we advocate the use of Bayesian deep learning models as teachers Gawlikowski et al. (2023) in KD. Bayesian deep learning brings forth a key advantage, as it typically results in better-calibrated probabilistic predictions, thereby producing more faithful approximations of the BCPs Jospin et al. (2022). We show this benefit can be effectively harnessed in KD, through two complementary strategies: $(i)$ training the teacher directly using Bayesian learning techniques, or $(ii)$ converting an existing deterministic (frequentist) pre-trained teacher into a Bayesian model via posterior approximation techniques. Our experimental study demonstrates the effectiveness of this approach, both empirically validating our theoretical analysis, as well as showing the usefulness of Bayesian teachers. Our results show consistent improvements in accuracy and variance reduction when distillation is performed with properly calibrated teachers. These empirical findings align with our theoretical analysis, supporting our claim that leveraging Bayesian teachers in KD improves both the convergence and generalization of SGD-based learners.

## 2 RELATED WORKS

**Theoretical Justification for KD**    Since the introduction of KD Hinton et al. (2015), several works focused on explaining its mechanisms from different theoretical perspectives. A common approach attributes its usefulness to the soft estimates of the teacher regularizing the learning procedure Dong et al. (2019); Yuan et al. (2020). Recently, Cha & Cho (2025) provided a minimal working explanation of KD through a precision–recall trade-off. Huang et al. (2021) framed KD as a trade-off between knowledge inheritance and exploration. Phuong & Lampert (2019) provided generalization bounds and identified factors explaining the success of KD, namely data geometry and optimization bias, and Lopez-Paz et al. (2015) connected KD to privileged information. In contrast to these works, both our theoretical analysis and our algorithmic contributions are based on a Bayesian perspective on KD, which facilitates characterizing the learning dynamics, as well as unveil how one can enhance a teacher model for KD.

**Bayesian Perspective on KD**    Menon et al. (2021) inspected the soft estimates of the teacher from a Bayesian perspective, aiming to provide theoretical justification for its empirical success. There, it was shown that learning from accurate BCPs can reduce the variance of the student's objective, and excess risk bounds were derived, depending on the $\ell_2$ distance between the teacher's predictions and the true BCPs. Similarly, Dao et al. (2021) provided generalization bounds on the student including terms such as the $\ell_2$ distance between the true BCPs and the teacher's predictions. While our work is motivated by this Bayesian viewpoint, we differ in focus: rather than analyzing the statistical properties of the *risk*, we study how the probabilistic teacher outputs affect the *behavior of learning algorithms*. Safaryan et al. (2024) is, to our knowledge, the first to examine the impact of distillation on SGD. However, their analysis considers specific scenarios such as self-distillation or having the student be a compressed version of the teacher, for which a dedicated gradient approximation is formulated. In contrast, our results hold for arbitrary teachers, do not rely on gradient approximations, and explicitly model the teacher's predictions as BCP estimates. Furthermore, our analysis reveals that learning from BCPs holds the desired interpolation property under mild assumptions, which allows student Neural Networks (NNs) to generalize in settings where learning from one-hot labels results in overfitting.

**Distilling from Calibrated Teachers**    Beyond theory, our work also contributes to the design of teacher models for KD, advocating for the use of Bayesian teachers to enhance calibration. Recently, Kim et al. (2025) and Fan et al. (2024b) recognized calibration as a key driver in KD and suggested algorithms to enhance calibration of deterministic models. Related recent works Ye et al. (2024), Hamidi et al. (2024) aimed to improve teacher calibration by modifying the training objective, encouraging predictions that better approximate the true BCPs. While these approaches operate by altering loss functions or improving calibration of a deterministic model, we instead adopt the Bayesian deep learning paradigm, which naturally yields calibrated predictions and quantifiable uncertainty measures. This both improves the accuracy of the distilled student and provides an indication of the calibration of the teacher. Beyond calibration, recent works have focused on how the divergence

used in KD affects how probability mass is transferred from teacher to student. ABKD Wang et al. (2025a) develops an $\alpha$–$\beta$ divergence framework that balances mode-concentration effects to improve distribution matching, and sequence level $f$-divergence methods Wen et al. (2023) similarly examine how different divergences influence the way probability mass is transferred from teacher to student in sequence models. In contrast to these approaches, rather than modifying the divergence, we focus on how the quality of the teacher's probability estimates impacts student performance. Other works have considered the intersection of Bayesian models and distillation, but with fundamentally different goals. For instance, Bulò et al. (2016), Gurau et al. (2018), and Korattikara Balan et al. (2015) focused on distilling Monte Carlo dropout ensembles to reduce inference cost or to improve uncertainty estimation in the student model. Similarly, Lee et al. (2023) studied self-distillation under dropout. In contrast, our work focuses on leveraging Bayesian teachers as a means to enhance student learning performance as a direct consequence of our theoretical analysis.

## 3 ANALYSIS OF SGD-BASED LEARNING FROM PROBABILITY ESTIMATES

This section characterizes the impact of supervision with BCP estimates on SGD-based learning. Specifically, we provide convergence guarantees for both cases of supervision with perfect and with noisy probability estimates, and compare the results to the standard case of learning from one-hot labels. We show that under common assumptions, learning in all cases converges in expectation to the same model, but with potentially better convergence rate and lower stochastic noise at the optimum when supervising with probability estimates. Intuitively, as the probability estimates become noisier, the convergence of SGD worsens, up to a point where it is no longer beneficial to supervise with probability estimates. The proofs of the results presented in this section are delegated to Appendix C.

### 3.1 PRELIMINARIES

**Learning Framework**  We consider the supervised learning of a classification task, where each input $\boldsymbol{x} \in \mathcal{X} \subseteq \mathbb{R}^d$ is associated with one of $K$ classes, represented by the one-hot label $\boldsymbol{y} \in \mathcal{Y}$. The inputs are related to the labels via a data generating distribution $\mathcal{P}$ Shalev-Shwartz & Ben-David (2014), which dictates *true BCPs* $\mathcal{P}(\boldsymbol{y}|\boldsymbol{x})$, i.e., the conditional distribution of a label (in one-hot form) $\boldsymbol{y}$ given an input $\boldsymbol{x}$. The target model, referred to as the *student model*, is a mapping $\phi_{\boldsymbol{\theta}} : \mathcal{X} \to \mathcal{Y}$, typically a NN, parameterized by $\boldsymbol{\theta} \in \mathbb{R}^d$, where $\mathcal{Y}$ is the $K$-category statistical manifold. The desired model is one that minimizes the *risk*, also referred to as the *generalization error*, namely,

$$\min_{\boldsymbol{\theta} \in \mathbb{R}^d} f_{\mathcal{P}}(\boldsymbol{\theta}) := \mathbb{E}_{(\boldsymbol{x},\boldsymbol{y}) \sim \mathcal{P}} \left[ \ell(\phi_{\boldsymbol{\theta}}(\boldsymbol{x}), \boldsymbol{y}) \right]. \tag{1}$$

Here, $\ell : \mathcal{Y} \times \mathcal{Y} \to \mathbb{R}$ is a loss function, typically the Cross-Entropy (CE), defined henceforth as $\ell_{\mathrm{CE}}$.

The learning procedure aims to approach (1) without knowing $\mathcal{P}$ by seeking Empirical Risk Minimization (ERM), formulated using a labeled dataset $\mathcal{D} = \{(\boldsymbol{x}_n, \boldsymbol{y}_n)\}_{n=1}^N$, i.e.,

$$\min_{\boldsymbol{\theta} \in \mathbb{R}^d} f_{\mathcal{D}}(\boldsymbol{\theta}) := \frac{1}{|\mathcal{D}|} \sum_{n=1}^{|\mathcal{D}|} \ell(\phi_{\boldsymbol{\theta}}(\boldsymbol{x}_n), \boldsymbol{y}_n). \tag{2}$$

Clearly, $\mathbb{E}[f_{\zeta}(\boldsymbol{\theta})] = f_{\mathcal{P}}(\boldsymbol{\theta})$ for a random sample (or an i.i.d. batch) $\zeta = (\boldsymbol{x}, \boldsymbol{y})$ distributed via $\zeta \sim \mathcal{P}$.

**SGD-based KD**  In KD, a teacher model $\Phi$ (often pretrained) is used along with the dataset $\mathcal{D}$. The teacher model usually has larger capacity and is a more complex model compared to the student. The softmax predictions of the teacher, $\Phi(\boldsymbol{x}_n) \in \mathcal{Y}$, are used as additional soft labels in training the student. The student model is then trained with distillation parameter $\lambda \in [0, 1]$, modifying (2) into

$$\min_{\boldsymbol{\theta} \in \mathbb{R}^d} f_{\mathcal{D}}^{\Phi}(\boldsymbol{\theta}) := \frac{1}{|\mathcal{D}|} \sum_{n=1}^{|\mathcal{D}|} \left[ (1-\lambda)\ell(\phi_{\boldsymbol{\theta}}(\boldsymbol{x}_n), \boldsymbol{y}_n) + \lambda\ell(\phi_{\boldsymbol{\theta}}(\boldsymbol{x}_n), \Phi(\boldsymbol{x}_n)) \right]. \tag{3}$$

When $\ell$ is linear in the second argument (e.g., CE or KL-divergence loss), the objective (3) simplifies into

$$\min_{\boldsymbol{\theta} \in \mathbb{R}^d} f_{\mathcal{D}}^{\Phi}(\boldsymbol{\theta}) = \frac{1}{|\mathcal{D}|} \sum_{n=1}^{|\mathcal{D}|} \ell(\phi_{\boldsymbol{\theta}}(\boldsymbol{x}_n), (1-\lambda)\boldsymbol{y}_n + \lambda\Phi(\boldsymbol{x}_n)). \tag{4}$$

The standard SGD framework can then be readily applied by iterating over

$$\boldsymbol{\theta}^{t+1} = \boldsymbol{\theta}^t - \alpha \nabla f_{\xi_t}^{\Phi}(\boldsymbol{\theta}^t), \tag{5}$$

where $\alpha > 0$ is the learning rate, and $\xi_t \subset \mathcal{D}$ is a randomly sampled data point or mini-batch from $\mathcal{D}$.

**Assumptions & Definitions** To analyze the convergence profile of the student model in SGD-based probabilistic supervision, we define the desired student parameters as $\boldsymbol{\theta}^* \in \arg\min_{\boldsymbol{\theta} \in \mathbb{R}^d} f_{\mathcal{P}}(\boldsymbol{\theta})$ and their corresponding risk $f_{\mathcal{P}}^* := f_{\mathcal{P}}(\boldsymbol{\theta}^*)$. We make use of the following standard assumptions, commonly adopted in the analysis of SGD.

> *AS1* (Strong quasi-convexity). The risk function $f_{\mathcal{P}} : \mathbb{R}^d \to \mathbb{R}$ is differentiable and $\mu$-strongly quasi-convex for some constant $\mu > 0$, i.e., for any $\boldsymbol{\theta} \in \mathbb{R}^d$ it holds that
> $$f_{\mathcal{P}}^* \geq f_{\mathcal{P}}(\boldsymbol{\theta}) + \langle \nabla f_{\mathcal{P}}(\boldsymbol{\theta}), \boldsymbol{\theta}^* - \boldsymbol{\theta} \rangle + \frac{\mu}{2}\|\boldsymbol{\theta}^* - \boldsymbol{\theta}\|^2.$$

A weaker assumption is the Polyak-Łojasiewicz (PL) condition.

> *AS2* (Polyak-Łojasiewicz condition). The risk function $f_{\mathcal{P}} : \mathbb{R}^d \to \mathbb{R}$ is differentiable and satisfies the PL condition with parameter $\mu > 0$, i.e., for any $\boldsymbol{\theta} \in \mathbb{R}^d$ it holds that
> $$\|\nabla f_{\mathcal{P}}(\boldsymbol{\theta})\|^2 \geq 2\mu(f_{\mathcal{P}}(\boldsymbol{\theta}) - f_{\mathcal{P}}^*).$$

Since strong quasi-convexity implies the PL condition, we use either *AS1* or *AS2* in our analysis, in addition to the following assumptions:

> *AS3* (Expected smoothness). The loss $f_{\xi}(\boldsymbol{\theta})$ is differentiable and $\mathcal{L}$-smooth in expectation (for some constant $\mathcal{L}$) for $\xi \sim \mathcal{P}$, i.e., for any $\boldsymbol{\theta} \in \mathbb{R}^d$ it holds that
> $$\mathbb{E}_{\xi \sim \mathcal{P}}\left[\|\nabla f_{\xi}(\boldsymbol{\theta}) - \nabla f_{\xi}(\boldsymbol{\theta}^*)\|^2\right] \leq 2\mathcal{L}(f_{\mathcal{P}}(\boldsymbol{\theta}) - f_{\mathcal{P}}^*).$$
>
> Moreover, the risk $f_{\mathcal{P}}$ is $L$-smooth, namely $\|\nabla f_{\mathcal{P}}(\boldsymbol{\theta}_1) - \nabla f_{\mathcal{P}}(\boldsymbol{\theta}_2)\| \leq L\|\boldsymbol{\theta}_1 - \boldsymbol{\theta}_2\|$ for any $\boldsymbol{\theta}_1, \boldsymbol{\theta}_2 \in \mathbb{R}^d$ Garrigos & Gower (2023).

*AS3* often holds for the CE loss with standard NN architectures. This stems from the fact that the training procedure or activations of NNs often lead to its Jacobians being locally bounded, which, combined with the fact that the CE loss is smooth in the logits, ensures that the gradients vary smoothly in expectation Hardt et al. (2016).

> *AS4* (Student expressiveness) The student model $\phi_{\boldsymbol{\theta}}$ is sufficiently expressive such that it can realize the true BCP, i.e., there exists some $\boldsymbol{\theta} \in \mathbb{R}^d$ such that $\phi_{\boldsymbol{\theta}}(\boldsymbol{x}) \equiv [\mathcal{P}(\boldsymbol{y}_1|\boldsymbol{x}), \ldots, \mathcal{P}(\boldsymbol{y}_K|\boldsymbol{x})]$ for all $\boldsymbol{x} \in \text{supp}(\mathcal{P})$, where $\boldsymbol{y}_k$ is the one-hot encoding of the $k$th label.

The above assumptions are used to maintain a tractable analysis of SGD-based probabilistic supervision. Moreover, the design guidelines that arise from our analysis are empirically shown in Section 4 to enhance SGD-based KD even in settings where *AS1-AS4* do not necessarily hold.

Our analysis also uses the notions of *interpolation* and *gradient noise*, defined as follows:

**Definition 1** (Interpolation). *The optimization problem (1) satisfies the* interpolation *property if there exists some $\boldsymbol{\theta}_{\text{int}}^* \in \mathbb{R}^d$ such that for all $(\boldsymbol{x}, \boldsymbol{y}) \in \text{supp}(\mathcal{P})$, the model perfectly fits the data, i.e.,*

$$\ell(\phi_{\boldsymbol{\theta}_{\text{int}}^*}(\boldsymbol{x}), \boldsymbol{y}) = \min_{\boldsymbol{\theta} \in \mathbb{R}^d} \ell(\phi_{\boldsymbol{\theta}}(\boldsymbol{x}), \boldsymbol{y}), \qquad \forall (\boldsymbol{x}, \boldsymbol{y}) \in \text{supp}(\mathcal{P}).$$

By Definition 1, interpolation holds if there exists $\boldsymbol{\theta}_{\text{int}}^*$ which minimizes the loss for every input-label pair in the support of the data generating distribution $\mathcal{P}$. When this condition holds, we clearly have that $\boldsymbol{\theta}_{\text{int}}^* \in \arg\min_{\boldsymbol{\theta} \in \mathbb{R}^d} f_{\mathcal{P}}(\boldsymbol{\theta})$, meaning that the interpolating model is also a global minimizer of the generalization error. This formulation is valid in expectation, and naturally extends the classical finite-sum interpolation definition to continuous distributions Garrigos & Gower (2023).

**Definition 2** (Gradient noise). *We define the gradient noise of the optimization problem (1) as*

$$\sigma_f^* = \inf_{\boldsymbol{\theta}^* \in \arg\min f_{\mathcal{P}}} \mathbb{E}_{(\boldsymbol{x}, \boldsymbol{y}) \sim \mathcal{P}}\left[\|\nabla_{\boldsymbol{\theta}} \ell(\phi_{\boldsymbol{\theta}^*}(\boldsymbol{x}), \boldsymbol{y})\|^2\right].$$

The gradient noise controls the variance of the gradients at the optimum and can be seen as a measure of how far one is from interpolation.

### 3.2 Supervision with Perfect BCPs

We proceed to examine the effects of distilling from a perfect teacher (from the Bayesian KD perspective) on the training dynamics of the student. In this case, the student is supervised with the *true BCPs*. The resulting setup can be viewed as distilling from a teacher whose mapping is given by $\Phi(\boldsymbol{x}) \equiv [\mathcal{P}(\boldsymbol{y}_1|\boldsymbol{x}), \ldots, \mathcal{P}(\boldsymbol{y}_K|\boldsymbol{x})]$. We henceforth focus on learning with the CE loss, in setting where the BCP is bounded away from zero (to avoid instability of the CE with vanishing probability mass functions) Haussler (1992). In our analysis, we first show that the formulation of the generalization error using the BCPs yields the same optimal model as the one using one-hot labels in (1), and then we analyze the convergence of SGD-based learning of the empirical version of the BCP-based risk.

**BCP-Based Risk**   To formulate the BCP-based risk function, we replace the one-hot labels $\boldsymbol{y}$ in the stochastic optimization problem (1) with the true BCPs. The resulting objective is given by

$$\min_{\hat{\boldsymbol{\theta}} \in \mathbb{R}^d} \hat{f}_{\mathcal{P}}(\hat{\boldsymbol{\theta}}) := \mathbb{E}_{(\boldsymbol{x},\boldsymbol{y}) \sim \mathcal{P}} \left[ \ell(\phi_{\hat{\boldsymbol{\theta}}}(\boldsymbol{x}), \mathcal{P}(\boldsymbol{y}|\boldsymbol{x})) \right]. \tag{6}$$

The loss landscape of $\hat{f}_{\mathcal{P}}$ over $\mathbb{R}^d$ may differ from that of $f_{\mathcal{P}}$. The convergence of SGD-based learning is analyzed w.r.t. the optimal value and weights of the considered objective function. Therefore, before studying convergence, we compare $\hat{f}_{\mathcal{P}}^*$ and its minimizer $\hat{\boldsymbol{\theta}}^*$ to $f_{\mathcal{P}}^*$ and its minimizer $\boldsymbol{\theta}^*$, respectively.

**Proposition 1.** *When* AS4 *holds, the inference rule which minimizes both (1) and (6) is the true BCP, i.e.,* $\phi(\boldsymbol{x}) \equiv [\mathcal{P}(\boldsymbol{y}_1|\boldsymbol{x}), \ldots, \mathcal{P}(\boldsymbol{y}_K|\boldsymbol{x})]$, *and the minimal loss is the conditional entropy of* $\boldsymbol{y}$ *given* $\boldsymbol{x}$.

Proposition 1, also claimed in a different form in Menon et al. (2021), proves that both (1) and (6) share the same minimum and the same minimizer. The key benefit of learning with true BCPs instead of one-hot labels comes from Proposition 2.

**Proposition 2.** *When* AS4 *holds, the stochastic optimization task (6) satisfies the interpolation property (Definition 1).*

Based on Definition 1, interpolation here means that the minimizer of (6) matches the true BCPs at each sample, as opposed to attaining zero error on the one-hot labels. In particular, matching the true BCPs implies that the CE loss is minimized at every individual sample, and therefore the gradient of each per-sample loss also vanishes, as stated in the following Lemma.

**Lemma 1.** *When* AS4 *holds, the minimizer* $\hat{\boldsymbol{\theta}}^*$ *of the stochastic optimization task (6) satisfies*

$$\nabla_{\hat{\boldsymbol{\theta}}} \ell(\phi_{\hat{\boldsymbol{\theta}}^*}(\boldsymbol{x}), \mathcal{P}(\boldsymbol{y}|\boldsymbol{x})) = 0 \qquad \textit{for all } \boldsymbol{x} \in supp(\mathcal{P}).$$

Proposition 2 indicates that the stochastic optimization task (6) has several benefits over (1). Particularly, when learning from data using an empirical risk measure, Proposition 2 implies that the model that minimizes the risk also minimizes the corresponding empirical risk, i.e., it achieves

$$\min_{\hat{\boldsymbol{\theta}} \in \mathbb{R}^d} \hat{f}_{\mathcal{D}}(\hat{\boldsymbol{\theta}}) := \frac{1}{|\mathcal{D}|} \sum_{n=1}^{|\mathcal{D}|} \ell(\phi_{\hat{\boldsymbol{\theta}}}(\boldsymbol{x}_n), \mathcal{P}(\boldsymbol{y}_n|\boldsymbol{x}_n)), \tag{7}$$

for any data set $\mathcal{D} \subset \text{supp}(\mathcal{P})$. This property does not hold when learning from one-hot labels, where the empirical risk is often minimized by an overfitted model which does not generalize.

**BCP-Based SGD Learning**   The observation in Proposition 2 motivates analyzing the convergence profile (in the sense of the risk function $\hat{f}_{\mathcal{P}}$) of SGD based on the empirical version of (7). We note that such learning coincides with SGD-based KD with a perfect teacher (in the BCP sense), as (3) reduces to (7) when $\Phi$ outputs the true BCPs and $\lambda = 1$. Specifically, using the Assumptions described in Subsection 3.1, we obtain convergence guarantees for SGD-based learning starting from an arbitrary student model $\hat{\boldsymbol{\theta}}^0$. The resulting bounds, formulated in the sense of both the model parameters difference, as well as the resulting risk function, are stated in the following theorems:

**Theorem 1.** *Let Assumptions* AS1, AS3, *and* AS4 *hold. For any step-size* $\alpha \leq \dfrac{1}{\mathcal{L}}$, *the parameters learned via SGD (5) with true BCPs instead of one-hot labels converge as*

$$\mathbb{E}[\|\hat{\boldsymbol{\theta}}^t - \boldsymbol{\theta}^*\|^2] \leq (1 - \alpha\mu)^t \|\hat{\boldsymbol{\theta}}^0 - \boldsymbol{\theta}^*\|^2. \tag{8}$$

**Theorem 2.** *Let Assumptions* AS2*,* AS3*, and* AS4 *hold. For any step-size* $\alpha \leq \frac{\mu}{L\mathcal{L}}$*, the risk achieved by the model learned via SGD (5) with true BCPs instead of one-hot labels converges as*

$$\mathbb{E}[\hat{f}_{\mathcal{P}}(\hat{\boldsymbol{\theta}}^t) - f_{\mathcal{P}}^*] \leq (1 - \alpha\mu)^t (\hat{f}_{\mathcal{P}}(\hat{\boldsymbol{\theta}}^0) - f_{\mathcal{P}}^*). \tag{9}$$

Theorems 1-2 showcase the usefulness of SGD-based learning from true BCPs by comparing them with the corresponding bounds in the general gradient-based stochastic optimization literature Garrigos & Gower (2023). The standard SGD convergence bounds, which hold for similar constraints on the learning rate, are made up of the *convergence speed* term, dictating the rate of convergence, and the *neighborhood* term, corresponding to learning stability. Contrasting these with Theorems 1-2 reveals two main differences:

1. *Model Variance*: Theorems 1-2 include only a convergence speed term, which decays with $(1-\alpha\mu)^t$ as in standard SGD Garrigos & Gower (2023). The neighborhood term, which in standard SGD is proportional to $\frac{\alpha}{\mu}\sigma_f^*$ (Garrigos & Gower, 2023, Thm. 5.8), vanishes due to Lemma 1, indicating stable learning.

2. *Supported Learning Rates:* The range of learning rates $\alpha$ for which convergence can be guaranteed is twice as large compared to (Garrigos & Gower, 2023, Sec. 5), allowing the usage of larger step-sizes which in turn lead to faster convergence.

Overall, the results indicate that distilling from a well-trained teacher that sufficiently estimates the BCPs alters the optimization task from seeking to fit the one-hot labels to an interpolation task.

### 3.3 SUPERVISION WITH NOISY BCPs

While distilling from perfect BCPs yields desirable convergence guarantees, KD is typically done with an imperfect teacher. Here, we examine the effects on the training dynamics of the student when distilling from an imperfect teacher, i.e., when the student is supervised with *noisy BCPs*. Such noise corresponds to modeling errors, limited training data, or flawed training of the teacher. We show that even with noisy BCPs, we are able to achieve variance reduction and improved generalization under certain conditions.

**Noisy BCP-Based Risk**    We model the noisy BCPs as the true BCPs with additive distortion $\boldsymbol{\epsilon}$. This can be viewed as having the teacher mapping be $\Phi(\boldsymbol{x}) \equiv [\mathcal{P}(\boldsymbol{y}_1|\boldsymbol{x}) + \epsilon_1, \ldots, \mathcal{P}(\boldsymbol{y}_K|\boldsymbol{x}) + \epsilon_K]$, where we model $\boldsymbol{\epsilon} \sim \mathcal{P}_{\boldsymbol{\epsilon}}$ as zero-mean noise with variance $\nu$ and uncorrelated entries. This representation facilitates incorporating the deviation of the teacher from the BCPs while maintaining analytical tractability. While this formulation focuses on modeling BCP deviation as additive noise, similar findings can also be obtained for alternative models for noisy probabilities which remain on the simplex, such as Dirichlet-distributed, as shown in Appendix D.

The noisy BCP-based risk function is formulated in a similar fashion to the BCP-based risk (6), but with the noisy BCPs instead of the perfect ones. The resulting objective is given by

$$\min_{\tilde{\boldsymbol{\theta}} \in \mathbb{R}^d} \tilde{f}_{\mathcal{P},\mathcal{P}_{\boldsymbol{\epsilon}}}(\tilde{\boldsymbol{\theta}}) := \mathbb{E}_{(\boldsymbol{x},\boldsymbol{y}) \sim \mathcal{P}, \boldsymbol{\epsilon} \sim \mathcal{P}_{\boldsymbol{\epsilon}}} \left[ \ell(\phi_{\tilde{\boldsymbol{\theta}}}(\boldsymbol{x}), \mathcal{P}(\boldsymbol{y}|\boldsymbol{x}) + \boldsymbol{\epsilon}) \right]. \tag{10}$$

**Noisy BCP-Based SGD Learning**    We now analyze the impact of noisy BCPs on the convergence of SGD-based learning. We again employ the assumptions described in Subsection 3.1, and start from an arbitrary student model $\tilde{\boldsymbol{\theta}}^0$. The resulting bounds, formulated in the sense of both the model parameters difference, as well as the resulting risk function, are stated in the following theorems:

**Theorem 3.** *Let Assumptions* AS1*,* AS3*, and* AS4 *hold. For any step-size* $\alpha \leq \frac{1}{2\mathcal{L}}$*, the parameters learned via SGD (5) with noisy BCPs instead of one-hot labels converge as*

$$\mathbb{E}[\|\tilde{\boldsymbol{\theta}}^t - \boldsymbol{\theta}^*\|^2] \leq (1 - \alpha\mu)^t \|\tilde{\boldsymbol{\theta}}^0 - \boldsymbol{\theta}^*\|^2 + \frac{2\alpha}{\mu}\sigma_{\tilde{f}}^*, \tag{11}$$

*where* $\sigma_{\tilde{f}}^*$ *is the gradient noise (Definition 2) for the noisy BCP risk of (10).*

**Theorem 4.** *Let Assumptions* AS2*,* AS3*, and* AS4 *hold. For any step-size* $\alpha \leq \frac{\mu}{2L\mathcal{L}}$*, the risk achieved by the model via SGD (5) with noisy BCPs instead of one-hot labels converges as*

$$\mathbb{E}[\tilde{f}_{\mathcal{P},\mathcal{P}_{\boldsymbol{\epsilon}}}(\tilde{\boldsymbol{\theta}}^t) - f_{\mathcal{P}}^*] \leq (1 - \alpha\mu)^t (\tilde{f}_{\mathcal{P},\mathcal{P}_{\boldsymbol{\epsilon}}}(\tilde{\boldsymbol{\theta}}^0) - f_{\mathcal{P}}^*) + \frac{L\alpha}{\mu}\sigma_{\tilde{f}}^*. \tag{12}$$

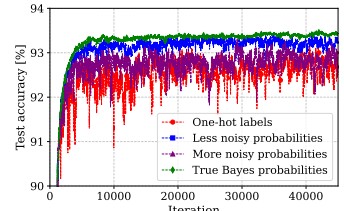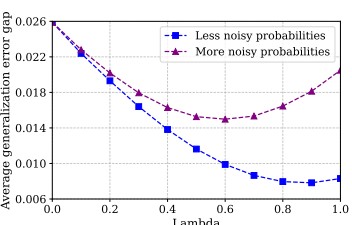

Figure 1: Learning curves, test accuracies, and average generalization error gaps in the learning curves when training with one-hot labels, when supervised with the true BCPs, when supervised with true BCPs corrupted with different noise levels, and supervision by combinations of one-hot labels and noisy BCPs adjusted based on several $\lambda$ values. Complete experimental details are provided in Appendix E.

Theorems 3-4 mirror the standard SGD convergence bounds Garrigos & Gower (2023), with one main difference: the gradient noise $\sigma_{\tilde{f}}^*$ reflects the variance induced by distilling from noisy BCPs. To understand this term, we compare it with the gradient noise in the standard case of learning with one-hot labels.

**Proposition 3.** *Under AS4 with $K$ classes, let $J_{\boldsymbol{\theta},k}[\phi_{\boldsymbol{\theta}}(\boldsymbol{x})]$ denote the $k$-th column of the Jacobian of the student model, i.e., $\nabla_{\boldsymbol{\theta}}\phi(\boldsymbol{x}) = [J_{\boldsymbol{\theta},1}[\phi_{\boldsymbol{\theta}}(\boldsymbol{x})], \ldots, J_{\boldsymbol{\theta},K}[\phi_{\boldsymbol{\theta}}(\boldsymbol{x})]]$. Then, for the CE loss, the gradient noise of the one-hot risk (1) is*

$$\sigma_f^* = \mathbb{E}_{\boldsymbol{x}\sim\mathcal{P}}\Big[ \sum_{k=1}^{K} \frac{1}{\mathcal{P}(\boldsymbol{y}_k|\boldsymbol{x})} \cdot \|J_{\boldsymbol{\theta},k}[\phi_{\boldsymbol{\theta}^*}(\boldsymbol{x})]\|^2\Big], \tag{13}$$

*and the gradient noise of the noisy BCP risk (10) is*

$$\sigma_{\tilde{f}}^* = \nu \cdot \mathbb{E}_{\boldsymbol{x}\sim\mathcal{P}}\Big[ \sum_{k=1}^{K} \frac{1}{[\mathcal{P}(\boldsymbol{y}_k|\boldsymbol{x})]^2} \cdot \|J_{\tilde{\boldsymbol{\theta}},k}[\phi_{\tilde{\boldsymbol{\theta}}^*}(\boldsymbol{x})]\|^2\Big]. \tag{14}$$

The expressions above demonstrate how gradient noise is affected by the form of supervision. In both cases, the resulting gradient noise is a weighted average of the $K$ columns of the Jacobian matrix evaluated at the optimum. In the standard case, the weights are the inverse of the $K$ entries of the true BCPs $\mathcal{P}(\boldsymbol{y}|\boldsymbol{x})$. When distilling from noisy BCPs, the weights are the noise variance $\nu$ multiplied by the inverse of the squared entries of the true BCPs $\mathcal{P}(\boldsymbol{y}|\boldsymbol{x})$. Since $\nu$ is a measure of how well the teacher estimates the true BCPs, as $\nu$ decreases, so does the gradient noise, and convergence becomes tighter. Note that when $\nu \to 0$, i.e., the teacher outputs the true BCPs, then (11) reduces to (8) and (12) reduces to (9).

An alternative point of view stems from Definition 1. Since the gradient noise is a measure of distance from interpolation, the gradient noise in the standard case can be seen as a measure of how far the one-hot labels are from the true BCPs, i.e., how noisy the data generating distribution is. Similarly, the gradient noise in (14) can be seen as a measure of how far the noisy BCPs are from the true BCPs.

Proposition 3 indicates that supervision with noisy BCPs is beneficial (under the above modeling assumptions) when $\sigma_{\tilde{f}}^* < \sigma_f^*$. That is, when the noise level is smaller than the inherent variance from one-hot labels. The point at which this balance flips depends on the data generating distribution (through $\mathcal{P}$), model smoothness (through the Jacobian), and teacher quality (through $\nu$).

**Numerical Study** The effects of the vanishing neighborhood term when supervising with true BCPs, as well as the neighborhood term when supervising with noisy BCPs on student performance, are demonstrated and explored in a synthetic experiment. We compare student models trained using SGD based on $(i)$ one-hot labels; $(ii)$ true BCPs; and $(iii)$ BCPs corrupted with two noise levels (denoted *more noisy* and *less noisy*). The latter are also compared with different combinations of one-hot labels and noisy BCPs adjusted based on $\lambda$ values. Noisy probabilities were generated using the Dirichlet distribution (Appendix D). The complete details of the study, alongside additional experiments, correlation analysis, and numerical validation of Proposition 3, are provided in Appendix E.

The learning curves in Figure 1 demonstrate that all students converge to the minimal loss at the same speed (for a given learning rate). Nevertheless, the loss of the student trained with true BCPs converges smoothly and exactly to the minimal loss, while students trained with one-hot labels or noisy BCPs

converge only up to a neighborhood term, which depends on the noise level. The smaller neighborhood term demonstrated in Theorem 4 and Proposition 3 is reflected in the experiments as both a lower error floor and reduced variance in the generalization error curves. The middle plot of Figure 1 shows that accuracy behaves in the same way: less noisy BCPs yield higher and more stable accuracy, while noisier BCPs result in larger fluctuations. This indicates that the variance reduction predicted by our analysis not only improves convergence in terms of loss but also translates directly into improved student performance. It is also demonstrated in the right plot in Figure 1 that the value of $\lambda$ that yields the best student performance depends on the level of noise in the BCP, i.e., on how calibrated the teacher is. This motivates future work to of finding optimal $\lambda$ values, possibly based on uncertainty measures provided by the teacher. In summary, supervision with noisy BCPs reshapes SGD convergence by changing the gradient noise, in a manner that depends on the data, the model structure, and the teacher calibration. Yet, KD with an imperfect teacher (providing noisy BCPs) can still outperform learning from one-hot labels, offering improved convergence and performance in SGD-based learning.

## 4 GUIDELINES FOR DESIGNING TEACHERS

### 4.1 METHOD

**Rationale**  The characterization of supervision with BCP estimates on SGD-based learning gives rise to concrete guidelines for designing teacher models in KD. Specifically, it implies that having a more calibrated teacher (in the sense of producing accurate BCP estimates), improves the convergence of the student model in SGD-based learning, aligning with the observations made by Menon et al. (2021) and Dao et al. (2021), which analyzed the KD-risk with the true BCPs. This motivates using teacher models based on Bayesian deep learning, as Bayesian Neural Networks (BNNs) are known to be better calibrated compared to standard (frequentist) NNs Jospin et al. (2022); Kristiadi et al. (2020); Gawlikowski et al. (2023).

**Bayesian Deep Learning**  BNNs extend traditional NNs by modeling the network parameters as random variables. Unlike traditional NNs which view learning from data $\mathcal{D}$ as seeking a single deterministic parameter vector, BNNs learn a posterior distribution over the parameter space. After the posterior distribution over the weights is estimated, inference is carried out by marginalizing the likelihood, typically by aggregating multiple realizations of the stochastic forward pass. Existing approaches to approximate such Bayesian inference by inferring the posterior distribution employ Markov Chain Monte Carlo sampling methods Neal (1993); Variational Inference (VI) approaches which approximate the posterior distribution by optimizing over a family of tractable distributions, e.g., Bernoulli (as in Monte Carlo Dropout) or Gaussian Fortuin (2022); Graves (2011); and Laplace Approximations (LAs), which use a quadratic (second-order Taylor) expansion of the log-posterior, followed by deriving a normal distribution over the NN weights Ritter et al. (2018). A key advantage of the LA approach is that it can be applied to pre-trained deterministic NNs Gawlikowski et al. (2023).

**Bayesian Teachers**  Our method employs KD with a BNN teacher. We consider both BNNs trained from scratch using VI, and transforming pre-trained NNs into BNNs via the LA. As such, our approach can be utilized both in cases where it is desirable to train a teacher model, and in cases where one wishes to leverage a pre-trained NN. Stochasticity can be introduced across all network weights, in specific layers, or only in the final layer. This flexibility enables one to adapt the complexity of the teacher model to the computational budget. The trained Bayesian teacher is incorporated into standard KD practice using response-based distillation. Specifically, the teacher's soft labels are obtained via the Monte Carlo approximation by averaging over $S$ stochastic forward passes (averaging is done after softmax is applied).

### 4.2 NUMERICAL STUDY

We evaluate our approach on the CIFAR-100 dataset, which contains 60,000 color images of size $32 \times 32$ across 100 classes, split into training and test sets with a 5:1 ratio. We evaluate 12 teacher–student pairs: 6 with matching architectures and 6 with different ones. For each pair, we compare six teacher types: *deterministic*, *Bayesian* (BNN trained with variational inference), *Laplace* (BNN obtained by applying the LA to the pre-trained deterministic teacher), *MCMI* Ye et al. (2024) (the deterministic teacher fine-tuned with a conditional mutual information loss to capture contextual structure), *TTDA* (the pre-trained deterministic teacher adapted to generate stochastic predictions using

Table 1: Test accuracy (%) of student networks on the CIFAR-100 dataset, averaged over 5 runs. Both the teachers and the students were trained with varying random seeds between runs. Results are reported for teacher-student pairs with both matching and different architectures. The subscript denotes changes relative to the corresponding deterministic teacher/student.

**Teachers and students with matching architectures**

| Teacher | ResNet-18 | | ResNet-50 | | ResNet-50 | | WRN-40-2 | | WRN-40-2 | | VGG-13 | |
|---|---|---|---|---|---|---|---|---|---|---|---|---|
| **Student** | ResNet-18 | | ResNet-18 | | ResNet-34 | | WRN-16-2 | | WRN-40-1 | | VGG-8 | |
| **Student accuracy (no KD)** | 73.23 | | 73.23 | | 73.61 | | 67.70 | | 65.03 | | 73.52 | |
| **Teacher Type** | Teacher | Student | Teacher | Student | Teacher | Student | Teacher | Student | Teacher | Student | Teacher | Student |
| Deterministic | 73.23 | 75.92 | 75.41 | 76.26 | 75.41 | 76.82 | 70.97 | 70.80 | 70.97 | 68.92 | 74.46 | 76.08 |
| Bayesian (Ours) | 74.61 $_{+1.38}$ | 76.92 $_{+1.00}$ | 75.67 $_{+0.26}$ | 77.27 $_{+1.01}$ | 75.67 $_{+0.26}$ | 77.63 $_{+0.81}$ | 74.27 $_{+3.30}$ | 72.94 $_{+2.14}$ | 74.27 $_{+3.30}$ | 70.77 $_{+1.85}$ | 74.72 $_{+0.27}$ | 77.61 $_{+1.53}$ |
| Laplace (Ours) | 72.72 $_{-0.50}$ | 76.30 $_{+0.38}$ | 74.83 $_{-0.58}$ | 76.69 $_{+0.43}$ | 74.83 $_{-0.58}$ | 76.23 $_{-0.59}$ | 70.45 $_{-0.51}$ | 71.80 $_{+0.99}$ | 70.45 $_{-0.51}$ | 70.04 $_{+1.12}$ | 73.95 $_{-0.50}$ | 76.09 $_{+0.01}$ |
| MCMI Ye et al. (2024) | 73.46 $_{+0.23}$ | 75.86 $_{-0.06}$ | 75.60 $_{+0.19}$ | 76.44 $_{+0.18}$ | 75.60 $_{+0.19}$ | 76.86 $_{+0.04}$ | 71.28 $_{+0.32}$ | 70.87 $_{+0.07}$ | 71.28 $_{+0.32}$ | 68.66 $_{-0.27}$ | 74.61 $_{+0.15}$ | 76.35 $_{+0.27}$ |
| TTDA | 73.23 | 76.21 $_{+0.29}$ | 75.41 | 76.30 $_{+0.04}$ | 75.41 | 76.83 $_{+0.01}$ | 70.97 | 70.75 $_{-0.05}$ | 70.97 | 69.05 $_{+0.13}$ | 74.46 | 76.61 $_{+0.53}$ |
| MSE Hamidi et al. (2024) | 74.56 $_{+1.34}$ | 75.01 $_{-0.90}$ | 71.76 $_{-3.65}$ | 74.89 $_{-1.37}$ | 71.76 $_{-3.65}$ | 75.58 $_{-1.24}$ | 69.99 $_{-0.97}$ | 68.62 $_{-2.19}$ | 69.99 $_{-0.97}$ | 65.42 $_{-3.50}$ | 73.31 $_{-1.15}$ | 73.86 $_{-2.22}$ |

**Teachers and students with different architectures**

| Teacher | ResNet-50 | | ResNet-50 | | VGG-13 | | VGG-13 | | ResNet-50 | | VGG-13 | |
|---|---|---|---|---|---|---|---|---|---|---|---|---|
| **Student** | WRN-40-2 | | VGG-8 | | ResNet-18 | | WRN-40-1 | | WRN-16-2 | | WRN-40-2 | |
| **Student accuracy (no KD)** | 70.76 | | 73.52 | | 73.23 | | 65.03 | | 67.70 | | 70.76 | |
| **Teacher Type** | Teacher | Student | Teacher | Student | Teacher | Student | Teacher | Student | Teacher | Student | Teacher | Student |
| Deterministic | 75.41 | 73.79 | 75.41 | 75.66 | 74.46 | 76.36 | 74.46 | 67.90 | 75.41 | 69.36 | 74.46 | 73.71 |
| Bayesian (Ours) | 75.67 $_{+0.26}$ | 75.82 $_{+2.03}$ | 75.67 $_{+0.26}$ | 77.27 $_{+1.62}$ | 74.72 $_{+0.27}$ | 77.41 $_{+1.04}$ | 74.72 $_{+0.27}$ | 71.37 $_{+3.47}$ | 75.67 $_{+0.26}$ | 73.63 $_{+4.27}$ | 74.72 $_{+0.27}$ | 75.45 $_{+1.75}$ |
| Laplace (Ours) | 74.83 $_{-0.58}$ | 74.57 $_{+0.77}$ | 74.83 $_{-0.58}$ | 76.11 $_{+0.46}$ | 73.95 $_{-0.50}$ | 76.42 $_{+0.06}$ | 73.95 $_{-0.50}$ | 70.87 $_{+2.97}$ | 74.83 $_{-0.58}$ | 72.52 $_{+3.16}$ | 73.95 $_{-0.50}$ | 74.39 $_{+0.69}$ |
| MCMI Ye et al. (2024) | 75.60 $_{+0.19}$ | 73.50 $_{-0.30}$ | 75.60 $_{+0.19}$ | 75.75 $_{+0.09}$ | 74.61 $_{+0.15}$ | 76.40 $_{+0.04}$ | 74.61 $_{+0.15}$ | 67.48 $_{-0.42}$ | 75.60 $_{+0.19}$ | 69.58 $_{+0.22}$ | 74.61 $_{+0.15}$ | 74.20 $_{+0.49}$ |
| TTDA | 75.41 | 73.76 $_{-0.03}$ | 75.41 | 76.00 $_{+0.34}$ | 74.46 | 76.80 $_{+0.44}$ | 74.46 | 67.53 $_{-0.37}$ | 75.41 | 69.24 $_{-0.12}$ | 74.46 | 74.05 $_{+0.34}$ |
| MSE Hamidi et al. (2024) | 71.76 $_{-3.65}$ | 71.09 $_{-2.70}$ | 71.76 $_{-3.65}$ | 74.09 $_{-1.56}$ | 73.31 $_{-1.15}$ | 74.87 $_{-1.49}$ | 73.31 $_{-1.15}$ | 65.61 $_{-2.29}$ | 71.76 $_{-3.65}$ | 68.55 $_{-0.81}$ | 73.31 $_{-1.15}$ | 71.39 $_{-2.32}$ |

test-time data augmentation Gawlikowski et al. (2023)), and *MSE* Hamidi et al. (2024) (trained with mean squared error loss instead of CE). Unless stated otherwise, we will refer to students trained from each teacher type as *deterministic students*, *Bayesian students*, *Laplace students*, etc., even though the student is always the same deterministic model for a given teacher–student pair.

Training follows standard response-based KD Hinton et al. (2015). For each teacher–student pair, we use 6 combinations of teacher temperature, student temperature, and $\lambda$ values, and repeat the experiment with each combination 5 times. We report the mean student accuracy achieved by the best configuration. Accuracy variances and results of all 6 combinations of temperature-$\lambda$ values are provided in Appendix K, while Appendix F contains full details of the training setup. The source code is available at `https://github.com/Bayesian-KD`. As shown in Table 1, Bayesian students consistently achieve the highest accuracies across the board. The gains over deterministic students reach up to 4.27% (ResNet-50→WRN-16-2), even though the corresponding Bayesian teachers themselves improve on deterministic teachers by up to 0.26% in almost all cases. Laplace students improve in all cases except one, with gains of up to 3.07% despite all Laplace teachers performing worse than their deterministic counterparts.

We also examine the stability of the students' performance during training. The average noise level during the last 50 epochs of the test accuracy curves is reported in Figure 2, measured by $(i)$ the gap between the highest and lowest accuracies reached and $(ii)$ the standard deviation around the mean. The results show that Bayesian students not only achieve higher accuracy than deterministic students, but also consistently converge with less variance, indicating highly stable learning. For example, in the ResNet-50→ResNet-18 case, Bayesian students show a 30% improvement in convergence noise compared to deterministic students.

These observations are aligned with our theoretical analysis in Subsection 3.3. There, we proved that as noise decreases in the supervisory noisy BCP estimates, the neighborhood term in the convergence

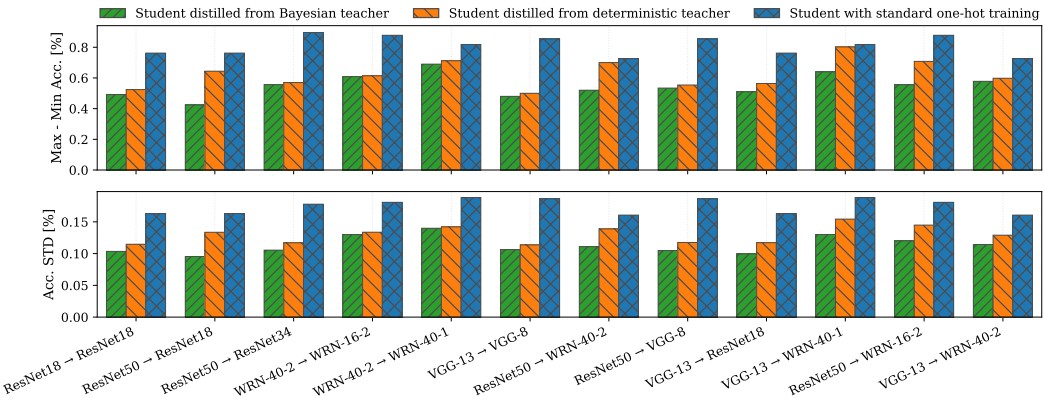

Figure 2: Test accuracy stability over the last 50 training epochs with standard one-hot training, deterministic teachers, and Bayesian teachers.

bounds for both the weights and the generalization error becomes smaller. In a synthetic experiment we confirmed this effect in the generalization error, and also observed that accuracy behaves analogously. The experiments reported here show the same: students distilled from Bayesian teachers, which provide more calibrated probability estimates (i.e., provide better approximations of the BCPs compared to deterministic teachers), converge with reduced variance and higher accuracy. The experiments were performed with the Adam optimizer in order to show that our theoretical results, which hold for standard SGD, generalize to related optimizers as well. This empirical generalization is in fact supported by recent theoretical studies Xia & Massei (2023) showing that Adam admits convergence guarantees under PL-condition and smoothness assumptions similar to those used in our analysis, with convergence up to a noise-dependent neighborhood term (analogous to SGD). These findings provide theoretical backing for the empirical applicability of our conclusions beyond SGD.

## 5 CONCLUSION

We characterized the learning dynamics of students trained from both exact and noisy approximations of the BCPs, corresponding to different levels of teacher calibration. Our analysis shows that teachers that better approximate the true BCPs yield variance reduction in the learning dynamics of the student, and numerically this leads to improved performance, both in terms of higher accuracy ceilings and reduced variance of the accuracy curve near convergence. Moreover, we advocate the use of Bayesian deep learning models as teachers, since they provide better-calibrated outputs. Aligned with our theoretical analysis, our experimental study shows that students trained from Bayesian teachers achieve higher accuracies and more stable accuracy curves compared to students trained from deterministic teachers.

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

APPENDIX

## A ADDITIONAL RELATED WORK

**Probabilistic supervision** The effects of supervision with probability estimates or soft labels have been investigated in several works. Fan et al. (2024a) examined the dark knowledge provided by teachers of different capacities, showing that larger teachers tend to produce probability vectors with lower distinction among non-ground-truth classes, and explored multiple ways to address capacity mismatch. Zhang et al. (2023) argued that a teacher minimizing the CE loss is sub-optimal, and proposed the PT-Loss objective, which transforms the original teacher into a proxy teacher whose distribution is closer to the ground truth. They further established a theoretical link between distributional closeness and student generalizability. Ren et al. (2022) studied the learning paths of models trained with noisy BCPs, one-hot labels, and BCPs. Yuan et al. (2023) analyzed whether biased soft labels can be effective, proposing conditions under which biased soft-label learning remains classifier-consistent and ERM-learnable. For binary classification, Jeong et al. (2024) studied probabilistic supervision and introduced theoretical properties of their estimator, including consistency, unbiasedness, convergence rate, and variance. While we also develop a theoretical analysis of probabilistic supervision, we focus on the dynamics of SGD-based convergence and their role in improving student performance. We introduce advantages of KD in stochastic optimization, whereas prior work focused on generalization gaps and estimator properties.

**Customized teachers and improved KD schemes** Additional works focused on modifying teacher predictions to enhance student performance. For example, Guo et al. (2024) recognized that uncertain classes in teacher outputs hinder learning and proposed increasing the weight on confident classes in teacher predictions, while Hossain et al. calibrated teacher logits based on the model's representational capacity, identifying calibration as a key driver of KD. Dong et al. (2022) established that an ERM minimizer can approximate the true label distribution under Lipschitz continuity and robustness of the feature extractor, and introduced a teacher training method with Lipschitz and consistency regularization. Others explored more novel directions: Yang et al. (2024) drew inspiration from human educational strategies, Wang et al. (2025b) facilitated knowledge sharing within the same class, Harutyunyan et al. (2023) aligned teacher predictions with the student's neural tangent kernel, and Yang et al. (2023) employed meta-learning to strengthen the student's ability to generate new knowledge. In contrast, our approach leverages Bayesian teachers, whose outputs are naturally better calibrated driven by the Bayesian paradigm, which was motivated by our theoretical analysis.

**BNNs and KD** In addition to the works discussed in Section 2, several additional studies considered BNNs and KD. The work of Chen et al. (2025) proposes a dynamic dropout (which can be viewed as a form of Bayesian modeling) self-distillation method for object segmentation, which solves this problem by discarding the knowledge that the student struggles to learn. The work of Fang et al. (2024) developed a method named Bayesian Knowledge Distillation (BKD) to provide a transparent interpretation of the working mechanism of KD. In BKD, the regularization imposed by the teacher model in KD is formulated as a teacher-informed prior for the student model's parameters. The work of Shen et al. (2021) performed KD with a complex BNN MCMC teacher into a BNN VI student. These works on BNNs and KD have fundamentally different goals from our work. Our approach towards utilizing BNNs is a direct consequence of our theoretical analysis, and is from the perspective of their supervisory signal containing better information in KD compared to deterministic models, and not from the perspective of reducing the cost of Bayesian inference and uncertainty quantification.

## B BASIC LINEAR ALGEBRA, FACTS, AND INEQUALITIES

We first review some standard inequalities and basic facts which are used in the proofs.

• The squared norm of the sum of two terms can simply be bounded by the sum of each squared norm individually

$$\|a + b\|^2 \leq 2\|a\|^2 + 2\|b\|^2. \tag{15}$$

• $f(\boldsymbol{\theta})$ is called $L-$smooth if its gradient is Lipschitz continuous with constant $L \geq 0$ (Definition 2.24 in Garrigos & Gower (2023)), which means that

$$\|\nabla f(\boldsymbol{\theta}_1) - \nabla f(\boldsymbol{\theta}_2)\| \leq L\|\boldsymbol{\theta}_1 - \boldsymbol{\theta}_2\|. \tag{16}$$

Additionally, if $f(\boldsymbol{\theta})$ is $L-$smooth then

$$f(\boldsymbol{\theta}_1) \leq f(\boldsymbol{\theta}_2) + \langle \nabla f(\boldsymbol{\theta}_2), \boldsymbol{\theta}_1 - \boldsymbol{\theta}_2 \rangle + \frac{L}{2}\|\boldsymbol{\theta}_1 - \boldsymbol{\theta}_2\|^2, \tag{17}$$

for all $\boldsymbol{\theta}_1, \boldsymbol{\theta}_2 \in \mathbb{R}^d$ (Lemma 2.25 in Garrigos & Gower (2023)).

• Similarly to other analyses that establish linear or exponential rates, in our proofs we rely on a standard technique: converting a one-step recurrence into a convergence inequality. Assume the sequence $\{z_t\}_{t\geq 0}$ satisfies

$$z_{t+1} \leq (1-\eta)z_t + N, \qquad t = 0, 1, 2, \dots$$

for some constants $\eta \in (0, 1]$ and $N \geq 0$. Unfolding the recursion gives

$$z_t \leq (1-\eta)^t z_0 + (1-\eta)^{t-1}N + (1-\eta)^{t-2}N + \cdots + (1-\eta)^0 N.$$

By recognizing the geometric sum, this yields

$$z_t \leq (1-\eta)^t z_0 + N\sum_{j=0}^{\infty}(1-\eta)^j = (1-\eta)^t z_0 + \frac{N}{\eta}. \tag{18}$$

• We make use of Gibbs' inequality, which states that for the discrete probability distributions $P = \{p_1, \dots, p_n\}$ and $Q = \{q_1, \dots, q_n\}$, it holds that

$$-\sum_{i=1}^{n} p_i \log p_i \leq -\sum_{i=1}^{n} p_i \log q_i,$$

with equality if and only if $p_i = q_i$ for $i = 1, \dots, n$. A direct consequence of Gibbs' inequality is that

$$\min_{\boldsymbol{q}, s.t. \boldsymbol{q} \text{ is a discrete probability distribution}} -\sum_{i=1}^{n} p_i \log q_i = -\sum_{i=1}^{n} p_i \log p_i. \tag{19}$$

• We make use of the law of total expectation, which states that for random variables $(X, Y)$ and any measurable $f$,
$$\mathbb{E}[f(X, Y)] = \mathbb{E}_X\left[\mathbb{E}_{Y|X}[f(X, Y) \mid X]\right]. \tag{20}$$

• It holds that
$$\frac{\partial \log(\phi(\boldsymbol{\theta}))}{\partial \boldsymbol{\theta}} = \frac{1}{\phi(\boldsymbol{\theta})} \cdot \frac{\partial \phi(\boldsymbol{\theta})}{\partial \boldsymbol{\theta}}, \tag{21}$$

for $\phi(\boldsymbol{\theta}) > 0$.

• For any $a \in \mathbb{R}^m$ and $M \in \mathbb{R}^{m \times m}$,
$$a^\top M a = \text{Tr}(M\, aa^\top), \tag{22}$$

where $\text{Tr}(\cdot)$ is the trace operator. Moreover, since both trace and expectation are linear operators, we can interchange them, i.e.,
$$\mathbb{E}[\text{Tr}(M)] = \text{Tr}(\mathbb{E}[M]). \tag{23}$$

• For the generalization error (1), the optimal inference rule in the sense of minimal cross-entropy risk is the true conditional distribution Shalev-Shwartz & Ben-David (2014), i.e.,
$$\phi(\boldsymbol{x}) = [\mathcal{P}(\boldsymbol{y}_1|\boldsymbol{x}), \dots, \mathcal{P}(\boldsymbol{y}_K|\boldsymbol{x})]. \tag{24}$$

## C    DETAILED PROOFS

### C.1    PROOF OF PROPOSITION 1

*Proof.* For ease of presentation, instead of using the $K$-categorical one-hot vector $\boldsymbol{y}$, we define the label variable $\boldsymbol{s} \in \{1, \dots, K\}$, and since $\phi_{\hat{\boldsymbol{\theta}}}(\boldsymbol{x})$ represents a $K$-dimensional probability mass function, we

write $\tilde{p}(\boldsymbol{s} = k|\boldsymbol{x}) = [\phi_{\hat{\boldsymbol{\theta}}}(\boldsymbol{x})]_k$. With these notations, the BCP-based risk (6) with the CE loss becomes

$$
\begin{aligned}
\hat{f}_{\mathcal{P}}(\hat{\boldsymbol{\theta}}) &= \mathbb{E}_{(\boldsymbol{x},\boldsymbol{y})\sim\mathcal{P}} \left[ \ell(\phi_{\hat{\boldsymbol{\theta}}}(\boldsymbol{x}), \mathcal{P}(\boldsymbol{y}|\boldsymbol{x})) \right] \\
&= \mathbb{E}_{(\boldsymbol{x},\boldsymbol{s})\sim\mathcal{P}} \left[ \ell(\phi_{\hat{\boldsymbol{\theta}}}(\boldsymbol{x}), \mathcal{P}(\boldsymbol{s}|\boldsymbol{x})) \right] \\
&= \mathbb{E}_{(\boldsymbol{x},\boldsymbol{s})\sim\mathcal{P}} \left[ - \log \tilde{p}(\boldsymbol{s}|\boldsymbol{x}) \right] \\
&= \mathbb{E}_{(\boldsymbol{x},\boldsymbol{s})\sim\mathcal{P}} \left[ - \log \frac{\tilde{p}(\boldsymbol{s}|\boldsymbol{x})}{\mathcal{P}(\boldsymbol{s}|\boldsymbol{x})} \right] + \mathbb{E}_{(\boldsymbol{x},\boldsymbol{s})\sim\mathcal{P}} \left[ - \log \mathcal{P}(\boldsymbol{s}|\boldsymbol{x}) \right] \\
&= D_{\mathrm{KL}} \left( \tilde{p}(\boldsymbol{s}|\boldsymbol{x}) \| \mathcal{P}(\boldsymbol{s}|\boldsymbol{x}) \right) + H(\boldsymbol{s}|\boldsymbol{x}),
\end{aligned}
$$

where $H(\boldsymbol{s}|\boldsymbol{x})$ is the conditional entropy of $\boldsymbol{s}$ given $\boldsymbol{x}$, while $D_{\mathrm{KL}}(\cdot\|\cdot)$ is the Kullback-Leibler divergence Shalev-Shwartz & Ben-David (2014). As $D_{\mathrm{KL}}$ is non-negative and equals zero only when $\tilde{p}(\boldsymbol{s}|\boldsymbol{x}) \equiv \mathcal{P}(\boldsymbol{s}|\boldsymbol{x})$, i.e., $[\phi_{\hat{\boldsymbol{\theta}}}(\boldsymbol{x})]_k = \mathcal{P}(\boldsymbol{y}_k|\boldsymbol{x})$, it holds that the BCP-based risk (6) is minimized when the inference output is the true conditional distribution. Under Assumption *AS4*, the minimizer is attainable by some $\hat{\boldsymbol{\theta}}^* \in \mathbb{R}^d$, and the proof is concluded. $\qquad\square$

## C.2 PROOF OF PROPOSITION 2

*Proof.* To prove that the interpolation property holds, we note that $\phi_{\hat{\boldsymbol{\theta}}}(\boldsymbol{x})$ represents a $K$-dimensional probability mass function. Accordingly, under the CE loss, it holds that

$$
\begin{aligned}
\min_{\hat{\boldsymbol{\theta}}\in\mathbb{R}^d} \ell(\phi_{\hat{\boldsymbol{\theta}}}(\boldsymbol{x}), \mathcal{P}(\boldsymbol{y}|\boldsymbol{x})) &= \min_{\hat{\boldsymbol{\theta}}\in\mathbb{R}^d} - \sum_{k=1}^{K} \mathcal{P}(\boldsymbol{y}_k|\boldsymbol{x}) \log [\phi_{\hat{\boldsymbol{\theta}}}(\boldsymbol{x})]_k \\
&\overset{(19)+(AS4)}{=} - \sum_{k=1}^{K} \mathcal{P}(\boldsymbol{y}_k|\boldsymbol{x}) \log \mathcal{P}(\boldsymbol{y}_k|\boldsymbol{x}) \\
&= \ell(\mathcal{P}(\boldsymbol{y}|\boldsymbol{x}), \mathcal{P}(\boldsymbol{y}|\boldsymbol{x})) \\
&\overset{(Prop\ 1)}{=} \ell(\phi_{\hat{\boldsymbol{\theta}}^*}(\boldsymbol{x}), \mathcal{P}(\boldsymbol{y}|\boldsymbol{x})).
\end{aligned}
$$

Therefore, $\phi_{\hat{\boldsymbol{\theta}}^*}(\boldsymbol{x}) = [\mathcal{P}(\boldsymbol{y}_1|\boldsymbol{x}), \dots, \mathcal{P}(\boldsymbol{y}_K|\boldsymbol{x})]$ brings $\ell(\phi_{\hat{\boldsymbol{\theta}}}(\boldsymbol{x}), \mathcal{P}(\boldsymbol{y}|\boldsymbol{x}))$ to a minimum for all $\boldsymbol{x} \in \mathrm{supp}(\mathcal{P})$ and interpolation holds at $\hat{\boldsymbol{\theta}}^*$, proving the proposition. $\qquad\square$

## C.3 PROOF OF PROPOSITION 3

*Proof.* In order to express the gradient noise of both the one-hot risk (1) and the noisy BCP risk (10), we recall the resulting expressions for the gradient noise based on Definition 2, i.e.,

$$
\sigma_f^* = \inf_{\boldsymbol{\theta}^*\in\arg\min f_{\mathcal{P}}} \mathbb{E}_{(\boldsymbol{x},\boldsymbol{y})\sim\mathcal{P}} \left[ \| \nabla_{\boldsymbol{\theta}} \ell(\phi_{\boldsymbol{\theta}^*}(\boldsymbol{x}), \boldsymbol{y}) \|^2 \right],
$$

and

$$
\sigma_{\tilde{f}}^* = \inf_{\tilde{\boldsymbol{\theta}}^*\in\arg\min \tilde{f}_{\mathcal{P},\mathcal{P}_\epsilon}} \mathbb{E}_{(\boldsymbol{x},\boldsymbol{y})\sim\mathcal{P},\boldsymbol{\epsilon}\sim\mathcal{P}_\epsilon} \left[ \| \nabla_{\tilde{\boldsymbol{\theta}}} \ell(\phi_{\tilde{\boldsymbol{\theta}}^*}(\boldsymbol{x}), \mathcal{P}(\boldsymbol{y}|\boldsymbol{x}) + \boldsymbol{\epsilon}) \|^2 \right].
$$

Next, we give an expression for $\nabla_{\boldsymbol{\theta}} \ell(\phi_{\boldsymbol{\theta}^*}(\boldsymbol{x}), \tilde{\boldsymbol{y}})$ for any $K$-dimensional probability mass function $\tilde{\boldsymbol{y}}$. In this case, we have that $\nabla_{\boldsymbol{\theta}} \ell(\phi_{\boldsymbol{\theta}^*}(\boldsymbol{x}), \tilde{\boldsymbol{y}}) = \nabla_{\boldsymbol{\theta}} [-\tilde{\boldsymbol{y}}^T \log \phi_{\boldsymbol{\theta}^*}(\boldsymbol{x})]$, whose $i$th element is

$$
\begin{aligned}
\frac{\partial \ell(\phi_{\boldsymbol{\theta}^*}(\boldsymbol{x}), \tilde{\boldsymbol{y}})}{\partial \boldsymbol{\theta}_i} &= - \sum_{k=1}^{K} [\tilde{\boldsymbol{y}}]_k \cdot \frac{\partial [\log \phi_{\boldsymbol{\theta}^*}(\boldsymbol{x})]_k}{\partial \boldsymbol{\theta}_i} \\
&\overset{(21)}{=} - \sum_{k=1}^{K} [\tilde{\boldsymbol{y}}]_k \cdot \frac{1}{[\phi_{\boldsymbol{\theta}^*}(\boldsymbol{x})]_k} \cdot \frac{\partial [\phi_{\boldsymbol{\theta}^*}(\boldsymbol{x})]_k}{\partial \boldsymbol{\theta}_i} \\
&\overset{(24)}{=} - \sum_{k=1}^{K} [\tilde{\boldsymbol{y}}]_k \cdot \frac{1}{\mathcal{P}(\boldsymbol{y}_k|\boldsymbol{x})} \cdot \frac{\partial [\phi_{\boldsymbol{\theta}^*}(\boldsymbol{x})]_k}{\partial \boldsymbol{\theta}_i}.
\end{aligned}
$$

Accordingly, $\nabla_{\boldsymbol{\theta}} \ell(\phi_{\boldsymbol{\theta}^*}(\boldsymbol{x}), \tilde{\boldsymbol{y}})$ can be written as

$$
\nabla_{\boldsymbol{\theta}} \ell(\phi_{\boldsymbol{\theta}^*}(\boldsymbol{x}), \tilde{\boldsymbol{y}}) = J_{\boldsymbol{\theta}} [\phi_{\boldsymbol{\theta}^*}(\boldsymbol{x})] \cdot \mathrm{diag}\left( \frac{1}{\mathcal{P}(\boldsymbol{y}|\boldsymbol{x})} \right) \cdot \tilde{\boldsymbol{y}}.
$$

Finally, we are able to plug in the expression for $\nabla_{\boldsymbol{\theta}}\ell(\phi_{\boldsymbol{\theta}^*}(\boldsymbol{x}),\boldsymbol{y})$ into the definition of the gradient noise.

$$
\begin{aligned}
\sigma_f^* &= \inf_{\boldsymbol{\theta}^* \in \arg\min f_{\mathcal{P}}} \mathbb{E}_{(\boldsymbol{x},\boldsymbol{y})\sim\mathcal{P}}\left[\|\nabla_{\boldsymbol{\theta}}\ell(\phi_{\boldsymbol{\theta}^*}(\boldsymbol{x}),\boldsymbol{y})\|^2\right] \\
&= \mathbb{E}_{(\boldsymbol{x},\boldsymbol{y})\sim\mathcal{P}}\left[\|J_{\boldsymbol{\theta}}[\phi_{\boldsymbol{\theta}^*}(\boldsymbol{x})]\cdot\mathrm{diag}(\frac{1}{\mathcal{P}(\boldsymbol{y}|\boldsymbol{x})})\cdot[\boldsymbol{y}]_k\|^2\right] \\
&\overset{(20)}{=} \mathbb{E}_{\boldsymbol{x}\sim\mathcal{P}}[\mathbb{E}_{\boldsymbol{y}|\boldsymbol{x}}[\|J_{\boldsymbol{\theta}}[\phi_{\boldsymbol{\theta}^*}(\boldsymbol{x})]\cdot\mathrm{diag}(\frac{1}{\mathcal{P}(\boldsymbol{y}|\boldsymbol{x})})\cdot[\boldsymbol{y}]_k\|^2|\boldsymbol{x}]] \\
&= \mathbb{E}_{\boldsymbol{x}\sim\mathcal{P}}[\sum_{k=1}^{K}\mathcal{P}(\boldsymbol{y}_k|\boldsymbol{x})\cdot\|\frac{1}{\mathcal{P}(\boldsymbol{y}_k|\boldsymbol{x})}\cdot J_{\boldsymbol{\theta},k}[\phi_{\boldsymbol{\theta}^*}(\boldsymbol{x})]\|^2] \\
&= \mathbb{E}_{\boldsymbol{x}\sim\mathcal{P}}[\sum_{k=1}^{K}\frac{1}{\mathcal{P}(\boldsymbol{y}_k|\boldsymbol{x})}\cdot\|J_{\boldsymbol{\theta},k}[\phi_{\boldsymbol{\theta}^*}(\boldsymbol{x})]\|^2]
\end{aligned}
$$

where $J_{\boldsymbol{\theta},k}[\phi_{\boldsymbol{\theta}}(\boldsymbol{x})]$ is the $k_{th}$ column of the Jacobian of the student model.

Next, we give an expression for $\nabla_{\tilde{\boldsymbol{\theta}}}\ell(\phi_{\tilde{\boldsymbol{\theta}}^*}(\boldsymbol{x}),\mathcal{P}(\boldsymbol{y}|\boldsymbol{x})+\boldsymbol{\epsilon})$. In this case, $\mathcal{P}(\boldsymbol{y}|\boldsymbol{x})$ is the true BCP which is a $K$-sized vector, and $\boldsymbol{\epsilon}$ is the zero-mean noise with variance $\nu$ and uncorrelated entries, also a $K$-sized vector. Therefore,

$$
\begin{aligned}
\nabla_{\tilde{\boldsymbol{\theta}}}\ell(\phi_{\tilde{\boldsymbol{\theta}}^*}(\boldsymbol{x}),\mathcal{P}(\boldsymbol{y}|\boldsymbol{x})+\boldsymbol{\epsilon}) &= \nabla_{\tilde{\boldsymbol{\theta}}}\ell(\phi_{\tilde{\boldsymbol{\theta}}^*}(\boldsymbol{x}),\mathcal{P}(\boldsymbol{y}|\boldsymbol{x}))+\nabla_{\tilde{\boldsymbol{\theta}}}\ell(\phi_{\tilde{\boldsymbol{\theta}}^*}(\boldsymbol{x}),\boldsymbol{\epsilon}) \\
&= \nabla_{\tilde{\boldsymbol{\theta}}}\ell(\phi_{\hat{\boldsymbol{\theta}}^*}(\boldsymbol{x}),\mathcal{P}(\boldsymbol{y}|\boldsymbol{x}))+\nabla_{\tilde{\boldsymbol{\theta}}}\ell(\phi_{\tilde{\boldsymbol{\theta}}^*}(\boldsymbol{x}),\boldsymbol{\epsilon}) \\
&\overset{\text{(Lemma 1)}+(AS4)+(\text{Prop 2})}{=} \nabla_{\tilde{\boldsymbol{\theta}}}\ell(\phi_{\tilde{\boldsymbol{\theta}}^*}(\boldsymbol{x}),\boldsymbol{\epsilon}) \\
&= \nabla_{\tilde{\boldsymbol{\theta}}}[-\boldsymbol{\epsilon}^T\log\phi_{\tilde{\boldsymbol{\theta}}^*}(\boldsymbol{x})]
\end{aligned}
$$

In the first equality we used the fact that the cross-entropy loss is linear in the second argument. In the second equality we used the fact that objectives 6 and 10 share the same minimizer, which we prove in Appendix C.7. Accordingly, we have that

$$
\begin{aligned}
\frac{\partial\nabla_{\tilde{\boldsymbol{\theta}}}\ell(\phi_{\tilde{\boldsymbol{\theta}}^*}(\boldsymbol{x}),\mathcal{P}(\boldsymbol{y}|\boldsymbol{x})+\boldsymbol{\epsilon})}{\partial\tilde{\boldsymbol{\theta}}_i} &= -\sum_{k=1}^{K}[\boldsymbol{\epsilon}]_k\cdot\frac{\partial[\log\phi_{\tilde{\boldsymbol{\theta}}^*}(\boldsymbol{x})]_k}{\partial\tilde{\boldsymbol{\theta}}_i} \\
&\overset{(21)}{=} -\sum_{k=1}^{K}[\boldsymbol{\epsilon}]_k\cdot\frac{1}{[\phi_{\tilde{\boldsymbol{\theta}}^*}(\boldsymbol{x})]_k}\cdot\frac{\partial[\phi_{\tilde{\boldsymbol{\theta}}^*}(\boldsymbol{x})]_k}{\partial\tilde{\boldsymbol{\theta}}_i} \\
&= -\sum_{k=1}^{K}[\boldsymbol{\epsilon}]_k\cdot\frac{1}{\mathcal{P}(\boldsymbol{y}_k|\boldsymbol{x})}\cdot\frac{\partial[\phi_{\tilde{\boldsymbol{\theta}}^*}(\boldsymbol{x})]_k}{\partial\tilde{\boldsymbol{\theta}}_i}
\end{aligned}
$$

In the third equality we used the fact that objectives 6 and 10 share the same minimizer, which we prove in Appendix C.7, and Proposition 1.

The gradient vector $\nabla_{\tilde{\boldsymbol{\theta}}}\ell(\phi_{\tilde{\boldsymbol{\theta}}^*}(\boldsymbol{x}),\mathcal{P}(\boldsymbol{y}|\boldsymbol{x})+\boldsymbol{\epsilon})$ can therefore be written as

$$
\nabla_{\tilde{\boldsymbol{\theta}}}\ell(\phi_{\tilde{\boldsymbol{\theta}}^*}(\boldsymbol{x}),\mathcal{P}(\boldsymbol{y}|\boldsymbol{x})+\boldsymbol{\epsilon}) = J_{\tilde{\boldsymbol{\theta}}}[\phi_{\tilde{\boldsymbol{\theta}}^*}(\boldsymbol{x})]\cdot\mathrm{diag}(\frac{1}{\mathcal{P}(\boldsymbol{y}|\boldsymbol{x})})\cdot\boldsymbol{\epsilon}.
$$

Finally, we are able to plug in the expression for $\nabla_{\tilde{\boldsymbol{\theta}}}\ell(\phi_{\tilde{\boldsymbol{\theta}}^*}(\boldsymbol{x}), \mathcal{P}(\boldsymbol{y}|\boldsymbol{x}) + \boldsymbol{\epsilon})$ into the definition of the gradient noise.

$$\sigma_{\tilde{f}}^* = \inf_{\tilde{\boldsymbol{\theta}}^* \in \arg\min \tilde{f}_{\mathcal{P}, \mathcal{P}_\epsilon}} \mathbb{E}_{(\boldsymbol{x}, \boldsymbol{y}) \sim \mathcal{P}, \boldsymbol{\epsilon} \sim \mathcal{P}_\epsilon} \left[ \|\nabla_{\tilde{\boldsymbol{\theta}}}\ell(\phi_{\tilde{\boldsymbol{\theta}}^*}(\boldsymbol{x}), \mathcal{P}(\boldsymbol{y}|\boldsymbol{x}) + \boldsymbol{\epsilon})\|^2 \right]$$

$$= \mathbb{E}_{(\boldsymbol{x}, \boldsymbol{y}) \sim \mathcal{P}, \boldsymbol{\epsilon} \sim \mathcal{P}_\epsilon} [\| J_{\tilde{\boldsymbol{\theta}}}[\phi_{\tilde{\boldsymbol{\theta}}^*}(\boldsymbol{x})] \cdot \operatorname{diag}(\frac{1}{\mathcal{P}(\boldsymbol{y}|\boldsymbol{x})}) \cdot \boldsymbol{\epsilon}\|^2]$$

$$= \mathbb{E}_{(\boldsymbol{x}, \boldsymbol{y}) \sim \mathcal{P}, \boldsymbol{\epsilon} \sim \mathcal{P}_\epsilon} [\boldsymbol{\epsilon}^T \cdot \operatorname{diag}(\frac{1}{\mathcal{P}(\boldsymbol{y}|\boldsymbol{x})}) \cdot J_{\tilde{\boldsymbol{\theta}}}^T[\phi_{\tilde{\boldsymbol{\theta}}^*}(\boldsymbol{x})] \cdot J_{\tilde{\boldsymbol{\theta}}}[\phi_{\tilde{\boldsymbol{\theta}}^*}(\boldsymbol{x})] \cdot \operatorname{diag}(\frac{1}{\mathcal{P}(\boldsymbol{y}|\boldsymbol{x})}) \cdot \boldsymbol{\epsilon}]$$

$$\stackrel{(22)}{=} \mathbb{E}_{(\boldsymbol{x}, \boldsymbol{y}) \sim \mathcal{P}, \boldsymbol{\epsilon} \sim \mathcal{P}_\epsilon} [\operatorname{Tr}[\boldsymbol{\epsilon} \cdot \boldsymbol{\epsilon}^T \cdot \operatorname{diag}(\frac{1}{\mathcal{P}(\boldsymbol{y}|\boldsymbol{x})}) \cdot J_{\tilde{\boldsymbol{\theta}}}^T[\phi_{\tilde{\boldsymbol{\theta}}^*}(\boldsymbol{x})] \cdot J_{\tilde{\boldsymbol{\theta}}}[\phi_{\tilde{\boldsymbol{\theta}}^*}(\boldsymbol{x})] \cdot \operatorname{diag}(\frac{1}{\mathcal{P}(\boldsymbol{y}|\boldsymbol{x})})]]$$

$$\stackrel{(23)}{=} \operatorname{Tr}[\mathbb{E}_{\boldsymbol{\epsilon} \sim \mathcal{P}_\epsilon}[\boldsymbol{\epsilon} \cdot \boldsymbol{\epsilon}^T] \cdot \mathbb{E}_{(\boldsymbol{x}, \boldsymbol{y}) \sim \mathcal{P}}[\operatorname{diag}(\frac{1}{\mathcal{P}(\boldsymbol{y}|\boldsymbol{x})}) \cdot J_{\tilde{\boldsymbol{\theta}}}^T[\phi_{\tilde{\boldsymbol{\theta}}^*}(\boldsymbol{x})] \cdot J_{\tilde{\boldsymbol{\theta}}}[\phi_{\tilde{\boldsymbol{\theta}}^*}(\boldsymbol{x})] \cdot \operatorname{diag}(\frac{1}{\mathcal{P}(\boldsymbol{y}|\boldsymbol{x})})]]$$

$$= \operatorname{Tr}[\nu \cdot I_K \cdot \mathbb{E}_{(\boldsymbol{x}, \boldsymbol{y}) \sim \mathcal{P}}[\operatorname{diag}(\frac{1}{\mathcal{P}(\boldsymbol{y}|\boldsymbol{x})}) \cdot J_{\tilde{\boldsymbol{\theta}}}^T[\phi_{\tilde{\boldsymbol{\theta}}^*}(\boldsymbol{x})] \cdot J_{\tilde{\boldsymbol{\theta}}}[\phi_{\tilde{\boldsymbol{\theta}}^*}(\boldsymbol{x})] \cdot \operatorname{diag}(\frac{1}{\mathcal{P}(\boldsymbol{y}|\boldsymbol{x})})]]$$

$$= \nu \cdot \operatorname{Tr}[\mathbb{E}_{(\boldsymbol{x}, \boldsymbol{y}) \sim \mathcal{P}}[\operatorname{diag}(\frac{1}{\mathcal{P}(\boldsymbol{y}|\boldsymbol{x})}) \cdot J_{\tilde{\boldsymbol{\theta}}}^T[\phi_{\tilde{\boldsymbol{\theta}}^*}(\boldsymbol{x})] \cdot J_{\tilde{\boldsymbol{\theta}}}[\phi_{\tilde{\boldsymbol{\theta}}^*}(\boldsymbol{x})] \cdot \operatorname{diag}(\frac{1}{\mathcal{P}(\boldsymbol{y}|\boldsymbol{x})})]]$$

$$\stackrel{(23)}{=} \nu \cdot \mathbb{E}_{(\boldsymbol{x}, \boldsymbol{y}) \sim \mathcal{P}}[\operatorname{Tr}[\operatorname{diag}(\frac{1}{\mathcal{P}(\boldsymbol{y}|\boldsymbol{x})}) \cdot J_{\tilde{\boldsymbol{\theta}}}^T[\phi_{\tilde{\boldsymbol{\theta}}^*}(\boldsymbol{x})] \cdot J_{\tilde{\boldsymbol{\theta}}}[\phi_{\tilde{\boldsymbol{\theta}}^*}(\boldsymbol{x})] \cdot \operatorname{diag}(\frac{1}{\mathcal{P}(\boldsymbol{y}|\boldsymbol{x})})]]$$

$$= \nu \cdot \mathbb{E}_{\boldsymbol{x} \sim \mathcal{P}}[\sum_{k=1}^K \|\frac{1}{\mathcal{P}(\boldsymbol{y}_k|\boldsymbol{x})} \cdot J_{\tilde{\boldsymbol{\theta}}, k}[\phi_{\tilde{\boldsymbol{\theta}}^*}(\boldsymbol{x})]\|^2]$$

$$= \nu \cdot \mathbb{E}_{\boldsymbol{x} \sim \mathcal{P}}[\sum_{k=1}^K \frac{1}{\mathcal{P}(\boldsymbol{y}_k|\boldsymbol{x})^2} \cdot \|J_{\tilde{\boldsymbol{\theta}}, k}[\phi_{\tilde{\boldsymbol{\theta}}^*}(\boldsymbol{x})]\|^2]$$

where $J_{\tilde{\boldsymbol{\theta}}, k}[\phi_{\tilde{\boldsymbol{\theta}}}(\boldsymbol{x})]$ is the $k_{th}$ column of the Jacobian of the student model.

We provided expressions for both the gradient noise (Definition 2) of the one-hot risk (1), $\sigma_f^*$, and the noisy BCP risk (10), $\sigma_{\tilde{f}}^*$, which concludes the proof. □

## C.4 PROOF OF LEMMA 1

*Proof.* In order to show that the gradient of each per-sample loss vanishes for the minimizer $\hat{\boldsymbol{\theta}}^*$ of the stochastic optimization task (6), we make use of Fermat's Theorem (proposition 8.9 in Garrigos & Gower (2023)). Fermat's Theorem states that for $\hat{f} : \mathbb{R}^d \to \mathbb{R} \cup \{+\infty\}$ and $\hat{\boldsymbol{\theta}}^* \in \mathbb{R}^d$; then $\hat{\boldsymbol{\theta}}^*$ is a minimizer of $\hat{f}$ if and only if $0 \in \partial \hat{f}(\hat{\boldsymbol{\theta}}^*)$. From the definition of interpolation (Definition 1) and from Fermat's theorem, it is implied that since interpolation holds for $\hat{f}$ at $\hat{\boldsymbol{\theta}}^*$ (Proposition 2), then $\nabla_{\hat{\boldsymbol{\theta}}}\ell(\phi_{\hat{\boldsymbol{\theta}}^*}(\boldsymbol{x}), \mathcal{P}(\boldsymbol{y}|\boldsymbol{x})) = 0$ for all $\boldsymbol{x} \in \operatorname{supp}(\mathcal{P})$, which concludes the proof. □

## C.5 PROOF OF THEOREM 1

*Proof.* In order to bound the error on the weights, we start by conditioning the expected error on the weights in the last iteration. We denote by $\mathbb{E}_t[\cdot] := \mathbb{E}[\cdot|\hat{\boldsymbol{\theta}}^t]$ the conditional expectation w.r.t $\hat{\boldsymbol{\theta}}^t$. We now have that

$$\mathbb{E}_t[\|\hat{\boldsymbol{\theta}}^{t+1} - \boldsymbol{\theta}^*\|^2] \stackrel{(5)}{=} \mathbb{E}_t[\|\hat{\boldsymbol{\theta}}^t - \alpha \nabla \hat{f}_\xi(\hat{\boldsymbol{\theta}}^t) - \boldsymbol{\theta}^*\|^2]$$

$$= \|\hat{\boldsymbol{\theta}}^t - \hat{\boldsymbol{\theta}}^*\|^2 - 2\alpha \langle \hat{\boldsymbol{\theta}}^t - \hat{\boldsymbol{\theta}}^*, \nabla \hat{f}(\hat{\boldsymbol{\theta}}^t) \rangle + \alpha^2 \mathbb{E}_t[\|\nabla \hat{f}_\xi(\hat{\boldsymbol{\theta}}^t)\|^2]$$

$$\stackrel{(\text{Lemma 1})}{=} \|\hat{\boldsymbol{\theta}}^t - \hat{\boldsymbol{\theta}}^*\|^2 - 2\alpha \langle \hat{\boldsymbol{\theta}}^t - \hat{\boldsymbol{\theta}}^*, \nabla \hat{f}(\hat{\boldsymbol{\theta}}^t) \rangle + \alpha^2 \mathbb{E}_t[\|\nabla \hat{f}_\xi(\hat{\boldsymbol{\theta}}^t) - \nabla \hat{f}_\xi(\hat{\boldsymbol{\theta}}^*)\|^2]$$

$$\stackrel{(AS1)}{\leq} (1 - \alpha\mu)\|\hat{\boldsymbol{\theta}}^t - \hat{\boldsymbol{\theta}}^*\|^2 - 2\alpha(\hat{f}(\hat{\boldsymbol{\theta}}^t) - \hat{f}(\hat{\boldsymbol{\theta}}^*)) + \alpha^2 \mathbb{E}_t[\|\nabla \hat{f}_\xi(\hat{\boldsymbol{\theta}}^t) - \nabla \hat{f}_\xi(\hat{\boldsymbol{\theta}}^*)\|^2]$$

$$\stackrel{(AS3)}{\leq} (1 - \alpha\mu)\|\hat{\boldsymbol{\theta}}^t - \hat{\boldsymbol{\theta}}^*\|^2 - 2\alpha(\hat{f}(\hat{\boldsymbol{\theta}}^t) - \hat{f}(\hat{\boldsymbol{\theta}}^*)) + 2\alpha^2 \mathcal{L}(\hat{f}(\hat{\boldsymbol{\theta}}^t) - \hat{f}(\hat{\boldsymbol{\theta}}^*))$$

$$= (1 - \alpha\mu)\|\hat{\boldsymbol{\theta}}^t - \hat{\boldsymbol{\theta}}^*\|^2 + 2\alpha(\alpha\mathcal{L} - 1)(\hat{f}(\hat{\boldsymbol{\theta}}^t) - \hat{f}(\hat{\boldsymbol{\theta}}^*))$$

$$\leq (1 - \alpha\mu)\|\hat{\boldsymbol{\theta}}^t - \hat{\boldsymbol{\theta}}^*\|^2.$$

In the last inequality we used the bound on the step-size $\alpha \leq \frac{1}{\mathcal{L}}$. Next, we apply expectation and unroll the recursion to get

$$\mathbb{E}[\|\hat{\boldsymbol{\theta}}^t - \hat{\boldsymbol{\theta}}^*\|^2] \leq (1 - \alpha\mu)\mathbb{E}[\|\hat{\boldsymbol{\theta}}^{t-1} - \hat{\boldsymbol{\theta}}^*\|^2]$$
$$\overset{(18)}{\leq} (1 - \alpha\mu)^t \|\hat{\boldsymbol{\theta}}^0 - \boldsymbol{\theta}^*\|^2,$$

which concludes the proof. $\qquad\qquad\square$

### C.6 PROOF OF THEOREM 2

*Proof.* In a similar fashion to the previous proof, we start by conditioning the expected error on the loss in the last iteration. We denote by $\mathbb{E}_t[\cdot] := \mathbb{E}[\cdot|\hat{f}^t]$ the conditional expectation w.r.t $\hat{f}^t$. We note that

$$\mathbb{E}_t[\hat{f}(\hat{\boldsymbol{\theta}}^{t+1}) - f^*] \overset{(17)}{\leq} (\hat{f}(\hat{\boldsymbol{\theta}}^t) - \hat{f}^*) - \alpha\langle\nabla\hat{f}(\hat{\boldsymbol{\theta}}^t), \nabla\hat{f}(\hat{\boldsymbol{\theta}}^t)\rangle + \frac{L\alpha^2}{2}\mathbb{E}_t[\|\nabla\hat{f}_\xi(\hat{\boldsymbol{\theta}}^t)\|^2]$$

$$\overset{(\text{Lemma 1})}{=} (\hat{f}(\hat{\boldsymbol{\theta}}^t) - \hat{f}^*) - \alpha\|\nabla\hat{f}(\hat{\boldsymbol{\theta}}^t)\|^2 + \frac{L\alpha^2}{2}\mathbb{E}_t[\|\nabla\hat{f}_\xi(\hat{\boldsymbol{\theta}}^t) - \nabla\hat{f}_\xi(\hat{\boldsymbol{\theta}}^*)\|^2]$$

$$\overset{(AS2)+(AS3)}{\leq} (\hat{f}(\hat{\boldsymbol{\theta}}^t) - \hat{f}^*) - 2\alpha\mu(\hat{f}(\hat{\boldsymbol{\theta}}^t) - \hat{f}^*) + L\mathcal{L}\alpha^2(\hat{f}(\hat{\boldsymbol{\theta}}^t) - \hat{f}^*)$$

$$= (1 - \alpha\mu)(\hat{f}(\hat{\boldsymbol{\theta}}^t) - \hat{f}^*) - \alpha(\mu - L\mathcal{L}\alpha)(\hat{f}(\hat{\boldsymbol{\theta}}^t) - \hat{f}^*)$$

$$\leq (1 - \alpha\mu)(\hat{f}(\hat{\boldsymbol{\theta}}^t - \hat{f}^*),$$

where in the last inequality we used the bound on the step-size $\alpha \leq \frac{\mu}{L\mathcal{L}}$. Next, we apply expectation and unroll the recursion to get

$$\mathbb{E}[\hat{f}(\hat{\boldsymbol{\theta}}^t) - \hat{f}^*] \leq (1 - \alpha\mu)\mathbb{E}[\hat{f}(\hat{\boldsymbol{\theta}}^{t-1}) - \hat{f}^*]$$
$$\overset{(18)}{\leq} (1 - \alpha\mu)^t(\hat{f}(\hat{\boldsymbol{\theta}}^0) - f^*),$$

Which concludes the proof. $\qquad\qquad\square$

### C.7 PROOF OF THEOREM 3

*Proof.* Before diving into the learning dynamics of the student in the noisy case, we explore the connection between the stochastic optimization problem of the noiseless case (6) and the noisy case (10). We now note that

$$\tilde{f}(\tilde{\boldsymbol{\theta}}) = \mathbb{E}_{(\boldsymbol{x},\boldsymbol{y})\sim\mathcal{P},\boldsymbol{\epsilon}\sim\mathcal{P}_\epsilon}\left[\ell(\phi_{\tilde{\boldsymbol{\theta}}}(\boldsymbol{x}), \mathcal{P}(\boldsymbol{y}|\boldsymbol{x}) + \boldsymbol{\epsilon})\right]$$

$$= \mathbb{E}_{(\boldsymbol{x},\boldsymbol{y})\sim\mathcal{P}}\left[-\sum_{k=1}^K \mathcal{P}(\boldsymbol{y}_k|\boldsymbol{x})\log\left[\phi_{\tilde{\boldsymbol{\theta}}}(\boldsymbol{x})\right]_k\right] + \mathbb{E}_{(\boldsymbol{x},\boldsymbol{y})\sim\mathcal{P},\boldsymbol{\epsilon}\sim\mathcal{P}_\epsilon}\left[-\sum_{k=1}^K \boldsymbol{\epsilon}_k\log\left[\phi_{\tilde{\boldsymbol{\theta}}}(\boldsymbol{x})\right]_k\right]$$

$$= \hat{f}(\tilde{\boldsymbol{\theta}}) + \mathbb{E}_{(\boldsymbol{x},\boldsymbol{y})\sim\mathcal{P}}\left[\mathbb{E}_{\boldsymbol{\epsilon}\sim\mathcal{P}_\epsilon}\left[-\boldsymbol{\epsilon}^T\log\phi_{\tilde{\boldsymbol{\theta}}}(\boldsymbol{x})\big|\boldsymbol{x},\boldsymbol{y}\right]\right]$$

$$= \hat{f}(\tilde{\boldsymbol{\theta}}),$$

where in the last equality we used the fact that $\boldsymbol{\epsilon}$ is independent of $\boldsymbol{x}, \boldsymbol{y}$ and is zero-mean. Note that for the current proof we change between $\hat{\boldsymbol{\theta}}^*$ and $\tilde{\boldsymbol{\theta}}^*$, which is allowed since assumption *AS1* holds, which implies uniqueness of the minimizer.

Even though we proved that the generalization error in the case of perfect probabilities and noisy probabilities is equal, the learning dynamics in the two cases are different. This is because we focus on SGD-based optimization, and not full gradient-descent. Nevertheless, importantly, this implies that $\hat{f}$ and $\tilde{f}$ share the same minimum and minimizer, which means that in our proofs we can switch between $\hat{\boldsymbol{\theta}}^*$ and $\tilde{\boldsymbol{\theta}}^*$ (assuming the minimizer is unique), as well as between $\hat{f}^*$ and $\tilde{f}^*$ interchangeably.

Again, in order to bound the error on the weights, we start by conditioning the expected error on the weights in the last iteration. We denote by $\mathbb{E}_t[\cdot] := \mathbb{E}[\cdot|\tilde{\boldsymbol{\theta}}^t]$ the conditional expectation w.r.t $\tilde{\boldsymbol{\theta}}^t$, and

obtain

$$\mathbb{E}_t[\|\tilde{\boldsymbol{\theta}}^{t+1} - \boldsymbol{\theta}^*\|^2] \stackrel{(5)}{=} \mathbb{E}_t[\|\tilde{\boldsymbol{\theta}}^t - \alpha\nabla\tilde{f}_\xi(\tilde{\boldsymbol{\theta}}^t) - \tilde{\boldsymbol{\theta}}^*\|^2]$$

$$= \|\tilde{\boldsymbol{\theta}}^t - \tilde{\boldsymbol{\theta}}^*\|^2 - 2\alpha\langle\tilde{\boldsymbol{\theta}}^t - \tilde{\boldsymbol{\theta}}^*, \nabla\tilde{f}(\tilde{\boldsymbol{\theta}}^t)\rangle + \alpha^2\mathbb{E}_t[\|\nabla\tilde{f}_\xi(\tilde{\boldsymbol{\theta}}^t)\|^2]$$

$$\stackrel{(15)}{\leq} \|\tilde{\boldsymbol{\theta}}^t - \tilde{\boldsymbol{\theta}}^*\|^2 - 2\alpha\langle\tilde{\boldsymbol{\theta}}^t - \tilde{\boldsymbol{\theta}}^*, \nabla\tilde{f}(\tilde{\boldsymbol{\theta}}^t)\rangle + 2\alpha^2\mathbb{E}_t[\|\nabla\tilde{f}_\xi(\tilde{\boldsymbol{\theta}}^t) - \nabla\tilde{f}_\xi(\tilde{\boldsymbol{\theta}}^*)\|^2] + 2\alpha^2\mathbb{E}_t[\|\nabla\tilde{f}_\xi(\tilde{\boldsymbol{\theta}}^*)\|^2]$$

$$\stackrel{(AS1)}{\leq} (1 - \alpha\mu)\|\tilde{\boldsymbol{\theta}}^t - \tilde{\boldsymbol{\theta}}^*\|^2 - 2\alpha(\tilde{f}(\tilde{\boldsymbol{\theta}}^t) - \tilde{f}(\tilde{\boldsymbol{\theta}}^*)) + 2\alpha^2\mathbb{E}_t[\|\nabla\tilde{f}_\xi(\tilde{\boldsymbol{\theta}}^t) - \nabla\tilde{f}_\xi(\tilde{\boldsymbol{\theta}}^*)\|^2] + 2\alpha^2\sigma_{\tilde{f}}^*$$

$$\stackrel{(AS3)}{\leq} (1 - \alpha\mu)\|\tilde{\boldsymbol{\theta}}^t - \tilde{\boldsymbol{\theta}}^*\|^2 - 2\alpha(\tilde{f}(\tilde{\boldsymbol{\theta}}^t) - \tilde{f}(\tilde{\boldsymbol{\theta}}^*)) + 4\alpha^2\mathcal{L}(\tilde{f}(\tilde{\boldsymbol{\theta}}^t) - \tilde{f}(\tilde{\boldsymbol{\theta}}^*)) + 2\alpha^2\sigma_{\tilde{f}}^*$$

$$= (1 - \alpha\mu)\|\tilde{\boldsymbol{\theta}}^t - \tilde{\boldsymbol{\theta}}^*\|^2 + 2\alpha(2\alpha\mathcal{L} - 1)(\tilde{f}(\tilde{\boldsymbol{\theta}}^t) - \tilde{f}(\tilde{\boldsymbol{\theta}}^*)) + 2\alpha^2\sigma_{\tilde{f}}^*$$

$$\leq (1 - \alpha\mu)\|\tilde{\boldsymbol{\theta}}^t - \tilde{\boldsymbol{\theta}}^*\|^2 + 2\alpha^2\sigma_{\tilde{f}}^*.$$

Where in the last inequality we used the bound on the step-size $\alpha \leq \frac{1}{2\mathcal{L}}$. Next, we apply expectation and unroll the recursion to get

$$\mathbb{E}[\|\tilde{\boldsymbol{\theta}}^t - \tilde{\boldsymbol{\theta}}^*\|^2] \leq (1 - \alpha\mu)\mathbb{E}[\|\tilde{\boldsymbol{\theta}}^{t-1} - \tilde{\boldsymbol{\theta}}^*\|^2] + 2\alpha^2\sigma_{\tilde{f}}^*$$

$$\stackrel{(18)}{\leq} (1 - \alpha\mu)^t\|\tilde{\boldsymbol{\theta}}^0 - \boldsymbol{\theta}^*\|^2 + \frac{2\alpha}{\mu}\sigma_{\tilde{f}}^*,$$

which concludes the proof. $\qquad\square$

## C.8 Proof of Theorem 4

*Proof.* Again, in order to bound the error on the loss, we start by conditioning the expected error on the loss in the last iteration. We denote by $\mathbb{E}_t[\cdot] := \mathbb{E}[\cdot|\tilde{f}^t]$ the conditional expectation w.r.t $\tilde{f}^t$.

Note that for the current proof we change between $\hat{f}^*$ and $\tilde{f}^*$, which is allowed even though the minimizer might not be unique, since this is the minimum value and not the minimizer.

$$\mathbb{E}_t[\tilde{f}(\tilde{\boldsymbol{\theta}}^{t+1}) - f^*] \stackrel{(17)}{\leq} (\tilde{f}(\tilde{\boldsymbol{\theta}}^t) - \tilde{f}^*) - \alpha\langle\nabla\tilde{f}(\tilde{\boldsymbol{\theta}}^t), \nabla\tilde{f}(\tilde{\boldsymbol{\theta}}^t)\rangle + \frac{L\alpha^2}{2}\mathbb{E}_t[\|\nabla\hat{f}_\xi(\tilde{\boldsymbol{\theta}}^t)\|^2]$$

$$\stackrel{(15)}{\leq} (\tilde{f}(\tilde{\boldsymbol{\theta}}^t) - \tilde{f}^*) - \alpha\|\nabla\tilde{f}(\tilde{\boldsymbol{\theta}}^t)\|^2 + L\alpha^2\mathbb{E}_t[\|\nabla\hat{f}_\xi(\tilde{\boldsymbol{\theta}}^t) - \nabla\hat{f}_\xi(\tilde{\boldsymbol{\theta}}^*)\|^2] + L\alpha^2\mathbb{E}_t[\|\nabla\hat{f}_\xi(\tilde{\boldsymbol{\theta}}^*)\|^2]$$

$$\stackrel{(AS2)+(AS3)}{\leq} (\tilde{f}(\tilde{\boldsymbol{\theta}}^t) - \tilde{f}^*) - 2\alpha\mu(\tilde{f}(\tilde{\boldsymbol{\theta}}^t) - \tilde{f}^*) + 2L\mathcal{L}\alpha^2(\tilde{f}(\tilde{\boldsymbol{\theta}}^t) - \tilde{f}^*) + L\alpha^2\sigma_{\tilde{f}}^*$$

$$= (1 - \alpha\mu)(\tilde{f}(\tilde{\boldsymbol{\theta}}^t) - \tilde{f}^*) - \alpha(\mu - 2L\mathcal{L}\alpha)(\tilde{f}(\tilde{\boldsymbol{\theta}}^t) - \tilde{f}^*) + L\alpha^2\sigma_{\tilde{f}}^*$$

$$\leq (1 - \alpha\mu)(\tilde{f}(\tilde{\boldsymbol{\theta}}^t - \tilde{f}^*) + L\alpha^2\sigma_{\tilde{f}}^*,$$

where in the last inequality we used the bound on the step-size $\alpha \leq \frac{\mu}{2L\mathcal{L}}$. Next, we apply expectation and unroll the recursion to get

$$\mathbb{E}[\tilde{f}(\tilde{\boldsymbol{\theta}}^t) - \tilde{f}^*] \leq (1 - \alpha\mu)\mathbb{E}[\tilde{f}(\tilde{\boldsymbol{\theta}}^{t-1}) - \tilde{f}^*] + L\alpha^2\sigma_{\tilde{f}}^*$$

$$\stackrel{(18)}{\leq} (1 - \alpha\mu)^t(\tilde{f}(\tilde{\boldsymbol{\theta}}^0) - f^*) + \frac{L\alpha}{\mu}\sigma_{\tilde{f}}^*,$$

which concludes the proof. $\qquad\square$

## D Dirichlet–Perturbed BCPs

Our analysis of SGD supervised by noisy BCPs provided in Section 3 used an additive noise model. In this appendix we show that the insights obtained there also hold for alternative models for noisy BCPs, using the Dirichlet distribution over probability mass functions. In this case, instead of using additive perturbations $\boldsymbol{\epsilon}$, we perturb each true BCP $\mathcal{P}(\boldsymbol{y}|\boldsymbol{x})$ using a Dirichlet distribution, which guarantees that the resulting target remains a valid probability distribution and is unbiased in expectation. Specifically, for some $\varepsilon > 0$, the perturbation is formulated as follows:

$$\bar{\mathcal{P}}(\boldsymbol{y}|\boldsymbol{x}) \sim \text{Dir}(\varepsilon\mathcal{P}(\boldsymbol{y}|\boldsymbol{x})). \tag{25}$$

Similarly to the definitions of the BCP-based risk 6 and the noisy BCP-based risk 10, the resulting *noisy Dirichlet BCP-based risk* is

$$\bar{f}_{\mathcal{P},\bar{\mathcal{P}}}(\bar{\boldsymbol{\theta}}) := \mathbb{E}_{(\boldsymbol{x},\boldsymbol{y})\sim\mathcal{P},\bar{\mathcal{P}}(\boldsymbol{y}|\boldsymbol{x})\sim\text{Dir}} \left[ \ell(\phi_{\bar{\boldsymbol{\theta}}}(\boldsymbol{x}), \bar{\mathcal{P}}(\boldsymbol{y}|\boldsymbol{x})) \right]. \tag{26}$$

**Dirichlet characteristics** For $\bar{\mathcal{P}}(\boldsymbol{y}|\boldsymbol{x}) \sim \text{Dir}(\varepsilon\mathcal{P}(\boldsymbol{y}|\boldsymbol{x}))$ we have Minka (2000)

$$\mathbb{E}\left[\bar{\mathcal{P}}(\boldsymbol{y}_k|\boldsymbol{x})|\boldsymbol{x}\right] = \mathcal{P}(\boldsymbol{y}_k|\boldsymbol{x}), \tag{27}$$

$$\text{Var}[\bar{\mathcal{P}}(\boldsymbol{y}_k|\boldsymbol{x})|\boldsymbol{x}] = \frac{\mathcal{P}(\boldsymbol{y}_k|\boldsymbol{x})(1 - \mathcal{P}(\boldsymbol{y}_k|\boldsymbol{x}))}{\varepsilon + 1}, \tag{28}$$

and

$$\text{Cov}[\bar{\mathcal{P}}(\boldsymbol{y}_i|\boldsymbol{x}), \bar{\mathcal{P}}(\boldsymbol{y}_j|\boldsymbol{x})|\boldsymbol{x}] = -\frac{\mathcal{P}(\boldsymbol{y}_i|\boldsymbol{x})\mathcal{P}(\boldsymbol{y}_j|\boldsymbol{x})}{\varepsilon + 1}. \tag{29}$$

**Proof of Theorems 3 and 4 for Dirichlet Noisy BCPs**

*Proof.* We explore the connection between the stochastic optimization problem of the noiseless case 6 and the noisy Dirichlet case 26.

$$\begin{aligned}
\bar{f}_{\mathcal{P},\bar{\mathcal{P}}}(\bar{\boldsymbol{\theta}}) &= \mathbb{E}_{(\boldsymbol{x},\boldsymbol{y})\sim\mathcal{P},\bar{\mathcal{P}}(\boldsymbol{y}|\boldsymbol{x})\sim\text{Dir}} \left[ -\sum_{k=1}^{K} \bar{\mathcal{P}}(\boldsymbol{y}_k|\boldsymbol{x}) \log[\phi_{\bar{\boldsymbol{\theta}}}(\boldsymbol{x})]_k \right] \\
&= \mathbb{E}_{(\boldsymbol{x},\boldsymbol{y})\sim\mathcal{P}} \left[ -\sum_{k=1}^{K} \mathbb{E}_{\bar{\mathcal{P}}(\boldsymbol{y}|\boldsymbol{x})\sim\text{Dir}}[\bar{\mathcal{P}}(\boldsymbol{y}_k|\boldsymbol{x}) \mid \mathcal{P}(\boldsymbol{y}|\boldsymbol{x})] \log[\phi_{\bar{\boldsymbol{\theta}}}(\boldsymbol{x})]_k \right] \\
&= \mathbb{E}_{(\boldsymbol{x},\boldsymbol{y})\sim\mathcal{P}} \left[ -\sum_{k=1}^{K} \mathcal{P}(\boldsymbol{y}_k|\boldsymbol{x}) \log[\phi_{\bar{\boldsymbol{\theta}}}(\boldsymbol{x})]_k \right] \\
&= \hat{f}_{\mathcal{P}}(\bar{\boldsymbol{\theta}}). \tag{30}
\end{aligned}$$

In the third equality in (30), we utilized the expectation characteristic of the Dirichlet distribution 27. Similarly to the proof of Theorem 3, even though we proved that the generalization error in the case of perfect probabilities and noisy Dirichlet probabilities is equal, the learning dynamics in the two cases are different. This is because we focus on SGD-based learning. Nevertheless, importantly this implies that $\hat{f}$ and $\bar{f}$ share the same minimum and minimizer, which means that in our proofs we can switch between $\hat{\boldsymbol{\theta}}^*$ and $\bar{\boldsymbol{\theta}}^*$ (assuming the minimizer is unique), as well as between $\hat{f}^*$ and $\bar{f}^*$ interchangeably.

The remainder of the proofs remain exactly the same as in Appendix C.7 and C.8, but with switching between $\tilde{f}$ and $\bar{f}$, and between $\tilde{\boldsymbol{\theta}}$ and $\bar{\boldsymbol{\theta}}$. The difference lies in the resulting expression for the gradient noise, which depends on how the noisy BCPs are modeled, as stated in the adaptation of Proposition 3 detailed below.

$\square$

**Adaptation of Proposition 3 for Dirichlet Noisy BCPs**

*Proof.* In order to express the gradient noise of the noisy Dirichlet BCP risk (26), we recall the resulting expressions for the gradient noise based on Definition 2, i.e.,

$$\sigma_{\bar{f}}^* = \inf_{\bar{\boldsymbol{\theta}}^*\in\arg\min \bar{f}_{\mathcal{P},\bar{\mathcal{P}}}} \mathbb{E}_{(\boldsymbol{x},\boldsymbol{y})\sim\mathcal{P}, \bar{\mathcal{P}}(\boldsymbol{y}|\boldsymbol{x})} \left[ \|\nabla_{\bar{\boldsymbol{\theta}}}\ell(\phi_{\bar{\boldsymbol{\theta}}^*}(\boldsymbol{x}), \bar{\mathcal{P}}(\boldsymbol{y}|\boldsymbol{x}))\|^2 \right].$$

We first give an expression for $\nabla_{\bar{\boldsymbol{\theta}}}\ell(\phi_{\bar{\boldsymbol{\theta}}^*}(\boldsymbol{x}), \bar{\mathcal{P}}(\boldsymbol{y}|\boldsymbol{x}))$. In this case, $\mathcal{P}(\boldsymbol{y}|\boldsymbol{x})$ is the true BCP which is a $K$-sized vector, and $\bar{\mathcal{P}}(\boldsymbol{y}|\boldsymbol{x})$ is its Dirichlet-perturbed version. Since the CE loss is linear in its second argument, we have

$$\begin{aligned}
\nabla_{\bar{\boldsymbol{\theta}}}\ell(\phi_{\bar{\boldsymbol{\theta}}^*}(\boldsymbol{x}), \bar{\mathcal{P}}(\boldsymbol{y}|\boldsymbol{x})) &= \nabla_{\bar{\boldsymbol{\theta}}}\ell(\phi_{\bar{\boldsymbol{\theta}}^*}(\boldsymbol{x}), \mathcal{P}(\boldsymbol{y}|\boldsymbol{x})) + \nabla_{\bar{\boldsymbol{\theta}}}\ell(\phi_{\bar{\boldsymbol{\theta}}^*}(\boldsymbol{x}), \bar{\mathcal{P}}(\boldsymbol{y}|\boldsymbol{x}) - \mathcal{P}(\boldsymbol{y}|\boldsymbol{x})) \\
&= \nabla_{\bar{\boldsymbol{\theta}}}\ell(\phi_{\hat{\boldsymbol{\theta}}^*}(\boldsymbol{x}), \mathcal{P}(\boldsymbol{y}|\boldsymbol{x})) + \nabla_{\bar{\boldsymbol{\theta}}}\ell(\phi_{\bar{\boldsymbol{\theta}}^*}(\boldsymbol{x}), \bar{\mathcal{P}}(\boldsymbol{y}|\boldsymbol{x}) - \mathcal{P}(\boldsymbol{y}|\boldsymbol{x})) \\
&\overset{\text{(Lemma 1) + (AS4) + (Prop 2)}}{=} \nabla_{\bar{\boldsymbol{\theta}}}\ell(\phi_{\bar{\boldsymbol{\theta}}^*}(\boldsymbol{x}), \delta(\boldsymbol{x})),
\end{aligned}$$

where we defined $\delta(\boldsymbol{x}) := \bar{\mathcal{P}}(\boldsymbol{y}|\boldsymbol{x}) - \mathcal{P}(\boldsymbol{y}|\boldsymbol{x})$. The second equality uses the fact that objectives (6) and (26) share the same minimizer, which we just proved above.

This means that we have,

$$\nabla_{\bar{\boldsymbol{\theta}}}\ell(\phi_{\bar{\boldsymbol{\theta}}^*}(\boldsymbol{x}), \bar{\mathcal{P}}(\boldsymbol{y}|\boldsymbol{x})) = \nabla_{\bar{\boldsymbol{\theta}}}\ell(\phi_{\bar{\boldsymbol{\theta}}^*}(\boldsymbol{x}), \delta(\boldsymbol{x}))$$
$$= -\sum_{k=1}^{K} \delta_k(\boldsymbol{x})\,\nabla_{\bar{\boldsymbol{\theta}}}\log[\phi_{\bar{\boldsymbol{\theta}}^*}(\boldsymbol{x})]_k$$
$$= J_{\bar{\boldsymbol{\theta}}}[\phi_{\bar{\boldsymbol{\theta}}^*}(\boldsymbol{x})] \cdot \mathrm{diag}(\frac{1}{\mathcal{P}(\boldsymbol{y}|\boldsymbol{x})}) \cdot \delta(\boldsymbol{x}),$$

where $J_{\bar{\boldsymbol{\theta}}}[\phi_{\bar{\boldsymbol{\theta}}^*}(\boldsymbol{x})]$ is the Jacobian of the student outputs with respect to $\bar{\boldsymbol{\theta}}$.

We can now plug the expression for $\nabla_{\bar{\boldsymbol{\theta}}}\ell(\phi_{\bar{\boldsymbol{\theta}}^*}(\boldsymbol{x}), \bar{\mathcal{P}}(\boldsymbol{y}|\boldsymbol{x}))$ into the definition of the gradient noise.

$$\sigma_{\bar{f}}^* = \inf_{\bar{\boldsymbol{\theta}}^* \in \arg\min \bar{f}_{\mathcal{P},\bar{\mathcal{P}}}} \mathbb{E}_{(\boldsymbol{x},\boldsymbol{y})\sim\mathcal{P},\,\bar{\mathcal{P}}(\boldsymbol{y}|\boldsymbol{x})}\left[\|\nabla_{\bar{\boldsymbol{\theta}}}\ell(\phi_{\bar{\boldsymbol{\theta}}^*}(\boldsymbol{x}), \bar{\mathcal{P}}(\boldsymbol{y}|\boldsymbol{x}))\|^2\right]$$

$$= \mathbb{E}_{(\boldsymbol{x},\boldsymbol{y})\sim\mathcal{P},\,\delta(\boldsymbol{x})}\left[\|J_{\bar{\boldsymbol{\theta}}}[\phi_{\bar{\boldsymbol{\theta}}^*}(\boldsymbol{x})] \cdot \mathrm{diag}(\frac{1}{\mathcal{P}(\boldsymbol{y}|\boldsymbol{x})}) \cdot \delta(\boldsymbol{x})\|^2\right]$$

$$= \mathbb{E}_{(\boldsymbol{x},\boldsymbol{y})\sim\mathcal{P},\,\delta(\boldsymbol{x})}\left[\delta(\boldsymbol{x})^{\top}\mathrm{diag}(\frac{1}{\mathcal{P}(\boldsymbol{y}|\boldsymbol{x})})J_{\bar{\boldsymbol{\theta}}}^{T}[\phi_{\bar{\boldsymbol{\theta}}^*}(\boldsymbol{x})]J_{\bar{\boldsymbol{\theta}}}[\phi_{\bar{\boldsymbol{\theta}}^*}(\boldsymbol{x})]\mathrm{diag}(\frac{1}{\mathcal{P}(\boldsymbol{y}|\boldsymbol{x})})\delta(\boldsymbol{x})\right]$$

$$\overset{(22)}{=} \mathbb{E}_{(\boldsymbol{x},\boldsymbol{y})\sim\mathcal{P},\,\delta(\boldsymbol{x})}\left[\mathrm{Tr}[\delta(\boldsymbol{x})\delta(\boldsymbol{x})^{\top}\mathrm{diag}(\frac{1}{\mathcal{P}(\boldsymbol{y}|\boldsymbol{x})})J_{\bar{\boldsymbol{\theta}}}^{T}[\phi_{\bar{\boldsymbol{\theta}}^*}(\boldsymbol{x})]J_{\bar{\boldsymbol{\theta}}}[\phi_{\bar{\boldsymbol{\theta}}^*}(\boldsymbol{x})]\mathrm{diag}(\frac{1}{\mathcal{P}(\boldsymbol{y}|\boldsymbol{x})})]\right]$$

$$\overset{(23)}{=} \mathbb{E}_{(\boldsymbol{x},\boldsymbol{y})\sim\mathcal{P}}\left[\mathrm{Tr}[\mathbb{E}_{\delta(\boldsymbol{x})|\boldsymbol{x}}[\delta(\boldsymbol{x})\delta(\boldsymbol{x})^{\top}|\boldsymbol{x}]\,\mathrm{diag}(\frac{1}{\mathcal{P}(\boldsymbol{y}|\boldsymbol{x})})J_{\bar{\boldsymbol{\theta}}}^{T}[\phi_{\bar{\boldsymbol{\theta}}^*}(\boldsymbol{x})]\,J_{\bar{\boldsymbol{\theta}}}[\phi_{\bar{\boldsymbol{\theta}}^*}(\boldsymbol{x})]\,\mathrm{diag}(\frac{1}{\mathcal{P}(\boldsymbol{y}|\boldsymbol{x})})]\right].$$

We now utilize the covariance characteristic 29 of the Dirichlet distribution, which means that

$$\mathbb{E}_{\delta(\boldsymbol{x})|\boldsymbol{x}}[\delta(\boldsymbol{x})\delta(\boldsymbol{x})^{\top}|\boldsymbol{x}] = \frac{1}{\varepsilon+1}\left(\mathrm{diag}(\mathcal{P}(\boldsymbol{y}|\boldsymbol{x})) - \mathcal{P}(\boldsymbol{y}|\boldsymbol{x})\mathcal{P}(\boldsymbol{y}|\boldsymbol{x})^{\top}\right).$$

Therefore,

$$\sigma_{\bar{f}}^* = \frac{1}{\varepsilon+1}\mathbb{E}_{(\boldsymbol{x},\boldsymbol{y})\sim\mathcal{P}}[\mathrm{Tr}[(\mathrm{diag}(\mathcal{P}(\boldsymbol{y}|\boldsymbol{x})) - \mathcal{P}(\boldsymbol{y}|\boldsymbol{x})\mathcal{P}(\boldsymbol{y}|\boldsymbol{x})^{\top})\,\mathrm{diag}(\frac{1}{\mathcal{P}(\boldsymbol{y}|\boldsymbol{x})})\cdot$$
$$\cdot J_{\bar{\boldsymbol{\theta}}}^{T}[\phi_{\bar{\boldsymbol{\theta}}^*}(\boldsymbol{x})]\,J_{\bar{\boldsymbol{\theta}}}[\phi_{\bar{\boldsymbol{\theta}}^*}(\boldsymbol{x})]\,\mathrm{diag}(\frac{1}{\mathcal{P}(\boldsymbol{y}|\boldsymbol{x})})]]. \tag{31}$$

Finally, using the simplex identity $J_{\bar{\boldsymbol{\theta}}}[\phi_{\bar{\boldsymbol{\theta}}^*}(\boldsymbol{x})]\mathbf{1} = 0$, the second term with $\mathcal{P}(\boldsymbol{y}|\boldsymbol{x})\mathcal{P}(\boldsymbol{y}|\boldsymbol{x})^{\top}$ vanishes. We thus obtain

$$\sigma_{\bar{f}}^* = \frac{1}{\varepsilon+1}\mathbb{E}_{(\boldsymbol{x},\boldsymbol{y})\sim\mathcal{P}}[\mathrm{Tr}[\mathrm{diag}(\mathcal{P}(\boldsymbol{y}|\boldsymbol{x}))\,\mathrm{diag}(\frac{1}{\mathcal{P}(\boldsymbol{y}|\boldsymbol{x})})\,J_{\bar{\boldsymbol{\theta}}}^{T}[\phi_{\bar{\boldsymbol{\theta}}^*}(\boldsymbol{x})]\,J_{\bar{\boldsymbol{\theta}}}[\phi_{\bar{\boldsymbol{\theta}}^*}(\boldsymbol{x})]\,\mathrm{diag}(\frac{1}{\mathcal{P}(\boldsymbol{y}|\boldsymbol{x})})]]$$

$$= \frac{1}{\varepsilon+1}\mathbb{E}_{(\boldsymbol{x},\boldsymbol{y})\sim\mathcal{P}}[\mathrm{Tr}[J_{\bar{\boldsymbol{\theta}}}^{T}[\phi_{\bar{\boldsymbol{\theta}}^*}(\boldsymbol{x})]\,J_{\bar{\boldsymbol{\theta}}}[\phi_{\bar{\boldsymbol{\theta}}^*}(\boldsymbol{x})]\,\mathrm{diag}(\frac{1}{\mathcal{P}(\boldsymbol{y}|\boldsymbol{x})})]]$$

$$= \frac{1}{\varepsilon+1}\mathbb{E}_{\boldsymbol{x}\sim\mathcal{P}}\left[\sum_{k=1}^{K}\frac{1}{\mathcal{P}(\boldsymbol{y}_k|\boldsymbol{x})}\|J_{\bar{\boldsymbol{\theta}},k}[\phi_{\bar{\boldsymbol{\theta}}^*}(\boldsymbol{x})]\|^2\right],$$

where $J_{\bar{\boldsymbol{\theta}},k}$ is the $k$-th column of the Jacobian. This shows that, compared to the additive uncorrelated noise model, the gradient-noise expression has: $(i)$ a scale factor $(\varepsilon+1)^{-1}$ instead of $\nu$; and $(ii)$ weighting by $1/\mathcal{P}(\boldsymbol{y}_k|\boldsymbol{x})$ instead of $1/\mathcal{P}(\boldsymbol{y}_k|\boldsymbol{x})^2$, due to the Dirichlet covariance structure.

$\square$

# E    SYNTHETIC EXPERIMENTS (SECTION 3)

In this section, we empirically demonstrate the effects of different supervisory signals on the learning dynamics of the student in SGD-based learning. Specifically, we consider the cases of supervision by one-hot labels (standard learning), perfect BCPs, BCPs corrupted by different noise levels, and supervision by combinations of one-hot labels and noisy BCPs adjusted based on several $\lambda$ values. Since the true BCPs are unknown for popular datasets such as MNIST or CIFAR, we generate a synthetic dataset for which we are able to calculate the true BCPs.

We generate a dataset of $N = 5 \times 10^4$ samples. Each input $\boldsymbol{x}_n$, which is a $30 \times 1$ vector, i.e., $\mathcal{X} = \mathbb{R}^{30}$, is associated with a label $y_n$ that takes a value from $K = 5$ potential classes. To get a sample, we first select the label $y_n = k$ from a uniform distribution over the $K$ classes. Each class is centered around $\boldsymbol{\mu}_k$, a $30 \times 1$ vector where each entry is randomly selected from $\{-1, 0, 1\}$. Once the label is selected, the input $\boldsymbol{x}_n$ is drawn from a Gaussian distribution centered at the selected class mean $\boldsymbol{x}|_{y=k} \sim \mathcal{N}(\boldsymbol{\mu}_k, \sigma^2 I)$. We use $\sigma^2 = 2.5$, which is the noise level for all samples. For this dataset, we are able to calculate the true BCP $p(y_n|\boldsymbol{x}_n)$ for each pair $(\boldsymbol{x}_n, y_n)$. The true BCP is calculated as follows.

In order to calculate $p(y|\boldsymbol{x})$ we use the Bayes' rule

$$p(y|\boldsymbol{x}) = \frac{p(\boldsymbol{x}|y)\, p(y)}{\sum\limits_{j=1}^{K} p(\boldsymbol{x}|y=j)\, p(y=j)}.$$

Since the label is sampled uniformly, $p(y) = \frac{1}{K}$ for all classes. We know that

$$p(\boldsymbol{x}|y=k) \sim \mathcal{N}(\boldsymbol{\mu}_k, \sigma^2 I),$$

so we have

$$p(\boldsymbol{x}|y=k) = \frac{1}{(2\pi\sigma^2)^{\frac{30}{2}}} \exp\left(-\frac{\|\boldsymbol{x} - \boldsymbol{\mu}_k\|_2^2}{2\sigma^2}\right).$$

If we plug the above expression into Bayes' rule, the expression simplifies into

$$p(y=k|\boldsymbol{x}) = \frac{\exp\left(-\frac{\|\boldsymbol{x} - \boldsymbol{\mu}_k\|_2^2}{2\sigma^2}\right)}{\sum\limits_{j=1}^{K} \exp\left(-\frac{\|\boldsymbol{x} - \boldsymbol{\mu}_j\|_2^2}{2\sigma^2}\right)}.$$

This allows us to compute the true BCP for each input.

We split our data into 50% for the train set and 50% for the test set. A total of 22 student models supervised by different signals were trained. All models were trained in the same manner for 45,000 iterations via SGD, with a learning rate of $5 \times 10^{-4}$. For the model architecture, we used a standard MLP with 2 hidden layers, each with 128 hidden units and ReLU activation functions. The first student model, referred to as "One-hot labels" was trained with the standard labels $y_n$. The second student model, referred to as "True Bayes probabilities" was supervised with the true BCPs $p(y_n|\boldsymbol{x}_n)$. This student corresponds to subsection 3.2 in which the dynamics of a student trained with the true BCPs are analyzed. The two student models, referred to as "Less noisy probabilities" and "More noisy probabilities" were supervised with the true BCPs corrupted with different noise levels. These students correspond to subsection 3.3 in which the dynamics of a student trained with noisy BCPs are analyzed. The noise was added by perturbing each BCP using a Dirichlet distribution, which guarantees that the resulting noisy BCP remains a valid probability distribution and is unbiased in expectation. Specifically, for some $\varepsilon > 0$, we sample

$$\tilde{\boldsymbol{p}}_n \sim \mathrm{Dir}\left(\varepsilon\, \boldsymbol{p}_n\right),$$

so that $\mathbb{E}[\tilde{\boldsymbol{p}}_n] = \boldsymbol{p}_n$. $\varepsilon$ controls how noisy the resulting BCPs are. For the "Less noisy probabilities" student we used $\varepsilon = 5$ and for the "More noisy probabilities" student we used $\varepsilon = 0.5$. The rest of the student models were supervised by combinations of one-hot labels and noisy BCPs adjusted based on different $\lambda$ values between 0 and 1, as in the standard KD framework. The objective in this case is described in (3).

In the left plot of Figure 1 we plot the generalization error of the model that is calculated using Equation (1) with the finite test set. The constant line referred to as the "Bayes classifier" is the

generalization error calculated with a perfect model that produces the true BCPs. In the middle plot of Figure 1 we plot the accuracy of the model on the test set in each iteration.

Finally, the right plot of Figure 1 illustrates how the generalization error behaves during training for different values of $\lambda$. Each point in the plot represents the average $L_1$ distance between the generalization error of a given model and the minimal generalization error achieved by the optimal Bayes classifier. Specifically, for a model trained with a specific $\lambda$, noisy BCPs, and one-hot labels, we compute

$$\sigma_f^* \approx \frac{1}{T} \sum_{i=t_0}^{t_0+T} [\ell_i^f - \ell^{\mathrm{perf}}],$$

where $\ell^{\mathrm{perf}}$ denotes the generalization error achieved by the optimal Bayes classifier, and $\ell_i^f$ is the generalization error at iteration $i$ of the corresponding noisy model. We use $t_0 = 20{,}000$, which is after the initial phase of rapid convergence. This metric represents the average generalization error gap between a given student model and the reference optimal loss throughout the training process, capturing both the overall error level and its variability.

Next, we quantitatively assess the effect of noise in the BCPs on the convergence behavior of student models. We introduce two complementary metrics, each applied to both accuracy and generalization error during the final convergence stage of training.

The first metric captures the average performance in the last $N$ iterations:

$$L_{\mathrm{avg}} = \frac{1}{N} \sum_{t=T-N+1}^{T} L_t, \qquad \mathrm{ACC}_{\mathrm{avg}} = \frac{1}{N} \sum_{t=T-N+1}^{T} A_t,$$

where $L_t$ and $A_t$ denote the generalization error and accuracy at iteration $t$, respectively. This metric quantifies the achievable performance while suppressing fluctuations in the curves.

The second metric captures the stability of training by measuring the noise level in the accuracy and generalization error curves. To smooth out slow transitions, we first compute a moving average $\bar{X}_t$ with window size $w$ for a time series $X_t \in \{L_t, A_t\}$. The noise is then calculated by:

$$\sigma_X = \sqrt{\frac{1}{N} \sum_{t=T-N+1}^{T} \left(X_t - \bar{X}_t\right)^2},$$

which reflects how stable the model's performance is.

We computed these four metrics for student models trained with different levels of noise corrupting the true BCPs. The results are shown in Figure 3. Each experiment was performed 15 times; standard deviations of the computed values are shown in the figure. The bottom plots show that as the noise level decreases, average accuracy improves, while its variability decreases, indicating improved student performance. The top plots show that for the generalization error metrics, lower noise leads to both reduced error and reduced variability. In Appendix D, we proved that the gradient noise in the convergence analysis of Theorems 3 and 4, the adapted case for Proposition 3 in the case of Dirichlet noise, is:

$$\sigma_{\bar{f}}^* = \frac{1}{\varepsilon + 1} \, \mathbb{E}_{\boldsymbol{x} \sim \mathcal{P}} \left[ \sum_{k=1}^{K} \frac{1}{\mathcal{P}(\boldsymbol{y}_k | \boldsymbol{x})} \left\| J_{\bar{\boldsymbol{\theta}}, k}[\phi_{\bar{\boldsymbol{\theta}}^*}(\boldsymbol{x})] \right\|^2 \right].$$

As seen in Figure 3, for all four accuracy and generalization error metrics, the performance aligns with the fitted curves proportional to $\frac{1}{1+\varepsilon}$. These results provide empirical validation for Proposition 3, demonstrating that decreasing the noise added to the true BCPs reduces variance and improves convergence behavior.

Next, we examine the relationship between average performance and stability, and between accuracy and generalization error. Figure 4 plots the four metrics against each other for different noise levels. Each experiment was performed 15 times; standard deviations of the computed values are shown in the figure. For comparison, we also include the student trained with the true BCPs and the one trained from one-hot labels. The plots show that higher average accuracy consistently coincides with lower variability, and the same pattern can be seen for the generalization error. Moreover, accuracy behaves similarly to the generalization error, both in terms of average values and noise. These findings empirically validate two key points: (1) the variance reduction from using less noisy BCPs (Theorem 4), translates directly into improved average performance and stability; and (2) improvements in generalization error are reflected as similar improvements in accuracy.

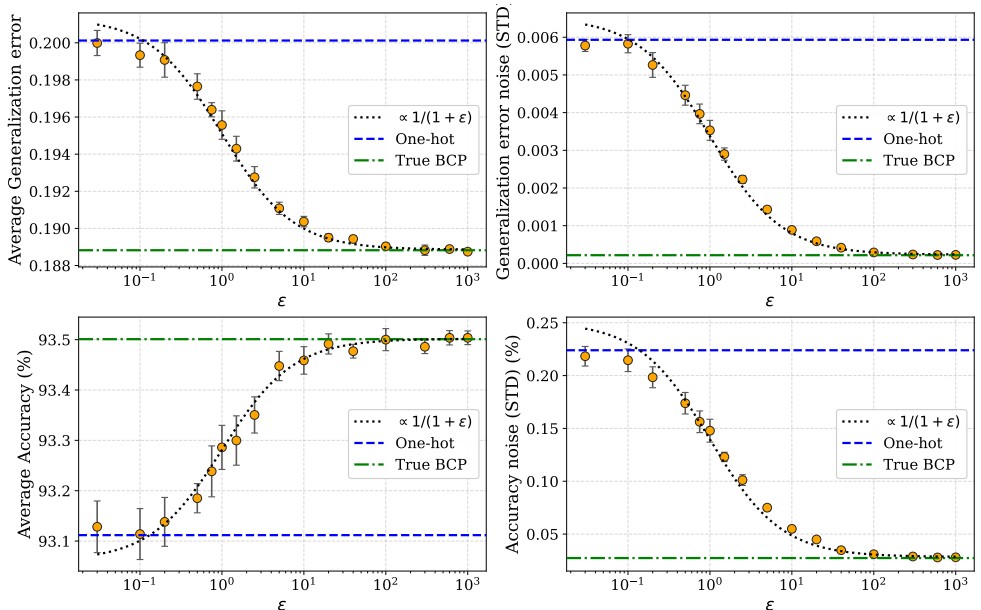

Figure 3: Effect of noise level $\epsilon$ on student performance. Each plot shows results for students trained with noisy BCPs. Top: average generalization error (left) and variability in generalization error (right). Bottom: average test accuracy (left) and variability in accuracy (right). Each plot includes a fit proportional to $\dfrac{1}{1+\varepsilon}$, alongside the result achieved by a student trained from One-hot labels, and a student trained with the true BCPs.

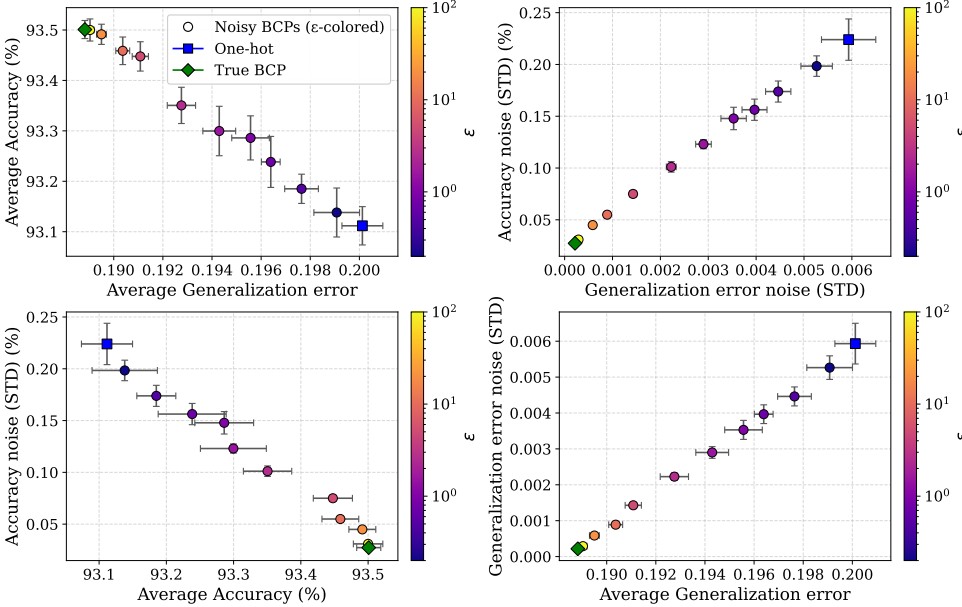

Figure 4: Correlations between performance and stability metrics across different noise levels $\epsilon$. Each point corresponds to a student trained with noisy BCPs, with baselines for one-hot labels and true BCPs also included. Top left: higher accuracy is strongly associated with lower generalization error. Top right: models with noisier generalization error curves also display noisier accuracy curves. Bottom left: higher accuracy coincides with reduced variability in accuracy. Bottom right: lower generalization error coincides with reduced generalization error variability.

## F  Implementation Details of the Experiments in Section 4

We elaborate on the implementation and training procedures of all teachers and students used in the experiments reported in Subsection 4.2. Importantly, all students were trained with an identical procedure, independent of the type of teacher (deterministic, Bayesian, Laplace, MCMI, TTDA, or MSE). The base training setup for students was the same as that of the deterministic teacher. Modifications were performed only where required by the specific teacher type:

1. *Deterministic teacher:* trained identically to the student.
2. *Bayesian teacher:* trained identically to the student in terms of hyperparameters, but with an additional NLL loss term for variational inference.
3. *Laplace teacher:* obtained by applying the Laplace approximation post-hoc to a pretrained deterministic teacher.
4. *MCMI teacher:* obtained by fine-tuning a pretrained deterministic teacher with an additional MCMI loss term.
5. *MSE teacher:* trained identically to the student but with the MSE loss replacing the standard CE loss.
6. *TTDA teacher:* predictions are obtained by applying test-time data augmentation as a drop-in method, where the pretrained deterministic teacher is evaluated on multiple augmented versions of each input and the outputs are averaged.

For all teacher and student training procedures, we train for 200 epochs, with ADAM as the optimizer. The initial learning rate is set to 0.001 by default, which is decayed by a factor of 0.1 at epoch 100. The momentum was set to 0.9, alongside a batch size of 100. Deterministic teachers were trained with the standard CE loss, and all students were trained with the distillation loss (3) with weighting parameter $\lambda$. Additionally, the teacher and student probabilities used in (3) were smoothed using $T_t$ and $T_s$, respectively.

For each teacher–student pair, we trained five independent trials under six different $(\lambda, T_t, T_s)$ configurations: $\{1, 1, 1\}, \{1, 2, 2\}, \{1, 4, 4\}, \{1, 2, 1\}, \{1, 4, 1\}, \{0.474, 4, 4\}$. The first combination corresponds to full distillation without temperature scaling, and the last combination corresponds to the original implementation in Hinton et al. (2015) with $\alpha = 0.9$ and $T = 4$. For each pair, reported values correspond to the $(\lambda, T_t, T_s)$ combination in which the best average student accuracy over the five trials was achieved. The reported noise values in Figure 2 were first computed per trial and then averaged across the five runs of the best configuration.

**Bayesian teacher**   To train Bayesian teachers with variational inference, we utilized the implementation provided in `https://github.com/microsoft/bayesianize`. This implementation turns a deterministic NN into a Bayesian NN with a configuration file that includes priors on the weights and related settings. We used the default configuration offered in their repository. Their training procedure includes an additional NLL loss term, in which we used their default schedule of 100 epochs with the NLL term and 50 epochs of gradual annealing. For distillation, teacher logits were obtained via Monte Carlo prediction averaging with 10 stochastic forward passes.

**Laplace teacher**   As described earlier, Laplace teachers are obtained by applying the Laplace approximation post-hoc to pretrained deterministic teachers. For this purpose, we used the implementation provided in `https://github.com/aleximmer/Laplace`, which allows loading pretrained model parameters and applying the approximation directly. In our experiments, we approximate the posterior distribution only over the weights of the final layer, using a Kronecker-factored Hessian to capture curvature information. The prior precision is optimized via marginal likelihood maximization on the training data, without requiring a validation split. For predictions, we employ the generalized linear model formulation with a closed-form probit approximation, which incorporates uncertainty from the Laplace posterior into the predictive probabilities. As a result, instead of point estimates from a deterministic teacher, the Laplace teacher produces predictive distributions that reflect uncertainty in the last-layer weights.

**MCMI teacher**   To realize the MCMI teacher proposed by Ye et al. (2024), we used the official implementation from `https://github.com/iclr2024mcmi/ICLRMCMI`. The approach

fine-tunes a pretrained deterministic teacher by adding an MCMI loss term. We used the hyperparameters which the authors recommend in their paper for fine-tuning. Specifically, we used a cosine annealing learning-rate schedule with an initial value of $2 \times 10^{-4}$, fine-tuned for 20 epochs, and set the MCMI weighting parameter to 0.2 for ResNet-18, ResNet-50, and VGG-13 teachers, and 0.15 for WRN-40-2. The MCMI weighting values correspond to the best settings reported in the original paper, which we adopted without hyperparameter searching, as they did. For the ResNet-18 teacher, no best value was reported, and we used 0.2 by default. It is worth noting that training hyperparameters (epochs, batch size, etc.) used in their training setup differ from those employed in this work. Additionally, in their work, they reported results for other values of $(\lambda, T_t, T_s)$ and did not run several combinations while reporting the best one. This may partly explain performance differences between our results and those reported in their paper.

**MSE teacher** To realize the Mean Squared Error (MSE) teacher proposed by Hamidi et al. (2024), we used the official implementation from `https://github.com/ECCV2024MSE`. In this setting, teacher models are trained with the MSE loss instead of the standard CE loss, as suggested by the authors. It is important to note, however, that the results reported in their paper were obtained under different experimental settings and hyperparameters than those used in this work. Additionally, they reported results for other values of $(\lambda, T_t, T_s)$ and did not run several combinations while reporting the best one. These differences may explain why the MSE teacher performs poorly in our experiments compared to other teacher types. In Appendix K, it is shown that for the case of standard distillation setting (temperatures and $\lambda$ are equal to 1), the MSE teacher performs well and often outperforms the deterministic teacher.

**TTDA teacher** As described earlier, predictions obtained from the TTDA teachers are obtained by applying the test-time data augmentation method Gawlikowski et al. (2023) to the outputs of the pre-trained deterministic teacher. Specifically, the teacher predictions were obtained by averaging the output probabilities over $N = 10$ stochastic forward passes, each computed on a differently augmented version of the same input. For each pass, a standard test-time data augmentation pipeline is applied, consisting of random cropping and random rotation. The resulting probability vectors are averaged to form the final teacher prediction used in the distillation process.

## G   FEW-SHOT CLASSIFICATION

Few-shot learning refers to training a model when only a limited number of labeled data is available for each class Chen et al. (2019). In the context of KD, the few-shot setting is adapted by utilizing a teacher that was trained on the full dataset, while the student is trained with limited data, specifically a $\beta$-proportion of the samples from every class Ye et al. (2024). Few-shot experiments allow us to examine how well the student can benefit from distillation when the data available for its training is limited. Additionally, it highlights the benefits of distillation and its ability to improve generalization in such scenarios.

We evaluate the effectiveness of Bayesian teachers in several few-shot settings on the CIFAR-100 dataset. We consider VGG-13 as the teacher architecture, and VGG-8, ResNet-18, WRN-40-2, and WRN-40-1 as student architectures. Students are trained either from Bayesian teachers or from deterministic teachers, and we compare their performance. For each teacher–student pair, we employ the $(T_t, T_s, \lambda)$ configuration that yielded the best performance in Section 4.2. Both teacher and student training follow the same implementation details and hyperparameters described in Appendix F. Experiments are conducted for different values of $\beta \in \{5, 10, 15, 25, 35, 50\}$, and each setting is repeated five times. For a fair comparison, we use the same partition of the training set across all methods in each few-shot level. The results are reported in Table 2.

Table 2: Test accuracy (%) of students trained from both Bayesian teachers and deterministic teachers, under few-shot setting, averaged over 5 runs. Results are displayed for several teacher-student pairs with both matching and different architectures. The subscript denotes improvement in the student trained from the Bayesian teacher relative to the corresponding student trained from the deterministic teacher.

| Teacher → Student | 5 | | 10 | | 15 | | 25 | | 35 | | 50 | |
|---|---|---|---|---|---|---|---|---|---|---|---|---|
| Teacher kind | Deter. | Bayesian | Deter. | Bayesian | Deter. | Bayesian | Deter. | Bayesian | Deter. | Bayesian | Deter. | Bayesian |
| VGG13 → ResNet18 | 35.79 | 41.11 +5.32 | 50.97 | 55.57 +4.61 | 58.94 | 63.43 +4.49 | 65.75 | 68.83 +3.08 | 69.33 | 71.87 +2.54 | 72.27 | 74.04 +1.77 |
| VGG13 → VGG8 | 41.57 | 52.45 +10.88 | 54.23 | 63.07 +8.84 | 61.28 | 67.97 +6.69 | 66.65 | 71.44 +4.79 | 69.50 | 73.42 +3.92 | 72.38 | 75.33 +2.96 |
| VGG13 → WRN-40-2 | 30.80 | 40.96 +10.16 | 44.10 | 54.17 +10.07 | 52.04 | 61.44 +9.40 | 59.76 | 67.33 +7.57 | 64.45 | 70.20 +5.76 | 68.08 | 72.53 +4.45 |
| VGG13 → WRN-40-1 | 24.53 | 32.88 +8.35 | 35.49 | 45.08 +9.59 | 43.21 | 52.76 +9.55 | 50.40 | 59.19 +8.79 | 55.67 | 63.45 +7.78 | 60.34 | 66.39 +6.05 |

As seen, students trained with Bayesian teachers consistently perform better across the board, compared to students trained with deterministic teachers. Specifically, as $\beta$ decreases the improvements are more substantial, with improvements of over 10% for the case of $\beta = 5$. These results show that Bayesian teachers are very useful in KD few-shot settings, consistently offering improved performance compared to deterministic teachers.

## H   TEMPERATURE SCALING

Temperature scaling was first introduced in KD by Hinton et al. (2015) as a way to soften the teacher's output probabilities. The temperature is introduced inside the "softmax" operation applied to the output of the network:

$$p^{(i)} = \frac{\exp(z^{(i)}/T)}{\sum_j \exp(z^{(j)}/T)},$$

where $z$ are the logits, $p^{(i)}$ is the softened probability for class $i$, and $T$ is the temperature. Choosing $T = 1$ corresponds to standard softmax without temperature scaling. Increasing $T$ produces a smoother distribution, which has been shown to act as a regularizer and improve the transfer of dark knowledge. Hinton et al. (2015) applied the same temperature to the student's logits during training, but this is not necessary. In fact, a recent work by Zheng & Yang (2024) studied temperature scaling, and shows that dropping temperature scaling on the student side causes the student to generalize better. From a calibration perspective, temperature scaling also serves as a simple yet effective method to reduce overconfidence and align predicted probabilities with the true BCPs, thereby improving the

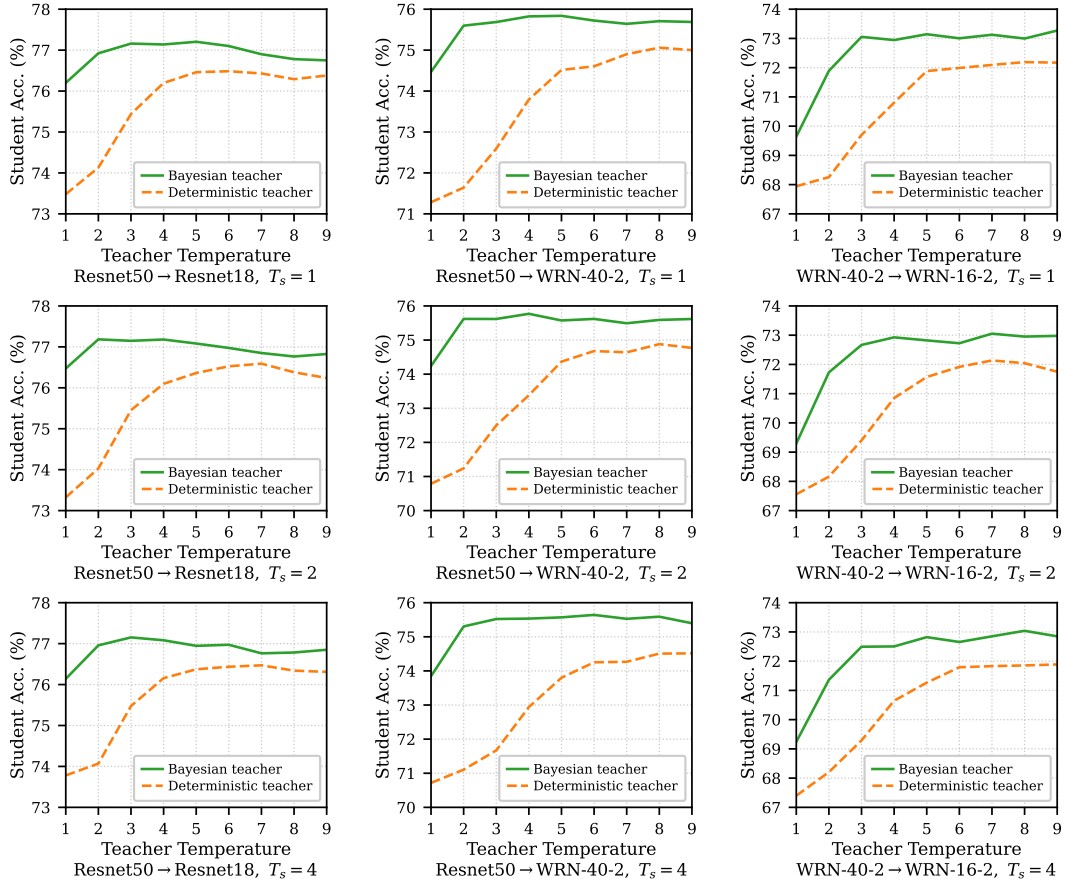

Figure 5: The accuracy achieved by students (averaged over 5 runs) trained from both deterministic and Bayesian teachers in KD with different teacher and student temperatures applied. The teacher-student pairs displayed are ResNet-50 → ResNet-18 (left), ResNet-50→WRN-40-2 (middle), and WRN-40-2→WRN-16-2 (right).

quality of supervision for distillation Kim et al. (2025). Here, we investigate the effects of temperature scaling in KD settings with both Bayesian (VI) and deterministic teachers, and compare them.

We consider three teacher–student pairs. For each pair, the student is trained with teacher temperatures ranging from $T_t = 1$ to $T_t = 9$, with three different student temperatures $T_s \in \{1, 2, 4\}$. Each experiment is repeated five times, and the average accuracies across runs are reported. All teacher and student trainings follow the implementation details and hyperparameters described in Appendix F, and results are displayed in Figure 5.

It can be seen that students trained with Bayesian teachers consistently achieve higher accuracy across all temperature combinations and architectures compared to students trained with deterministic teachers. In particular, under standard distillation without temperature scaling ($T_t = 1, T_s = 1$), the gains are substantial: student accuracy improves by up to 3.18% (ResNet-50→WRN-40-2), even though the Bayesian teacher itself is only 0.26% more accurate than the deterministic teacher.

Additionally, the effect of temperature scaling is consistently smaller for students trained from Bayesian teachers compared to those trained from deterministic teachers. For example, in the ResNet-50 → ResNet-18 case, the gap between the highest and lowest student accuracy across all temperature combinations is 3.32% when using deterministic teachers. On the other hand, for students distilled from Bayesian teachers this gap is only 1.07%, about three times smaller. This shows that Bayesian teachers produce probability estimates that are inherently better calibrated than those of deterministic teachers, making them more suitable for KD. Additionally, it shows that Bayesian teachers are less sensitive to hyperparameters and are easier to tune, further strengthening them as effective teachers in KD.

## I    BAYESIAN TEACHER PARAMETERS

BNNs trained with variational inference represent model parameters not as fixed values but as probability distributions Blundell et al. (2015); Jospin et al. (2022). This allows the model to capture uncertainty by considering different possible configurations of the weights, rather than a single point estimate Kabir et al. (2018). As a result, running the same input through the model may yield different predicted probabilities each time. To obtain reliable predictions, one typically performs Monte Carlo sampling at inference time, averaging outputs from several stochastic forward passes Gawlikowski et al. (2023). Increasing the number of samples generally improves the stability and accuracy of the predictive distribution of the teacher model.

We next investigate how the number of Monte Carlo samples used to compute the teacher's probability estimates for distillation affects the performance of the student model. To that aim, we trained two teacher–student pairs with varying numbers of samples used for Monte Carlo prediction averaging. For each setting, we report both the average accuracy of the teacher model and the average accuracy of the corresponding student model. Each experiment was repeated five times, and the average values across runs are reported. For reference, we also include the performance of the deterministic teacher and its student. Distillation in this experiment was carried out without temperature scaling and with $\lambda = 1$ (full distillation). In addition, to assess the effect of the number of Monte Carlo samples used on gradient noise, we report the average noise in the accuracy curves, in a similar fashion to the one conducted in Subsection 4.2 and shown in Figure 2. Both teacher and student training follow the same implementation details and hyperparameters described in Appendix F.

The results are displayed in Figure 6 and Figure 7. Teacher performance is in line with prior work on BNNs trained with VI; accuracy improves substantially as the number of Monte Carlo samples increases Shen et al. (2024). We observe gains of up to $6\%$ when moving from 1 to 12 samples, with most of the improvement being in the low-sample regime. Interestingly, the corresponding student accuracy acts differently. While adding more samples does not harm the student, its effect is modest. For VGG-13 $\rightarrow$ VGG-8, the student improves by only about $0.4\%$ when increasing from 1 to 7 samples, after which accuracy saturates, compared to the teacher's much larger improvement of about $5.5\%$. For VGG-13 $\rightarrow$ WRN-40-1, student accuracy roughly remains unchanged regardless of the number of samples used. This is explained by the fact that although a single realization is used, the fact that it is randomized anew on each sample leads to improved calibration when used for supervision in KD as it is averaged in multiple epochs of SGD. Specifically, the student is trained using the BNN teacher outputs throughout multiple epochs. This means that although training in each individual iteration is performed with a noisy BNN teacher sample, over multiple iterations training is performed with the expected value of the BNN teacher sample, effectively capturing the Monte Carlo sample average. The noise in this case refers to the noise introduced by the distributions on the weights of the Bayesian model and not the noise in the probability estimates.

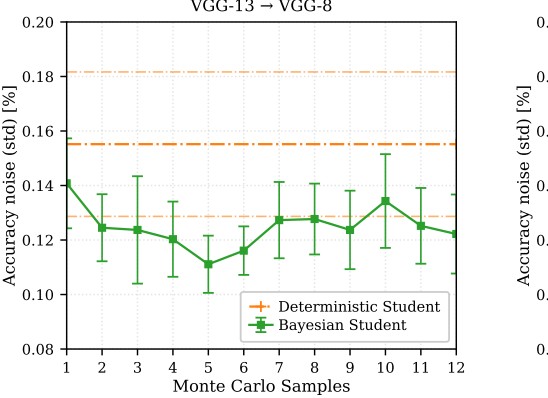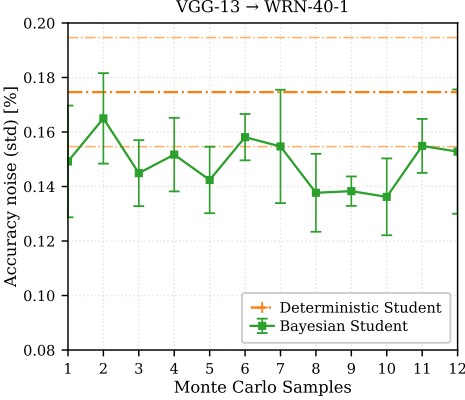

Figure 7: The average noise in the accuracy curves present during training for students trained from Bayesian teachers using a varying number of Monte Carlo samples to compute teacher predictions in the distillation process (averaged over 5 runs). Corresponding average noise in the accuracy curves present during training of the students trained from deterministic teachers are reported for reference. The teacher-student pairs displayed are VGG-13 $\rightarrow$ VGG-8 (left) and VGG-13$\rightarrow$WRN-40-1 (right).

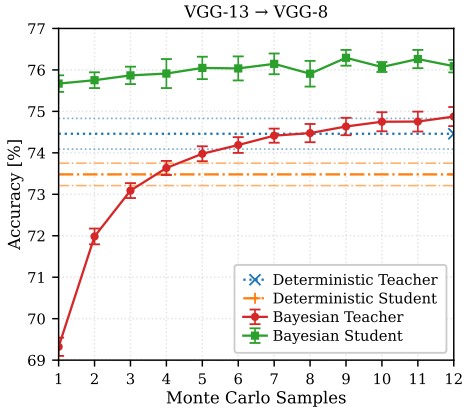 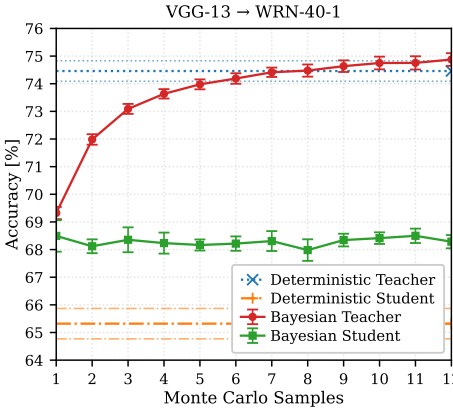

Figure 6: The accuracy achieved by Bayesian teachers, and their corresponding students (averaged over 5 runs). Teacher accuracies correspond to the average accuracy achieved while doing inference with a different number of Monte Carlo samples, and student accuracies correspond to the accuracy achieved while using that number of Monte Carlo samples to compute teacher predictions for distillation. Corresponding deterministic teacher and student accuracies are reported for reference. The teacher-student pairs displayed are VGG-13 → VGG-8 (left) and VGG-13→WRN-40-1 (right).

Moreover, for all cases, students distilled from Bayesian teachers show better performance compared to students distilled from deterministic teachers, regardless of the number of samples used. This shows that Bayesian teachers are better suited for KD compared to deterministic teachers, even when only a single sample is used. Finally, the number of samples used seems to have no noticeable effect on the noise present in the accuracy curves. Nevertheless, students trained from Bayesian teachers exhibit 10-30% less noise in the accuracy curves compared to students trained from deterministic teachers, regardless of the number of samples used.

## J   ADDITIONAL DISTILLATION METHODS & TINY IMAGENET

In this section, we present additional experimental results obtained using several response-based distillation methods, as well as experiments conducted on the Tiny ImageNet dataset. In this paper, we develop a theoretical understanding showing that teachers that better approximate the true BCPs yield improved student models. Building on this insight, we propose leveraging BNN teachers for standard response-based KD Hinton et al. (2015) to enhance student performance. While our method focuses on improving teacher predictions, many works instead improve response-based KD by modifying the distillation procedure itself. Typically, these methods manipulate teacher outputs, student outputs, or the distillation loss.

Here, we demonstrate that combining different response-based distillers with Bayesian NN teachers trained via VI leads to consistently better students compared to using deterministic teachers. The experiments are conducted on the CIFAR-100 dataset, and we follow the hyperparameter settings recommended in the original papers.

Huang et al. (2022) propose DIST, which includes a correlation-based loss to capture the intrinsic inter-class relations from the teacher, and extend the relational match to the intra-class level. Going by their notations, we use $\beta = 2$ and $\gamma = 2$ as suggested by them. For $\alpha$, the parameter balancing between one-hot training and KD, we use both $\alpha = 0$ and $\alpha = 1$ and report the student achieving the highest accuracy between the options.

Zhao et al. (2022) propose Decoupled KD (DKD), which decomposes the KD loss into target-class and non-target-class components to separately preserve sample-difficulty information and amplify the core benefits of response-based KD. Going by their notations, we use $\alpha = 1$, $\beta = 8$ for Resnet teachers, and $\beta = 6$ for WRN and VGG teachers. For the parameter balancing between one-hot training and KD, we use both 0 and 1 and report the student achieving the highest accuracy between the options.

Zheng & Yang (2024) propose weighted transformed teacher matching (WTTM), which augments transformed teacher matching with a sample-adaptive weighting scheme to better align the student with the teacher's power-transformed distribution. Going by their notations, we use $\gamma = 0.3, \beta = 1.5$ for Resnet teachers, $\gamma = 0.1, \beta = 2.25$ for VGG teachers, $\gamma = 0.1, \beta = 3$ for the WRN-40-2 teacher with the WRN-40-1 student, and $\gamma = 0.1, \beta = 4$ for the WRN-40-2 teacher with the WRN-16-2 student. We use 1 for the parameter balancing between one-hot training and KD.

The deterministic teacher, Bayesian VI teacher, and student training procedures match those used in the main experiments (Table 1) and are detailed in Appendix F. Additionally, for each teacher–student pair we train students under multiple teacher temperatures ($T_t \in \{1, 2, 4\}$). For each pair, reported values correspond to the teacher temperature in which the best average student accuracy over five trials was achieved. The results, provided in Table 3, show that using BNN teachers consistently improves student performance across all three methods compared to using deterministic NN teachers. These findings demonstrate that the advantages of BNN teachers in KD extend beyond the standard distillation loss and generalize to additional response-based distillation methods.

Table 3: Test accuracy (%) of student networks on the CIFAR-100 dataset, averaged over 5 runs. Both the teachers and the students were trained with varying random seeds between runs. Results are reported for teacher-student pairs with both matching and different architectures. The subscript in the Bayesian columns denotes changes relative to the corresponding deterministic distillation method. Deter. stands for Deterministic. Bayesian VI and Deterministic teacher accuracies are reported in Table 1.

| Teachers and students with matching architectures | | | | | | | | | | | | |
|---|---|---|---|---|---|---|---|---|---|---|---|
| **Teacher** | ResNet-18 | | ResNet-50 | | ResNet-50 | | WRN-40-2 | | WRN-40-2 | | VGG-13 | |
| **Student** | ResNet-18 | | ResNet-18 | | ResNet-34 | | WRN-16-2 | | WRN-40-1 | | VGG-8 | |
| **Student acc. (no KD)** | 73.23 | | 73.23 | | 73.61 | | 67.70 | | 65.03 | | 73.52 | |
| **Teacher type** | Deter. | Bayesian | Deter. | Bayesian | Deter. | Bayesian | Deter. | Bayesian | Deter. | Bayesian | Deter. | Bayesian |
| KD Hinton et al. (2015) | 75.92 | 76.92 +1.00 | 76.26 | 77.27 +1.01 | 76.82 | 77.63 +0.81 | 70.80 | 72.94 +2.14 | 68.92 | 70.77 +1.85 | 76.08 | 77.61 +1.53 |
| DIST Huang et al. (2022) | 75.01 | 76.93 +1.92 | 75.39 | 77.19 +1.80 | 74.52 | 75.23 +0.71 | 70.84 | 72.53 +1.69 | 68.92 | 70.69 +1.77 | 75.43 | 77.10 +1.67 |
| DKD Zhao et al. (2022) | 74.86 | 77.14 +2.28 | 75.75 | 77.21 +1.46 | 76.10 | 77.29 +1.19 | 71.04 | 73.32 +2.28 | 69.77 | 71.68 +1.91 | 75.41 | 77.85 +2.44 |
| WTTM Zheng & Yang (2024) | 76.05 | 76.89 +0.84 | 77.11 | 77.28 +0.17 | 76.54 | 77.23 +0.69 | 72.16 | 73.54 +1.38 | 70.78 | 70.82 +0.04 | 76.81 | 77.64 +0.83 |
| **Teachers and students with different architectures** | | | | | | | | | | | | |
| **Teacher** | ResNet-50 | | ResNet-50 | | VGG-13 | | VGG-13 | | ResNet-50 | | VGG-13 | |
| **Student** | WRN-40-2 | | VGG-8 | | ResNet-18 | | WRN-40-1 | | WRN-16-2 | | WRN-40-2 | |
| **Student acc. (no KD)** | 70.76 | | 73.52 | | 73.23 | | 65.03 | | 67.70 | | 70.76 | |
| **Teacher type** | Deter. | Bayesian | Deter. | Bayesian | Deter. | Bayesian | Deter. | Bayesian | Deter. | Bayesian | Deter. | Bayesian |
| KD Hinton et al. (2015) | 73.79 | 75.82 +2.03 | 75.66 | 77.27 +1.61 | 76.36 | 77.41 +1.05 | 67.90 | 71.37 +3.47 | 69.36 | 73.63 +4.27 | 73.71 | 75.45 +1.74 |
| DIST Huang et al. (2022) | 73.40 | 76.04 +2.64 | 75.42 | 76.88 +1.46 | 75.32 | 77.34 +2.02 | 68.61 | 71.81 +3.20 | 70.45 | 73.03 +2.58 | 73.41 | 76.10 +2.69 |
| DKD Zhao et al. (2022) | 74.11 | 75.90 +1.79 | 75.78 | 77.79 +2.01 | 75.68 | 77.61 +1.93 | 69.78 | 71.86 +2.08 | 70.82 | 74.10 +3.28 | 74.02 | 75.66 +1.64 |
| WTTM Zheng & Yang (2024) | 75.34 | 75.39 +0.05 | 77.40 | 77.94 +0.54 | 75.68 | 77.43 +1.75 | 71.71 | 70.12 -1.59 | 73.82 | 73.86 +0.04 | 74.78 | 74.71 -0.07 |

Next, we present results evaluating our approach on the Tiny ImageNet dataset, which is a smaller subset of the larger ImageNet dataset. Tiny ImageNet contains 110,000 images of size $64 \times 64$ across 200 classes, split into training and test sets with a 10:1 ratio. Compared to CIFAR-100, both datasets contain 500 examples per class, but Tiny ImageNet has double the number of classes, making the classification task more challenging. In this set of experiments, we compare student models distilled from either a deterministic NN teacher or a Bayesian NN teacher trained via VI.

The teacher and student training procedures follow those described in the main experiments (Table 1) and detailed in Appendix F, with two modifications: training is performed for 120 epochs, and the learning rate is decayed by a factor of 0.1 at epoch 60. For each teacher–student pair, we conduct five independent trials under six different $(\lambda, T_t, T_s)$ configurations, following the setup used in Appendix F. For each pair, reported values correspond to the $(\lambda, T_t, T_s)$ combination in which the best average student accuracy over the five trials was achieved. The results are provided in Table 4. Consistent with the results on CIFAR-100, students distilled from Bayesian NN teachers achieve higher accuracy than those trained using deterministic NN teachers.

Table 4: Top-1 and Top-5 accuracy (%) on Tiny ImageNet for different teacher-student pairs, averaged over 5 runs. Both the teachers and the students were trained with varying random seeds between runs.

| Teacher-Student | Deterministic Teacher | | Bayesian Teacher | | Student (No KD) | | Student from Deterministic Teacher | | Student from Bayesian Teacher | |
|---|---|---|---|---|---|---|---|---|---|---|
| | Top 1 | Top 5 | Top 1 | Top 5 | Top 1 | Top 5 | Top 1 | Top 5 | Top 1 | Top 5 |
| Resnet-34→Resnet-18 | 59.76 | 80.85 | 60.68 | 81.73 | 58.47 | 80.24 | 62.40 | 82.71 | 64.77 | 84.63 |
| Resnet-34→WRN-16-2 | 59.76 | 80.85 | 60.68 | 81.73 | 52.17 | 77.23 | 55.62 | 79.13 | 56.83 | 80.46 |

## K ACCURACY VARIANCES AND FULL RESULTS

Table 5: Mean and variance of the accuracies corresponding to the numerical study in Subsection 4.2 performed with distillation hyperparameters $(\lambda, T_t, T_s) = (1, 1, 1)$ (averaged over 5 runs).

| Teacher Student | ResNet-18 ResNet-18 | ResNet-50 ResNet-18 | ResNet-50 ResNet-34 | WRN-40-2 WRN-16-2 | WRN-40-2 WRN-40-1 | VGG-13 VGG-8 |
|---|---|---|---|---|---|---|
| Deterministic | $73.32 \pm 0.21$ | $73.48 \pm 0.21$ | $73.89 \pm 0.36$ | $67.94 \pm 0.33$ | $65.07 \pm 0.60$ | $73.48 \pm 0.27$ |
| Bayesian | $76.28 \pm 0.26$ | $76.20 \pm 0.27$ | $76.76 \pm 0.29$ | $69.64 \pm 0.16$ | $67.55 \pm 0.39$ | $76.24 \pm 0.15$ |
| Laplace | $74.88 \pm 0.40$ | $75.80 \pm 0.20$ | $75.56 \pm 0.38$ | $70.45 \pm 0.41$ | $68.87 \pm 0.42$ | $75.79 \pm 0.31$ |
| MCMI | $73.61 \pm 0.28$ | $73.61 \pm 0.27$ | $73.85 \pm 0.22$ | $68.20 \pm 0.18$ | $65.27 \pm 0.39$ | $73.36 \pm 0.41$ |
| MSE | $74.84 \pm 0.45$ | $74.84 \pm 0.28$ | $73.41 \pm 1.78$ | $67.74 \pm 0.31$ | $54.30 \pm 0.54$ | $73.25 \pm 0.30$ |

| Teacher Student | ResNet-50 WRN-40-2 | ResNet-50 VGG-8 | VGG-13 ResNet-18 | VGG-13 WRN-40-1 | ResNet-50 WRN-16-2 | VGG-13 WRN-40-2 |
|---|---|---|---|---|---|---|
| Deterministic | $71.28 \pm 0.38$ | $73.41 \pm 0.30$ | $73.44 \pm 0.21$ | $65.32 \pm 0.55$ | $67.70 \pm 0.34$ | $71.12 \pm 0.37$ |
| Bayesian | $74.47 \pm 0.17$ | $75.56 \pm 0.10$ | $76.48 \pm 0.22$ | $68.37 \pm 0.40$ | $70.84 \pm 0.30$ | $74.26 \pm 0.17$ |
| Laplace | $74.57 \pm 0.41$ | $75.97 \pm 0.23$ | $75.84 \pm 0.48$ | $70.53 \pm 0.33$ | $72.52 \pm 0.45$ | $74.39 \pm 0.36$ |
| MCMI | $71.17 \pm 0.47$ | $73.34 \pm 0.27$ | $73.28 \pm 0.21$ | $65.33 \pm 0.27$ | $67.99 \pm 0.23$ | $71.14 \pm 0.17$ |
| MSE | $70.71 \pm 0.24$ | $73.44 \pm 0.32$ | $74.65 \pm 0.11$ | $53.56 \pm 0.73$ | $67.42 \pm 0.18$ | $70.64 \pm 0.53$ |

Table 6: Mean and variance of the accuracies corresponding to the numerical study in Subsection 4.2 performed with distillation hyperparameters $(\lambda, T_t, T_s) = (0.474, 4, 4)$ (averaged over 5 runs).

| Teacher Student | ResNet-18 ResNet-18 | ResNet-50 ResNet-18 | ResNet-50 ResNet-34 | WRN-40-2 WRN-16-2 | WRN-40-2 WRN-40-1 | VGG-13 VGG-8 |
|---|---|---|---|---|---|---|
| Deterministic | $75.92 \pm 0.24$ | $76.05 \pm 0.40$ | $76.76 \pm 0.25$ | $70.38 \pm 0.27$ | $68.06 \pm 0.65$ | $75.62 \pm 0.15$ |
| Bayesian | $76.84 \pm 0.17$ | $77.27 \pm 0.27$ | $77.54 \pm 0.10$ | $72.42 \pm 0.47$ | $70.14 \pm 0.27$ | $77.51 \pm 0.19$ |
| Laplace | $76.30 \pm 0.25$ | $76.69 \pm 0.34$ | $76.23 \pm 0.12$ | $71.66 \pm 0.38$ | $69.65 \pm 0.29$ | $76.09 \pm 0.26$ |
| MCMI | $75.83 \pm 0.35$ | $76.35 \pm 0.20$ | $76.55 \pm 0.40$ | $70.40 \pm 0.61$ | $68.10 \pm 0.20$ | $75.66 \pm 0.53$ |
| MSE | $74.48 \pm 0.12$ | $74.46 \pm 0.23$ | $75.58 \pm 0.22$ | $68.62 \pm 0.44$ | $65.42 \pm 0.39$ | $73.86 \pm 0.34$ |

| Teacher Student | ResNet-50 WRN-40-2 | ResNet-50 VGG-8 | VGG-13 ResNet-18 | VGG-13 WRN-40-1 | ResNet-50 WRN-16-2 | VGG-13 WRN-40-2 |
|---|---|---|---|---|---|---|
| Deterministic | $72.73 \pm 0.21$ | $75.56 \pm 0.21$ | $76.33 \pm 0.27$ | $66.98 \pm 0.38$ | $68.64 \pm 0.40$ | $73.13 \pm 0.26$ |
| Bayesian | $75.35 \pm 0.27$ | $77.27 \pm 0.46$ | $77.41 \pm 0.20$ | $70.64 \pm 0.28$ | $73.19 \pm 0.38$ | $75.37 \pm 0.23$ |
| Laplace | $72.84 \pm 0.05$ | $76.11 \pm 0.35$ | $76.42 \pm 0.10$ | $68.02 \pm 0.26$ | $70.69 \pm 0.36$ | $72.95 \pm 0.31$ |
| MCMI | $72.44 \pm 0.34$ | $75.27 \pm 0.29$ | $76.31 \pm 0.14$ | $67.10 \pm 0.51$ | $68.47 \pm 0.41$ | $73.23 \pm 0.10$ |
| MSE | $71.09 \pm 0.27$ | $74.09 \pm 0.20$ | $74.48 \pm 0.26$ | $65.61 \pm 0.07$ | $68.55 \pm 0.26$ | $71.39 \pm 0.35$ |

Table 7: Mean and variance of the accuracies corresponding to the numerical study in Subsection 4.2 performed with distillation hyperparameters $(\lambda, T_t, T_s) = (1, 2, 1)$ (averaged over 5 runs).

| Teacher
Student | ResNet-18
ResNet-18 | ResNet-50
ResNet-18 | ResNet-50
ResNet-34 | WRN-40-2
WRN-16-2 | WRN-40-2
WRN-40-1 | VGG-13
VGG-8 |
|---|---|---|---|---|---|---|
| Deterministic | $74.27 \pm 0.26$ | $74.13 \pm 0.07$ | $74.66 \pm 0.21$ | $68.26 \pm 0.34$ | $65.80 \pm 0.49$ | $74.50 \pm 0.21$ |
| Bayesian | $76.81 \pm 0.26$ | $76.92 \pm 0.23$ | $77.30 \pm 0.04$ | $71.89 \pm 0.60$ | $69.75 \pm 0.26$ | $77.33 \pm 0.15$ |
| Laplace | $74.14 \pm 0.26$ | $75.07 \pm 0.40$ | $75.19 \pm 0.29$ | $71.79 \pm 0.38$ | $69.96 \pm 0.28$ | $75.26 \pm 0.28$ |
| MCMI | $74.22 \pm 0.21$ | $74.05 \pm 0.08$ | $74.78 \pm 0.19$ | $68.10 \pm 0.38$ | $65.86 \pm 0.24$ | $74.39 \pm 0.18$ |
| MSE | $74.19 \pm 0.35$ | $74.22 \pm 0.34$ | $73.70 \pm 2.00$ | $67.73 \pm 0.33$ | $49.99 \pm 0.55$ | $72.61 \pm 1.20$ |

| Teacher
Student | ResNet-50
WRN-40-2 | ResNet-50
VGG-8 | VGG-13
ResNet-18 | VGG-13
WRN-40-1 | ResNet-50
WRN-16-2 | VGG-13
WRN-40-2 |
|---|---|---|---|---|---|---|
| Deterministic | $71.64 \pm 0.32$ | $74.16 \pm 0.27$ | $74.62 \pm 0.35$ | $65.41 \pm 0.39$ | $68.02 \pm 0.18$ | $71.56 \pm 0.26$ |
| Bayesian | $75.60 \pm 0.21$ | $76.92 \pm 0.30$ | $77.37 \pm 0.18$ | $70.75 \pm 0.17$ | $72.78 \pm 0.25$ | $75.29 \pm 0.31$ |
| Laplace | $74.24 \pm 0.22$ | $75.53 \pm 0.29$ | $75.00 \pm 0.40$ | $70.87 \pm 0.44$ | $71.89 \pm 0.25$ | $74.18 \pm 0.30$ |
| MCMI | $71.86 \pm 0.38$ | $73.95 \pm 0.29$ | $74.54 \pm 0.13$ | $65.70 \pm 0.37$ | $67.82 \pm 0.25$ | $71.92 \pm 0.17$ |
| MSE | $69.81 \pm 0.33$ | $72.61 \pm 1.00$ | $74.55 \pm 0.24$ | $49.86 \pm 0.30$ | $67.99 \pm 0.67$ | $70.06 \pm 0.33$ |

Table 8: Mean and variance of the accuracies corresponding to the numerical study in Subsection 4.2 performed with distillation hyperparameters $(\lambda, T_t, T_s) = (1, 4, 1)$ (averaged over 5 runs).

| Teacher
Student | ResNet-18
ResNet-18 | ResNet-50
ResNet-18 | ResNet-50
ResNet-34 | WRN-40-2
WRN-16-2 | WRN-40-2
WRN-40-1 | VGG-13
VGG-8 |
|---|---|---|---|---|---|---|
| Deterministic | $75.69 \pm 0.29$ | $76.20 \pm 0.16$ | $76.78 \pm 0.35$ | $70.80 \pm 0.46$ | $68.92 \pm 0.42$ | $76.08 \pm 0.25$ |
| Bayesian | $76.83 \pm 0.27$ | $77.14 \pm 0.35$ | $77.25 \pm 0.16$ | $72.94 \pm 0.19$ | $70.77 \pm 0.49$ | $77.61 \pm 0.20$ |
| Laplace | $73.77 \pm 0.42$ | $74.68 \pm 0.41$ | $74.91 \pm 0.68$ | $71.01 \pm 0.27$ | $70.04 \pm 0.33$ | $74.92 \pm 0.25$ |
| MCMI | $75.66 \pm 0.32$ | $76.26 \pm 0.29$ | $76.86 \pm 0.30$ | $70.87 \pm 0.52$ | $68.66 \pm 0.54$ | $76.35 \pm 0.14$ |
| MSE | $74.03 \pm 0.13$ | $73.88 \pm 0.14$ | $73.98 \pm 0.27$ | $67.73 \pm 0.41$ | $49.30 \pm 0.21$ | $72.88 \pm 0.06$ |

| Teacher
Student | ResNet-50
WRN-40-2 | ResNet-50
VGG-8 | VGG-13
ResNet-18 | VGG-13
WRN-40-1 | ResNet-50
WRN-16-2 | VGG-13
WRN-40-2 |
|---|---|---|---|---|---|---|
| Deterministic | $73.79 \pm 0.33$ | $75.66 \pm 0.39$ | $76.36 \pm 0.22$ | $67.90 \pm 0.30$ | $69.36 \pm 0.38$ | $73.71 \pm 0.20$ |
| Bayesian | $75.82 \pm 0.17$ | $77.27 \pm 0.14$ | $77.26 \pm 0.20$ | $71.37 \pm 0.23$ | $73.63 \pm 0.36$ | $75.45 \pm 0.31$ |
| Laplace | $73.97 \pm 0.33$ | $74.96 \pm 0.32$ | $74.57 \pm 0.22$ | $69.83 \pm 0.30$ | $71.30 \pm 0.21$ | $73.76 \pm 0.44$ |
| MCMI | $73.50 \pm 0.16$ | $75.75 \pm 0.13$ | $76.40 \pm 0.34$ | $67.48 \pm 0.50$ | $69.58 \pm 0.27$ | $74.20 \pm 0.27$ |
| MSE | $69.59 \pm 0.49$ | $72.59 \pm 0.49$ | $73.89 \pm 0.29$ | $49.39 \pm 0.33$ | $67.88 \pm 0.43$ | $70.11 \pm 0.54$ |

Table 9: Mean and variance of the accuracies corresponding to the numerical study in Subsection 4.2 performed with distillation hyperparameters $(\lambda, T_t, T_s) = (1, 2, 2)$ (averaged over 5 runs).

| Teacher
Student | ResNet-18
ResNet-18 | ResNet-50
ResNet-18 | ResNet-50
ResNet-34 | WRN-40-2
WRN-16-2 | WRN-40-2
WRN-40-1 | VGG-13
VGG-8 |
|---|---|---|---|---|---|---|
| Deterministic | $74.36 \pm 0.12$ | $74.23 \pm 0.40$ | $74.58 \pm 0.14$ | $68.16 \pm 0.45$ | $65.57 \pm 0.40$ | $74.16 \pm 0.22$ |
| Bayesian | $76.92 \pm 0.27$ | $77.18 \pm 0.28$ | $77.63 \pm 0.10$ | $71.73 \pm 0.16$ | $69.73 \pm 0.19$ | $77.16 \pm 0.24$ |
| Laplace | $74.10 \pm 0.30$ | $75.26 \pm 0.34$ | $75.20 \pm 0.50$ | $71.80 \pm 0.40$ | $69.83 \pm 0.21$ | $75.51 \pm 0.34$ |
| MCMI | $74.14 \pm 0.34$ | $74.16 \pm 0.41$ | $74.59 \pm 0.22$ | $68.25 \pm 0.46$ | $65.70 \pm 0.28$ | $74.16 \pm 0.34$ |
| MSE | $74.65 \pm 0.21$ | $74.89 \pm 0.24$ | $73.34 \pm 1.71$ | $67.78 \pm 0.63$ | $50.02 \pm 0.50$ | $73.09 \pm 0.32$ |

| Teacher
Student | ResNet-50
WRN-40-2 | ResNet-50
VGG-8 | VGG-13
ResNet-18 | VGG-13
WRN-40-1 | ResNet-50
WRN-16-2 | VGG-13
WRN-40-2 |
|---|---|---|---|---|---|---|
| Deterministic | $71.24 \pm 0.35$ | $73.55 \pm 0.32$ | $74.66 \pm 0.32$ | $65.45 \pm 0.44$ | $67.55 \pm 0.47$ | $71.66 \pm 0.22$ |
| Bayesian | $75.62 \pm 0.19$ | $77.01 \pm 0.25$ | $77.41 \pm 0.10$ | $70.58 \pm 0.16$ | $72.94 \pm 0.35$ | $75.43 \pm 0.30$ |
| Laplace | $74.36 \pm 0.27$ | $75.49 \pm 0.39$ | $75.27 \pm 0.43$ | $70.63 \pm 0.23$ | $71.96 \pm 0.44$ | $74.09 \pm 0.38$ |
| MCMI | $70.87 \pm 0.37$ | $73.75 \pm 0.15$ | $74.54 \pm 0.18$ | $65.33 \pm 0.43$ | $67.81 \pm 0.31$ | $71.58 \pm 0.40$ |
| MSE | $70.24 \pm 0.13$ | $73.25 \pm 0.28$ | $74.74 \pm 0.39$ | $50.17 \pm 0.12$ | $67.58 \pm 0.44$ | $70.22 \pm 0.51$ |

Table 10: Mean and variance of the accuracies corresponding to the numerical study in Subsection 4.2 performed with distillation hyperparameters $(\lambda, T_t, T_s) = (1, 4, 4)$ (averaged over 5 runs).

| **Teacher** **Student** | ResNet-18 ResNet-18 | ResNet-50 ResNet-18 | ResNet-50 ResNet-34 | WRN-40-2 WRN-16-2 | WRN-40-2 WRN-40-1 | VGG-13 VGG-8 |
|---|---|---|---|---|---|---|
| Deterministic | $75.91 \pm 0.32$ | $76.26 \pm 0.22$ | $76.82 \pm 0.27$ | $70.65 \pm 0.14$ | $68.70 \pm 0.18$ | $75.74 \pm 0.34$ |
| Bayesian | $76.77 \pm 0.23$ | $77.08 \pm 0.23$ | $77.18 \pm 0.14$ | $72.50 \pm 0.27$ | $70.29 \pm 0.24$ | $77.30 \pm 0.13$ |
| Laplace | $73.78 \pm 0.45$ | $74.83 \pm 0.09$ | $75.03 \pm 0.56$ | $70.61 \pm 0.20$ | $69.66 \pm 0.17$ | $74.90 \pm 0.41$ |
| MCMI | $75.86 \pm 0.33$ | $76.44 \pm 0.21$ | $76.68 \pm 0.23$ | $70.64 \pm 0.13$ | $68.33 \pm 0.14$ | $75.64 \pm 0.26$ |
| MSE | $75.01 \pm 0.20$ | $74.75 \pm 0.36$ | $74.46 \pm 0.47$ | $67.41 \pm 0.46$ | $49.62 \pm 0.24$ | $72.97 \pm 0.29$ |

| **Teacher** **Student** | ResNet-50 WRN-40-2 | ResNet-50 VGG-8 | VGG-13 ResNet-18 | VGG-13 WRN-40-1 | ResNet-50 WRN-16-2 | VGG-13 WRN-40-2 |
|---|---|---|---|---|---|---|
| Deterministic | $72.94 \pm 0.31$ | $75.43 \pm 0.23$ | $76.22 \pm 0.28$ | $66.99 \pm 0.35$ | $68.71 \pm 0.37$ | $73.29 \pm 0.33$ |
| Bayesian | $75.53 \pm 0.09$ | $77.06 \pm 0.24$ | $76.95 \pm 0.21$ | $70.92 \pm 0.37$ | $73.14 \pm 0.23$ | $75.18 \pm 0.33$ |
| Laplace | $73.88 \pm 0.35$ | $75.10 \pm 0.31$ | $74.96 \pm 0.27$ | $69.79 \pm 0.28$ | $71.14 \pm 0.38$ | $73.79 \pm 0.17$ |
| MCMI | $72.68 \pm 0.37$ | $75.45 \pm 0.15$ | $76.40 \pm 0.23$ | $67.14 \pm 0.22$ | $68.45 \pm 0.40$ | $73.09 \pm 0.38$ |
| MSE | $69.74 \pm 0.44$ | $73.05 \pm 0.33$ | $74.87 \pm 0.21$ | $49.65 \pm 0.50$ | $67.25 \pm 0.24$ | $69.90 \pm 0.36$ |

## L  LIMITATIONS

• Our theoretical results in Section 3 rely on several assumptions commonly used in optimization literature, such as strong quasi-convexity or the Polyak–Łojasiewicz condition. We also assume model expressiveness to ensure convergence to an optimum. These assumptions, while standard, can of course be questioned.

• To make our claims rigorous, we modeled noisy BCPs as either perfect BCPs with additive noise, or as Dirichlet-distributed BCPs. One can always question this type of modeling and suggested more complex modeling, such as incorporating bias or correlation between samples.

• Our work advocates the use of BNNs as teachers in KD, possibly obtained by converting a pre-trained deterministic teacher model into a BNN using LA. Still, one has to account for the additional computational complexity of BNNs.

