# OpenReview forum: "SGD-Based Knowledge Distillation with Bayesian Teachers: Theory and Guidelines"
_ICLR.cc/2026/Conference — ICLR 2026 Poster_

### Official Review · Reviewer_Faxx · 2025-10-28

**Soundness:** 3
**Presentation:** 3
**Contribution:** 2
**Rating:** 4
**Confidence:** 3

**Summary:**

The paper carries out convergence analysis for SGD-based KD with Bayesian teacher and noisy Bayesian teacher, showing faster and more stable convergence compared to standard SGD. Based on the above analysis, the paper proposes to use BNNs as Bayesian teachers, either trained from scratch or converted from deterministic pretrained models. Experimental results are provided to validate the theoretical analysis and some performance gain is demonstrated.

**Strengths:**

1. The paper studies the benefit of Bayesian teachers in KD from the perspective of SGD convergence, which is a solid and principled choice.
2. The paper connects the literature on BNNs to KD through the use of Bayesian teachers.

**Weaknesses:**

1. The paper lacks high novelty. The Bayesian teacher perspective of KD has been well-studied in the literature and many results, both theoretical and empirical, have been presented to show that Bayesian teacher is optimal for student learning. The real important issue is how to pratically obtain a Bayesian teacher. However, the paper doesn't emphasize much on this issue, and simply resorts to some existing Bayesian DL methods.
2. The experimental results are not comprehensive enough. (1) All results are based on the standard KD, without extending to any latest logit-based distillers. (2) No result on ImageNet is presented. (3) No result for transformer-based models, e.g., ViT, is presented.

**Questions:**

1. For converting a deterministic pretrained model into a BNN, is there any way to introduce stochasticity without modifying the teacher model itself (since in many cases, it's not desirable/feasible to modify the teacher model)? For example, through the use of data augmentation or adding auxiliary probabilistic modules to the teacher model.
2. The paper refers probability distributions that are closer to the BCPs as more "calibrated" probability distributions. Then, why not show some results on model calibration such as expected calibration error (ECE) and reliability diagram [1].

[1] C. Guo et al. On Calibration of Modern Neural Networks. ICML 2017.

---

> ### Author Response · Authors · 2025-11-20
> **Response part (1/2)**
>
> We thank the reviewer for their thoughtful and constructive comments. We respond to the concerns and suggestions raised as follows.
>
> **W1.** We would like to clarify that the main novelty of this work is not in proposing a Bayesian framework of knowledge distillation (KD), but in providing a new theoretical understanding of SGD-based learning under supervision with exact and noisy Bayes class probabilities (BCPs). As the reviewer pointed out, the work by Menon et al. [1] was the first to propose a Bayesian perspective to analyze KD. Nevertheless, to the best of our knowledge, no prior work analyzes how BCP-based supervision alters the optimization dynamics of SGD, nor derives the corresponding convergence bounds, variance reduction behavior, and interpolation properties. Existing works that adopt a Bayesian perspective on KD focus primarily on statistical aspects (e.g., excess risk, calibration) rather than on the convergence behavior of learning algorithms. Our results therefore introduce a substantially different viewpoint on why and when probabilistic teachers benefit student learning.
>
> Importantly, our work is also the first to advocate Bayesian deep-learning teachers from the perspective of the Bayesian view of KD, and to show that the theoretical predictions (improved performance and more stable convergence arising from better BCP approximation) hold in practice when using Bayesian NNs. While the focus of our work is not in developing new types of Bayesian deep learning, we demonstrate that our rigorous connection between calibration and reliable SGD-based teaching indeed leads to enhanced KD using existing Bayesian deep learning frameworks, and particularly with Variational Inference BNNs and with Bayesianized deterministic networks obtained via the Laplace approximation. We hope the reviewer recognizes that the novelty of the paper lies in its theoretical analysis of SGD under probabilistic supervision, together with the demonstration that Bayesian deep-learning teachers concretely realize the benefits predicted by this theory and can therefore serve as stronger and more reliable teachers than deterministic networks for practitioners performing KD. We strongly believe that our theoretical findings are relevant and fill a concrete gap in current understanding of how and why response-based KD is effective in classification tasks.
>
> **W2.** We thank the reviewer for these suggestions. We emphasize that our contributions are in the analytical characterization of the learning dynamics of KD with SGD based on a Bayesian perspective, as well as the unveiling of the usefulness of Bayesian NNs in KD. These contributions are both novel to our work and are fully supported by our experimental study, which spans both the numerical results reported in the main body, as well as those reported in the appendices.
>
> Specifically, we make the observation that Bayesian NNs, which are known in the literature to provide improved calibration, are naturally suitable for approximating the true BCPs. Our empirical results validate that such teachers, based either on variational inference or on applying the Laplace approximation to a pre-trained deterministic teacher, lead to better-performing student models and exhibit more stable learning curves, in line with our theoretical results.
>
> Our analysis is specific to response-based distillation, where the supervisory signal is a probability vector. Many recent response-based or feature-based distillers incorporate additional objectives (feature matching, relational, contrastive losses, etc.) that are not grounded in probabilistic classification. In such cases, the improvement gained from a better BCP approximation would likely be diluted by the presence of other losses, resulting in marginal overall gains. Performing experiments with improved distillation methods is an interesting direction, but would not directly validate our theoretical contributions and would likely lead to only minor gains. Moreover, we already include comparisons with alternative methods for enhancing KD (e.g., MCMI and MSE teachers).
>
> We hope the reviewer recognizes that our contributions are primarily theoretical, but also algorithmic, and that the experimental study we provide is both supportive and indicative of the insights drawn from our analytical findings, and is well aligned with the scope and claims of the paper.
>
> **Q1.**
> We thank the reviewer for this thoughtful question. Our theoretical analysis is agnostic to the specific mechanism by which Bayes class probability (BCP) estimates are obtained, and therefore applies to a wide range of teacher constructions that yield calibrated or noisy probability estimates. In our empirical study, we demonstrate two representative approaches for improving calibration: $(i)$ training a Bayesian neural network via variational inference; and $(ii)$ converting a pretrained deterministic network using a Laplace approximation.

---

> > ### Author Response · Authors · 2025-11-20
> > **Response part (2/2)**
> >
> > Importantly, the literature includes additional techniques for enhancing calibration without modifying teacher weights, such as test-time data augmentation (TTDA), as suggested by the reviewer (e.g., [6, Sec. 3.4]). Following the reviewer’s suggestion, we adapted TTDA as a drop-in method for generating stochastic teacher predictions for KD. We incorporated this approach as an additional baseline in the main experiments and added the corresponding results to the revised manuscript.
> >
> > Specifically, the teacher predictions were obtained by averaging the output probabilities over $N=10$ stochastic forward passes, each computed on a differently augmented version of the same input. For each pass, we applied a standard TTDA pipeline consisting of random cropping and random rotation. The resulting probability vectors were averaged to form the final teacher prediction used in the distillation process. We performed 5 independent teacher-student trials, the results are presented:
> >
> > |    | Teacher architecture | Student architecture | TTDA student accuracy (\%) | Δ TTDA (\%) | Δ LA (\%) | Δ VI (\%) |
> > |----|----------------------|----------------------|------------------|----------------------|----------|----------|
> > | 1 |ResNet-18|ResNet-18|76.21 ± 0.21|0.29|0.38|1.00|
> > | 2 |ResNet-50|ResNet-18|76.30 ± 0.31|0.04|0.43|1.01|
> > | 3 |ResNet-50|ResNet-34|76.83 ± 0.27|0.01|-0.59|0.81|
> > | 4 |WRN-40-2|WRN-16-2|70.75 ± 0.24|-0.05|0.99|2.14|
> > | 5 |WRN-40-2|WRN-40-1|69.05 ± 0.32|0.13|1.12|1.85|
> > | 6 |VGG-13|VGG-8|76.61 ± 0.21|0.53|0.01|1.53|
> > | 7 |ResNet-50|WRN-40-2|73.76 ± 0.21|-0.03|0.77|2.03|
> > | 8 |ResNet-50|VGG-8|76.00 ± 0.24|0.34|0.46|1.62|
> > | 9 |VGG-13|ResNet-18|76.80 ± 0.11|0.44|0.06|1.04|
> > | 10 |VGG-13|WRN-40-1|67.53 ± 0.33| -0.37|2.97|3.47|
> > | 11 |ResNet-50|WRN-16-2|69.24 ± 0.42|-0.12|3.16|4.27|
> > | 12 |VGG-13|WRN-40-2|74.05 ± 0.23|0.34|0.49|1.75|
> >
> > The fourth column shows the accuracy of the student trained using the TTDA teacher, and the next three columns show the differences between students trained with each corresponding method and the student trained with the deterministic teacher. As shown in our results, the TTDA teachers improve student performance in most settings, although in a few cases the students underperform compared to students trained with deterministic teachers. Notably, students trained with TTDA teachers occasionally even surpass the performance of students trained with Bayesian Laplace approximation teachers. Nevertheless, students trained with Bayesian variational inference teachers consistently achieve the highest accuracies.
> >
> > We thank the reviewer for this valuable suggestion and are pleased to incorporate this additional baseline and its corresponding results into the revised version of our paper.
> >
> > **Q2.** We thank the reviewer for the suggestion. We emphasize that our contribution is not in demonstrating that Bayesian deep learning methods yield better calibrated predictive distributions. This property has already been extensively studied and established in the Bayesian deep learning literature [4-6]. Instead, our work builds upon these well-established findings and leverages them to advocate the use of Bayesian teachers from a theoretical perspective: we show that improved BCP approximation directly translates into more stable learning and improved student performance.
> >
> > [1] Menon, Aditya K., et al. "A statistical perspective on distillation." International Conference on Machine Learning. PMLR, 2021.
> >
> > [2] Suyoung Kim, Seonguk Park, Junhoo Lee, and Nojun Kwak. The role of teacher calibration in
> > knowledge distillation. IEEE Access, 2025.
> >
> > [3] Wen-Shu Fan, Su Lu, Xin-Chun Li, De-Chuan Zhan, and Le Gan. Revisit the essence of distilling
> > knowledge through calibration. In Forty-first International Conference on Machine Learning, 2024b.
> >
> > [4] Laurent Valentin Jospin, Hamid Laga, Farid Boussaid, Wray Buntine, and Mohammed Bennamoun.
> > Hands-on bayesian neural networks—a tutorial for deep learning users. IEEE Computational
> > Intelligence Magazine, 17(2):29–48, 2022.
> >
> > [5] Agustinus Kristiadi, Matthias Hein, and Philipp Hennig. Being bayesian, even just a bit, fixes
> > overconfidence in relu networks. In International conference on machine learning, pp. 5436–5446.
> > PMLR, 2020.
> >
> > [6] Jakob Gawlikowski, Cedrique Rovile Njieutcheu Tassi, Mohsin Ali, Jongseok Lee, Matthias Humt,
> > Jianxiang Feng, Anna Kruspe, Rudolph Triebel, Peter Jung, Ribana Roscher, et al. A survey of
> > uncertainty in deep neural networks. Artificial Intelligence Review, 56(Suppl 1):1513–1589, 2023.
> >
> > We thank the reviewer for their time and careful consideration of our work. We appreciate the constructive feedback and thoughtful comments, and we hope that our responses address the concerns raised.

---

> ### Comment · Reviewer_Faxx · 2025-11-26
>
> Thank the authors for their detailed response. Most of my concerns are addressed, but the the lack of empirical results remains my major concern. The authors compare their proposed method with MCMI and MSE teachers in their experiments; while both compared methods have shown experimental results with various distillers and on different datasets, it's unknown whether the proposed method in this paper has the same level of generalizability. Even if the proposed method is limited to response-based KD, there're still plenty of distillers to try, such as DKD, DIST and WTTM [1-3].
>
> [1] Borui Zhao et al., "Decoupled Knowledge Distillation", CVPR 2022.\
> [2] Tao Huang et al., "Knowledge Distillation from A Stronger Teacher", NeurIPS 2022.\
> [3] Kaixiang Zheng & En-Hui Yang, "Knowledge Distillation Based on Transformed Teacher Matching", ICLR 2024.

---

> > ### Author Response · Authors · 2025-12-03
> >
> > We are delighted to read that our previous reply has alleviated most of the reviewer's original concerns.
> > Regarding the remaining request for additional experiments, based on the reviewer's suggestion, we have now included the following additional studies in our revised paper (the results are included here, and full details are provided in **Appendix J**):
> > 1. **Evaluation with alternative distillation losses**: DKD, DIST, and WTTM, suggested in the reviewer's comment, define enhanced loss functions for response-based distillation. As such, they can be readily applied with our proposed method for setting the teacher model. Indeed, as we numerically show, our usage of Bayesian teachers combined with stochastic gradient-based learning leads to improved student performance across all three losses compared to using deterministic teachers. We are thus grateful to the reviewer for encouraging us to demonstrate this additional aspect of our work.
> > 2. **Additional dataset**: as suggested by the reviewer, we further include experiments on an additional dataset, Tiny ImageNet, distilling a ResNet-34 teacher into ResNet-18 and WRN-16-2 students. The trends observed in this newly added study are fully in line with those reported so far, demonstrating that our findings hold over additional datasets.
> >
> > All reported values in the tables correspond to accuracies expressed as percentages.
> > |  | Teacher architecture | Student architecture | Student from Deter. DIST teacher | Student from Bayesian DIST teacher | Student from Deter. DKD teacher | Student from Bayesian DKD teacher | Student from Deter. WTTM teacher | Student from Bayesian WTTM teacher |
> > |----|----------------------|----------------------|-------------------|------------------|------------------|-----------------|-------------------|------------------|
> > | 1  | ResNet-18            | ResNet-18            | 75.01             | 76.93            | 74.86            | 77.14           | 76.05             | 76.89            |
> > | 2  | ResNet-50            | ResNet-18            | 75.39             | 77.19            | 75.75            | 77.21           | 77.11             | 77.28            |
> > | 3  | ResNet-50            | ResNet-34            | 74.52             | 75.23            | 76.10            | 77.29           | 76.54             | 77.23            |
> > | 4  | WRN-40-2             | WRN-16-2             | 70.84             | 72.53            | 71.04            | 73.32           | 72.16             | 73.54            |
> > | 5  | WRN-40-2             | WRN-40-1             | 68.92             | 70.69            | 69.77            | 71.68           | 70.78             | 70.82            |
> > | 6  | VGG-13               | VGG-8                | 75.43             | 77.10            | 75.41            | 77.85           | 76.81             | 77.64            |
> > | 7  | ResNet-50            | WRN-40-2             | 73.40             | 76.04            | 74.11            | 75.90           | 75.34             | 75.39            |
> > | 8  | ResNet-50            | VGG-8                | 75.42             | 76.88            | 75.78            | 77.79           | 77.40             | 77.94            |
> > | 9  | VGG-13               | ResNet-18            | 75.32             | 77.34            | 75.68            | 77.61           | 75.68             | 77.43            |
> > | 10 | VGG-13               | WRN-40-1             | 68.61             | 71.81            | 69.78            | 71.86           | 71.71             | 70.12            |
> > | 11 | ResNet-50            | WRN-16-2             | 70.45             | 73.03            | 70.82            | 74.10           | 73.82             | 73.86            |
> > | 12 | VGG-13               | WRN-40-2             | 73.41             | 76.10            | 74.02            | 75.66           | 74.78             | 74.71            |
> >
> >
> > | | Teacher architecture | Student architecture | Deter. Teacher Top-1 | Deter. Teacher Top-5 | Bayes. Teacher Top-1 | Bayes. Teacher Top-5 | Student No-KD Top-1 | Student No-KD Top-5 | Student (Deter. Teacher) Top-1 | Student (Deter. Teacher) Top-5 | Student (Bayes. Teacher) Top-1 | Student (Bayes. Teacher) Top-5 |
> > |---|----------------------|----------------------|--------------|---------------|----------------|----------------|-------------|-------------|---------------------------|---------------------------|---------------------------|---------------------------|
> > | 1 | ResNet-34 | ResNet-18 | 59.76 | 80.85 | 60.68 | 81.73 | 58.47 | 80.24 | 62.40 | 82.71 | 64.77 | 84.63 |
> > | 2 | ResNet-34 | WRN-16-2  | 59.76 | 80.85 | 60.68 | 81.73 | 52.17 | 77.23 | 55.62 | 79.13 | 56.83 | 80.46 |
> >
> >
> > We hope that these newly added studies alleviate the last remaining concerns of the reviewer.

---

### Official Review · Reviewer_FywD · 2025-10-31

**Soundness:** 4
**Presentation:** 3
**Contribution:** 3
**Rating:** 8
**Confidence:** 3

**Summary:**

This paper aims to provide a theoretical analysis of logit-based knowledge distillation from a Bayesian perspective. The authors provide analyses for both perfect BCPs and noisy BCPs, drawing the conclusion that knowledge distillation can lead to variance reduction and neighborhood term removal. Based on the theoretical findings, the authors further propose to utilize Bayesian deep learning models to improve effectiveness.

**Strengths:**

1.	This paper is well organized, highly detailed, and balanced between theoretical depth and readability.
2.	The theoretical analysis is supported by empirical evidence.
3.	There is potential practicality, as the authors also show the benefit of converting pre-trained models into BNNs to improve the effectiveness of knowledge distillation.

**Weaknesses:**

1.	(Minor) The analysis is based on SGD, but the experiments are conducted on Adam. Although this can show that the analysis also applies to other SGD-related optimizers, it would be better if there were some analyses or at least citations to show such generalizability from a theoretical perspective.
2.	(Minor) The experiments are based on image classification solely. Could there be more complex tasks, such as semantic segmentation or object detection?
3.	(Minor) Some related work is recommended to be discussed, such as [1, 2]

[1] ABKD: Pursuing a Proper Allocation of the Probability Mass in Knowledge Distillation via α-β-Divergence. ICML 2025

[2] f-Divergence Minimization for Sequence-Level Knowledge Distillation. ACL 2023.

**Questions:**

See Weaknesses.

---

> ### Author Response · Authors · 2025-11-20
>
> We thank the reviewer for their comments and positive evaluation of our work. We address the weaknesses and questions raised by the reviewer.
>
> **W1.** We thank the reviewer for this valuable observation. While our theoretical analysis focuses on SGD, our experiments indeed use Adam to demonstrate that the observed variance-reduction effect extends to adaptive gradient methods in practice. This empirical generalization is in fact supported by recent theoretical studies, e.g., [1] showing that Adam admits convergence guarantees under PL-condition and smoothness assumptions similar to those used in our analysis, with convergence up to a noise-dependent neighborhood term (analogous to SGD). Since our results explicitly show that supervision via Bayes class probabilities (BCPs) reduces gradient noise, these findings provide theoretical backing for the empirical applicability of our conclusions beyond SGD. A short discussion referencing this theoretical perspective has been added to Section 4.2.
>
> **W2.** We thank the reviewer for the suggestion to explore additional tasks such as semantic segmentation or object detection. Our primary theoretical contribution is an analysis of how supervision with BCPs affects the convergence behavior of SGD, with the analysis tied to probabilistic supervision in classification tasks. Since our conclusions rely on how well the teacher approximates the true BCPs, the practical benefits are mostly for response-based distillation for classification, where the supervisory signal is a probability vector.
>
> For more complex tasks such as detection or segmentation, the loss usually involves multiple loss components (e.g., bounding box regression, objectness scores, feature distillation), many of which do not rely on class-posterior probabilities. For these tasks, the classification part contributes only a fraction of the total objective. In such cases, the improvement gained from a better BCP approximation would likely be diluted by the presence of other losses, resulting in marginal overall gains.
>
> That said, we view our analysis and algorithmic contribution as a foundation that can support future extensions beyond pure classification. While this work focuses on BCP-based supervision, Bayesian teachers naturally provide calibrated uncertainty estimates, and are able to distinguish between aleatoric and epistemic per-sample uncertainty. We are currently exploring directions of uncertainty-guided supervision, and we will definitely look into tasks beyond classification. We thank the reviewer for the suggestion and intend to explore such extensions in the future.
>
> **W3.** We thank the reviewer for suggesting additional related works. Following the recommendation, we added a brief discussion referencing these works in the related works section.
>
> [1] Xia, Lu, and Stefano Massei. "Influence of Hyperparameters on the Convergence of Adam Under the Polyak-Lojasiewicz Inequality." Numerical Mathematics and Advanced Applications ENUMATH 2023, Volume 2: European Conference, September 4-8, Lisbon, Portugal. Vol. 154. Springer Nature, 2025.

---

### Official Review · Reviewer_Ki2F · 2025-10-31

**Soundness:** 3
**Presentation:** 1
**Contribution:** 3
**Rating:** 8
**Confidence:** 2

**Summary:**

The given paper demonstrates, from a Bayesian theoretical perspective, that utilizing soft probabilistic outputs in Knowledge Distillation (KD) leads to the highest performance. Based on the SGD optimizer, it shows that when the teacher’s accurate Bayes Class Probabilities (BCPs) are used as target data, the performance surpasses that obtained using one-hot encoding. Furthermore, the paper analyzes how the deviation of errors and accuracy change as noise levels are added to the original BCP distribution.

**Strengths:**

1.	Originality:
The paper provides a mathematical proof, from a Bayesian theoretical perspective, explaining why using soft probabilistic outputs in KD leads to better performance under the setting of an SGD optimizer. As mentioned in the Related Work section, the authors generalize this theoretical result beyond special cases such as self-distillation or model compression to more general classification settings, which represents a clear contribution compared to prior research.

2.	Quality:
The paper offers mathematically rigorous explanations throughout all derivations, giving the theoretical sections a strong sense of completeness and internal consistency.

3.	Clarity:
The logical flow of the paper is well-structured, and the authors successfully connect theoretical findings with experimental results, presenting a coherent narrative from theory to practice. However, the extensive mathematical derivations make it somewhat difficult for readers to follow the core ideas and fully grasp the main contributions.

4.	Significance:
By providing a mathematically complete explanation for why soft probabilistic outputs improve performance in KD—a question that has not been sufficiently addressed in previous KD research—the paper offers meaningful theoretical significance within the field of knowledge distillation.

**Weaknesses:**

1.	In Figure 1, it would be beneficial to further quantify the amount of noise and present this quantitatively. Moreover, based on the plots of generalization error and test accuracy per epoch, it seems that the results were obtained using a single random seed. If the authors were to test with multiple seeds and compute the variance of generalization error and test accuracy per epoch for the four cases, it could more clearly demonstrate that the true Bayes probabilities exhibit significantly lower variance. Additionally, instead of using abstract expressions such as “less noisy probabilities” or “more noisy probabilities”, it would be clearer to explicitly specify the exact noise levels, which would also enable a direct comparison with one-hot labels.
2.	Although the paper’s overall contribution is meaningful, it is somewhat disappointing that it does not go beyond showing that using BCPs as labels for the student model improves training stability due to lower variance.
3.	The transition from Equation (10) to Equation (11) omits too many intermediate steps, making it difficult to follow the derivation. Furthermore, mathematical expressions are overly complex, and the overall explanations feel somewhat unfriendly and difficult to read.

**Questions:**

1.	Does the proposed SGD_Based_Knowledge_Dist property still hold if optimizers other than SGD - such as Adam or RMSprop - are used?
2.	Is the proposed method applicable only to classification tasks, or could a similar approach be extended to regression problems as well?
3.	In Table 1, under what specific conditions were the tests conducted? (For example, were results averaged over multiple runs with different random seeds, such as Seed 0 to Seed 5?)
4.	In line 272 on page 6, the paper states that “we model ϵ ∼ P_{ϵ} as zero-mean noise with variance ν and uncorrelated entries.” - How was the variance ν determined or chosen?

---

> ### Author Response · Authors · 2025-11-20
> **Response part (1/2)**
>
> We thank the reviewer for their comments and positive evaluation of our work. We address the weaknesses and questions raised by the reviewer.
>
> **W1.** We appreciate the important comment. Appendix E provides the full experimental details behind Figure 1. While the left and middle plots in Figure 1 indeed present a single error/accuracy curve for visualization clarity, the right-most plot (average generalization error gap) and all corresponding results in Appendix E are computed over multiple random seeds and include standard deviations. In particular, Figures 3 and 4 in Appendix E report both the average generalization error/test accuracy and the variability around these quantities, showing mean and standard deviation across multiple runs. These plots also contain direct comparisons with the one-hot label baseline as well as the perfect Bayes class probabilities (BCPs) case, and they report results for several noise levels. Additionally, the terms “less noisy” and “more noisy” are explicitly defined in Appendix E, corresponding to noise levels of 5 and 0.5, respectively. The terms “less noisy” and “more noisy” are used in the main paper to intuitively convey how the level of noise in the BCPs manifests in the learning curves, while the precise definitions and noise values, along with the corresponding mathematical details, are fully specified in the Appendix.
>
> **W2.** Thank you for appreciating our overall contribution. While our main contribution indeed lies in our theoretical analysis of distillation from BCPs and its ability to  reduce variance and improve stability, our work does  go beyond this analysis and leverages it to propose concrete distillation methods. Specifically, our theoretical results show that the quality of a teacher should be measured by how well it approximates the true BCPs, rather than by its raw classification accuracy. This insight led us to investigate Bayesian teachers. Empirically, we demonstrate that transforming deterministic teachers into Bayesian ones via the Laplace approximation, reduces their test accuracy, yet the resulting BCPs are closer to the true posterior and improve student performance. We also show this using BNNs trained with variational inference. In both cases, students trained from these Bayesian teachers consistently achieve higher test accuracy (Table 1) and exhibit more stable (less noisy) learning curves (Table 2). We believe this finding is valuable because it provides a practical and theoretically motivated guideline: users can improve deterministic student models by employing Bayesian teachers, even when these teachers are less accurate.
>
> **W3.** We agree that the transition from Equation (10) to Equation (11) is not immediate, and involves a nontrivial set of mathematical steps. For ease of presentation, these mathematical steps are delegated to Appendix C.7 (which was Appendix C.6 of the original submission), where each step in obtaining Equation (11) is stated and justified. We agree that these expressions may be perceived as overly complex and unfriendly to read.
> While our analysis is necessarily technical, we aimed at presenting similarly to other theoretical papers in this area. Nevertheless, we agree that clarity is important, and we have aimed to keep the presentation as clear as possible while still balancing the necessary mathematical depth.
>
> **Q1.** Our theoretical findings are derived for learning based on SGD. We believe that rigorously extending these characterizations to adaptive optimizers like Adam or RMSProp requires further work as their data-dependent step sizes do not necessarily comply with the assumptions used in our analysis. However, to demonstrate that our theoretical findings can be extended to other optimizers, we also used Adam in our experimental study. These experiments show the same qualitative effects, which are more stable learning (reduced noise in learning curves) and improved performance (higher test accuracy), indicating that the empirical benefits of using BCPs are not restricted to SGD. Moreover, recent studies [1] provide convergence guarantees for Adam, up to a neighborhood term determined by the gradient noise, under assumptions similar to those used in our work. Since we demonstrate that learning from BCPs reduces this noise, we added this reference to the paper and will further investigate providing convergence guarantees for our setting with the Adam optimizer in the future.

---

> > ### Author Response · Authors · 2025-11-20
> > **Response part (2/2)**
> >
> > **Q2.** Thank you for the question. Our current derivation is tailored for classification, as the proposed method relies on using BCPs as soft targets for the student, which naturally arise when each input is associated with a probability distribution over discrete classes. Since our framework fundamentally compares probability distributions, it does not directly extend to standard regression tasks, where the target is typically a continuous-valued variable rather than a categorical distribution. While probabilistic regression models that provide full predictive distributions (e.g., [2,3]) could, in principle, enable an analogous formulation, doing so would require a dedicated and nontrivial theoretical reformulation that may not align with common distillation practices. We therefore view such extensions as an interesting direction for future work.
> >
> > **Q3.** The results displayed in Table 1 are indeed the average of 5 independent runs. Both the teacher and the student were trained with varying random seeds between runs. While the implementation details of the experiments can be found in Appendix F, we modified the caption of Figure 1 to include a more detailed explanation of how the experiments were conducted. Moreover, the full experimental results, along with standard deviations, can be found in Appendix J.
> >
> > **Q4.** The variance $\nu$ is not a learned or data-dependent quantity, it is a modeling choice used to represent the level of inaccuracy in the BCPs produced by the noisy teacher. Its incorporation allows us to model teachers that provide an estimate of the BCP that is not fully accurate with different levels of deviation. In our experiments, we use different values of $\nu$ to illustrate and quantify how different noise levels affect convergence, rather than to optimize performance, as demonstrated in Appendix E.
> >
> > [1] Xia, Lu, and Stefano Massei. "Influence of Hyperparameters on the Convergence of Adam Under the Polyak-Lojasiewicz Inequality." Numerical Mathematics and Advanced Applications ENUMATH 2023, Volume 2: European Conference, September 4-8, Lisbon, Portugal. Vol. 154. Springer Nature, 2025.
> >
> > [2] Stirn, Andrew, et al. "Faithful heteroscedastic regression with neural networks." International Conference on Artificial Intelligence and Statistics. PMLR, 2023.
> >
> > [3] Becker, Philipp, et al. "Recurrent Kalman networks: Factorized inference in high-dimensional deep feature spaces." International conference on machine learning. PMLR, 2019.

---

### Official Review · Reviewer_afuu · 2025-11-01

**Soundness:** 3
**Presentation:** 3
**Contribution:** 3
**Rating:** 6
**Confidence:** 4

**Summary:**

The paper analyzes SGD when the student is supervised by (i) exact Bayes class probabilities (BCPs) and (ii) noisy BCP estimates, then argues for Bayesian teachers as better BCP estimators. Core technical claims: (a) when supervising with exact BCPs, the empirical optimization interpolates and classical SGD “neighborhood” terms vanish, enabling larger admissible stepsizes (Thms. 1–2); (b) with noisy BCPs, convergence rates include a variance (gradient-noise) term whose magnitude depends on how close the teacher is to the true BCPs (Thms. 3–4 and Prop. 3).

**Strengths:**

1.  Showing that the CE risk with BCP supervision shares the same minimizer as standard supervision (the Bayes posterior; the minimum equals (H(Y|X))), then establishing interpolation for the BCP-supervised objective (Props. 1–2), is crisp and well-grounded.

2. Thms. 1–2 remove the variance neighborhood term found in standard SGD and allow a wider stepsize range, formalizing a compelling optimization advantage of distillation from *accurate* probabilities.

3.  The Dirichlet perturbation appendix helps ensure the targets remain on the simplex and shows the main conclusions persist.

**Weaknesses:**

1. Prop. 3 weights Jacobian norms by (1/P(y_k|x)) (or (1/P(y_k|x)^2) with noisy BCPs). If any class probability can be arbitrarily small, the gradient-noise bounds can blow up. You should make explicit an assumption like (P(y_k|x)\ge \epsilon>0) (or work with smoothed targets) and reflect this in all statements depending on Eqs. (13)–(14).

2. Additive perturbations can leave the simplex. While Appendix D covers a Dirichlet alternative, the main text should either use the Dirichlet model (preferred) or state an explicit projection/renormalization step and argue it does not break linearity in the second argument of CE used in the proofs.

3. Prop. 2 proves interpolation for the *BCP-supervised* objective under AS4. It would help to make explicit that interpolation is in the sense of *matching the Bayes distribution* (not zero training error on one-hots). Also, connect more tightly to Def. 1 and spell out that interpolation implies zero gradient at every sample (Eq. (25)), which underpins the disappearance of neighborhood terms. A short lemma bridging these steps in the main text (not only App. C) would aid readability.

**Questions:**

AS1/AS2/AS3 are standard in optimization but nontrivial for deep CE losses. Two concrete requests:

   (a) Explain where **expected smoothness** (AS3) comes from for CE with typical architectures (e.g., via bounded logits/Jacobians).

   (b) In Thms. 2–4 the stepsize bounds use “(\mu/(LL))” and “(\mu/(2LL))”. Please define “(LL)” or fix the notation—likely (L) vs. another constant (L'). As written, it’s ambiguous.

---

> ### Author Response · Authors · 2025-11-20
>
> We thank the reviewer for their careful review and thoughtful comments.
>
> We revised the manuscript in response to each of the weaknesses and questions raised by the reviewer, as detailed in our answers.
>
> **W1.** Thank you for bringing up this subtle but important point. Indeed, the expressions in Prop. 3 can in principle become unbounded if the class posterior is allowed to be arbitrarily close to zero. This phenomenon is not unique to our setting: it is a known caveat in analyses involving the cross-entropy / log-loss, whose definition becomes problematic when probability masses vanish.
>
> In the theoretical machine-learning literature, this issue is typically resolved by explicitly assuming that the underlying probability mass function is bounded away from zero, or equivalently that distributions lie in the interior of the probability simplex. This assumption appears, for example, in theoretical analyses learning with logistic and cross-entropy losses (see, e.g., [1, Sec. 2.1], [2. Sec. 4.1]. [3. Sec. VII-C]). It is implicitly enforced in deep-learning analyses through softmax parameterizations, which guarantee strictly positive class probabilities. Following the reviewer’s suggestion, we now make this assumption explicit in the revised version (Section 3.2).
>
> **W2.** As the reviewer noted, additive perturbations may leave the simplex. We use this noise model in the theoretical development because it leads to a clean and transparent analytical characterization of the effect of perturbations on convergence. This choice makes the exposition of the main ideas easier to follow, while the more technically involved Dirichlet alternative is presented in Appendix D to show that the conclusions continue to hold when the perturbations are constrained to the simplex. To address the reviewer’s concern, we have revised the main text to explicitly highlight this distinction and to point the reader to Appendix D. We also emphasize that all numerical experiments in the theoretical section use Dirichlet-perturbed probability vectors which remain on the simplex.
>
> **W3.**  Agreed. We have made the following changes to the main body of the manuscript: (a) we added the proposed clarification regarding interpolation, after Proposition 2; (b) we connected Definition 1 more tightly to Proposition 2; and (c) as suggested, we included a new short lemma (with the proof in the appendix) which shows that interpolation implies zero gradient of each per-sample loss. The lemma aids in clarifying why the desired interpolation property yields the vanishing neighborhood term in the convergence bounds.
>
> **Q1.**  Thanks for this suggestion. We added a short explanation following the expected smoothness assumption in the main body clarifying how CE being smooth in logits alongside locally bounded network Jacobians yields expected smoothness. The claim that such boundness is often induced via activations and/or training methods is based on [4].
>
> **Q2.**  Thanks for the keen observation. The symbol $\mathcal{L}$ represents the expected smoothness bound on the stochastic gradients, as defined in AS3. The symbol $L$ represents the smoothness bound on the risk, as in Definition 2.24 of Garrigos \& Gower [5]. In the original submission, the latter was only stated in the Appendix. Following your observation, we now include the definition of $L$ also in the main body in the statement of AS3 for completeness.
>
> [1] Haussler, David. "Decision theoretic generalizations of the PAC model for neural net and other learning applications." Information and computation 100.1 (1992): 78-150.
>
> [2] Cesa-Bianchi, Nicolo, and Gabor Lugosi. "Worst-case bounds for the logarithmic loss of predictors." Machine Learning 43.3 (2001): 247-264.
>
> [3] Painsky, Amichai, and Gregory W. Wornell. "Bregman divergence bounds and universality properties of the logarithmic loss." IEEE Transactions on Information Theory 66.3 (2019): 1658-1673.
>
> [4] Hardt, Moritz, Ben Recht, and Yoram Singer. "Train faster, generalize better: Stability of stochastic gradient descent." International conference on machine learning. PMLR, 2016.
>
> [5] Garrigos, Guillaume, and Robert M. Gower. "Handbook of convergence theorems for (stochastic) gradient methods." arXiv preprint arXiv:2301.11235 (2023).
>
> We appreciate the reviewer’s careful reading and the mathematically detailed suggestions, which have helped us tighten the presentation and improve the precision and clarity of our paper.

---

> ### Comment · Reviewer_afuu · 2025-11-20
> **Response to the Author's Rebuttal**
>
> Thank you for your detailed response to my concerns and questions. After a careful review of your responses, my concerns have been mainly addressed, and thus I will keep my positive score.

---

### Comment · Area_Chair_cxm7 · 2025-11-22

Dear Reviewers,

Thank you for your time and effort in reviewing submissions for ICLR  2026. As we begin the author-reviewer discussion process, we kindly remind you to submit your responses to the author rebuttals by **December  2**.


Your engagement in this discussion phase is crucial to ensuring a fair and thorough evaluation of each submission.

**Action Required**


- Carefully consider the authors’ rebuttal and any additional evidence they provide.

- Update your review (if applicable) to reflect your revised perspective.

-  **Discuss with the authors if further details are required**


Your AC

---

### Meta-Review · Area_Chair_X7th · 2025-12-17

**Summary:**

This paper studies knowledge distillation (KD) from a Bayesian perspective, with a particular focus on the optimization dynamics of SGD when students are trained on probabilistic supervision. The authors analyze two regimes: (i) distillation from exact Bayes Class Probabilities (BCPs) and (ii) distillation from noisy approximations of BCPs, modeling imperfect teachers. Under standard assumptions, they show that learning from BCPs eliminates the usual SGD neighborhood term (variance at convergence), while noisy BCPs interpolate between this ideal regime and standard one-hot supervision. The theoretical analysis is complemented by design guidelines advocating Bayesian teachers, and extensive CIFAR-100 experiments showing improved student accuracy and significantly reduced training variance relative to deterministic teachers.

The reviewers generally agreed that this paper presents a technically sound and rigorous theoretical analysis of SGD-based knowledge distillation from a Bayesian perspective, with a clear focus on how supervision with exact or noisy Bayes class probabilities affects convergence, variance, and stability. Theoretical contributions were viewed positively and regarded as well grounded. The main concerns raised by reviewers fell into four categories:

1. Strength and clarity of assumptions and derivations. Reviewers asked for clearer statements regarding assumptions such as bounded class probabilities, expected smoothness for cross-entropy losses, and the interpretation of interpolation when supervising with BCPs. They also noted some notation ambiguities and gaps in the logical exposition linking interpolation to vanishing gradient noise.

2. Mismatch between theory (SGD) and experiments (Adam). Several reviewers questioned whether conclusions derived for SGD extend to adaptive optimizers used in experiments, and requested either theoretical justification or clearer discussion.

3. Empirical scope and generalizability. One reviewer in particular expressed concern that experiments were initially limited to CIFAR-100 and standard response-based KD, and requested validation across additional datasets and distillation losses to better support general claims.

4. Presentation density and accessibility. While the theory was considered rigorous, some reviewers felt the exposition was mathematically dense and could be difficult to follow without additional clarifications.

During the rebuttal and revision phase, the authors effectively addressed these concerns. They clarified and strengthened assumptions, fixed notation issues, added missing explanatory lemmas, expanded the discussion on expected smoothness and optimizer generalization, and significantly broadened the experimental evaluation to include additional distillation losses, an additional dataset, and further baselines. After these revisions, most reviewers explicitly indicated that their concerns were resolved.

Overall, the remaining concerns are primarily about scope and idealized assumptions rather than correctness, and the reviewers broadly agree that the paper makes a meaningful and well-supported contribution to the theoretical understanding of knowledge distillation.

**Reviewer Concerns:**

Concerns addressed by the rebuttal:

1. Theoretical clarity and assumptions. Reviewers’ concerns about bounded class probabilities, expected smoothness for cross-entropy losses, notation ambiguities, and the link between interpolation and vanishing gradient noise were addressed through clearer assumptions, corrected notation, added explanations, and an additional clarifying lemma.

2. Modeling of noisy BCPs. The authors clarified the use of additive noise in the main text, explicitly pointed to the Dirichlet-based simplex-preserving alternative, and emphasized that experiments use valid probability vectors.

3. SGD vs. Adam mismatch. Although the theory is derived for SGD, the authors added discussion and citations showing that similar noise-dependent convergence behavior holds for Adam under comparable assumptions, supported by empirical evidence.

4. Limited empirical scope. Initial concerns about experimental breadth were addressed by adding experiments with additional distillation losses, an additional dataset, and extra baselines.

Concerns still partially outstanding:

1. Idealized assumptions. The analysis relies on standard but strong optimization assumptions (e.g., PL condition, expressivity), which may not strictly hold in practice.

2. Scope of tasks. The framework is limited to response-based classification distillation and does not cover regression or structured prediction tasks.

**Reviewer Scores:**

Reviewer afuu (initial score: 6 – marginal accept): After the rebuttal, the reviewer explicitly stated that their concerns were mainly addressed and that they would keep their positive score. I expect this reviewer would remain at 6.

Reviewer Ki2F (initial score: 8 – accept): This reviewer was positive about the contribution and significance from the outset, with concerns primarily about presentation and experimental clarity. These concerns were addressed in the revision, and there was no indication of dissatisfaction after rebuttal. I expect the score would remain at 8.

Reviewer FywD (initial score: 8 – accept): This reviewer viewed the paper as strong and well supported, with only minor concerns about optimizer generalization and experimental breadth. These issues were addressed in the rebuttal and revision. I expect the score would remain at 8.

Reviewer Faxx (initial score: 4 – marginal reject): This reviewer initially raised concerns about novelty and limited empirical validation. After the rebuttal and subsequent revision, the reviewer acknowledged that most concerns were addressed, with remaining hesitation focused on empirical breadth. Given the added experiments, I expect this reviewer would move upward to approximately 6 (borderline to weak accept).

---

### Decision · Program_Chairs · 2026-01-26

Accept (Poster)